

# SOIL GREENHOUSE GAS EMISSIONS UNDER DIFFERENT LAND-USE TYPES IN SAVANNA ECOSYSTEMS OF KENYA

**Authors:** *Sheila Wachiye[1,2,5], Lutz Merbold[3], Timo Vesala[2], Janne Rinne[4], Matti Räsänen[2], Sonja Leitner[3], and Petri Pellikka[1, 2]

1) Earth Change Observation Laboratory, Department of Geosciences and Geography, University of Helsinki, Finland

2) Institute for Atmosphere and Earth System Research, University of Helsinki, Finland

3) Mazingira Centre, International Livestock Research Institute (ILRI), Nairobi, Kenya

4) Department of Physical Geography and Ecosystem Science, Lund University, Sweden

5) School of Natural Resources and Environmental Management, University of Kabianga, Kenya

*Correspondence email: sheila.wachiye@helsinki.fi

## Abstract

For effective climate change mitigation strategies, adequate data on greenhouse gas (GHG) emissions from a wide range of land-use and land cover types is a prerequisite. However, GHG field measurement data are still scarce for many land-use types in Africa, causing a high uncertainty in GHG budgets. To address this knowledge gap, we present *in situ* measurements of carbon dioxide ($CO_2$), nitrous oxide ($N_2O$), and methane ($CH_4$) emissions in the lowland part of southern Kenya. We conducted chamber measurements on gas exchange from four dominant land-use types (LUTs) including: 1) cropland, 2) grazed savanna, 3) bushland, and 4) conservation land. Between 29 November 2017 to 3 November 2018, eight measurement campaigns were conducted accounting for regional seasonality (including wet and dry seasons and transitions periods) in each LUT. Mean $CO_2$ emissions for the whole observation period were significantly higher (p-value<0.05) in the conservation land ($75\pm6$ mg $CO_2$-C m$^{-2}$ h$^{-1}$) compared to the three other sites, which ranged from $45\pm4$ mg $CO_2$-C m$^{-2}$ h$^{-1}$ (bushland) to $50\pm5$ mg $CO_2$-C m$^{-2}$ h$^{-1}$ (grazing land). Furthermore, $CO_2$ emissions varied between seasons, with significantly higher emissions during the wet season than the dry season. In contrast, mean $N_2O$ emissions were not different between the four sites, ranging from $1.2\pm0.4$ µg $N_2O$-N m$^{-2}$ h$^{-1}$ (in bushland) to $2.7\pm0.6$ µg $N_2O$-N m$^{-2}$ h$^{-1}$ (in cropland). However, $N_2O$ emissions were slightly elevated during the early days of the wet season. $CH_4$ emissions did not show any significant differences between LUTs and seasons, and most values were below the limit of detection (LOD, 0.03 mg $CH_4$-C m$^{-2}$ h$^{-1}$). We attributed the difference in soil $CO_2$ emissions between





the four sites to soil C content, which also differed between the sites and was highest in the conservation land. $CO_2$ and $N_2O$ emissions positively correlated to soil moisture, thus an increase in soil moisture led to an increase in emissions. Soil temperature did not show a clear correlation with either, most likely due to the low annual variation in soil temperature. We found a strong positive correlation between soil $CO_2$ and the normalized difference vegetation index (NDVI), but we

observed no correlation with soil $N_2O$ emissions. We conclude that soil moisture is a key factor in soil GHG emissions in these tropical savanna LUTs. In addition, including vegetation indices in the model greatly improved the results, thus showing the importance of vegetation cover in predicting soil emissions. Our results are within the range of previous GHG flux measurements from soils from various land-use types in other parts of Kenya and contribute to more accurate baseline GHG emission

estimates from Africa, which are key for informing policymakers when discussing low-emission development strategies.

**KEYWORDS: Carbon Dioxide, Nitrous Oxide, Methane, Bushland, Conservation, Grazing land, Cropland.**



## 1. Introduction

Soil is a major source, and in many cases also a sink, of the atmospheric greenhouse gases (GHG) carbon dioxide ($CO_2$), nitrous oxide ($N_2O$), and methane ($CH_4$) (Oertel et al., 2016). The concentrations of these gases have increased since the onset of industrialization in 1970, leading to global warming (IPCC, 2013). GHGs trap the long-wave radiation emitted by the Earth's surface, thus increasing surface temperatures (Arrhenius, 1896). The production and consumption of GHGs in soil largely depend on the physical and chemical properties of the soil (Davidson et al., 2006) (e.g. texture, soil organic matter (SOM), pH) and are driven by environmental factors such as soil moisture and soil temperature (Davidson et al., 2006; Stehfest and Bouwman, 2006). Soil GHG emissions and uptake along with their controlling factors therefore differ between biomes. In addition, land use and its management greatly influence soil GHG emissions.

Conversion and overutilization of lands, such as cultivation and the use of fertilizers, excessive removal of vegetation, residue burning or tree planting (Baggs et al., 2006; Hickman et al., 2015), affect the physical and biological properties of the soil (Kim and Kirschbaum, 2015). This further affects soil carbon (C) and nitrogen (N) cycling, thus influencing the exchange of GHGs between the soil and atmosphere (Muñoz et al., 2010). Land-use changes are reportedly the largest source of anthropogenic GHG emissions in Africa (Valentini et al., 2014). However, *in situ* studies on the effect of land use and land-use change on soil GHG emission from various ecosystems still remain limited, particularly in savanna ecosystems (Castaldi et al., 2006).

Savanna is an important land cover type in Africa, covering more than 40 % of the total area (Scholes et al., 1997). In Kenya, savanna and grassland ecosystems collectively known as Arid and Semi-Arids (ASALs) occupy approximately 80 % of the total land area under various land-use types (GoK, 2013). Dominated by open grasslands, shrublands, and scattered woody vegetation, ASALs include national parks and reserves supporting nearly 80 % of Kenya's wildlife population. ASALs further encompass major rangelands and group ranches, supporting approximately 50 % of the Kenyan livestock population (GoK, 2013). The remaining land cover within ASALs outside of conservation land, national parks, and ranches are mainly dryland-agriculture croplands, dry thickets, and bushland.

Savannas in Africa are subject to accelerating land-use change (Grace et al., 2006), which has been attributed to population growth (Meyer and Turner, 1992) and land-use management activities (Valentini et al., 2014). Conversion of savanna for large-scale livestock production, cultivation, and human settlement is a common occurrence in Africa (Bombelli et al., 2009). This transition affects



vegetation cover, net primary productivity, and the above- and belowground allocation of C and nutrients in plants, thus affecting soil organic carbon (SOC) (Burke et al., 1998) and soil respiration (Abdalla et al., 2018; Carbone et al., 2008). For example, poor land management associated with overstocking and overgrazing is a major cause of soil and vegetation degradation in most African

savannas (Patton et al., 2007; GoK, 2013; Abdalla et al., 2018). Other factors associated with grazing include animal feeding preferences of certain plant species (thus creating higher pressure for certain species, which decline in numbers over time, subsequent lower pasture nutritive value, and species loss) (Patton et al. 2007). Soil trampling increases soil bulk density and decreases soil water infiltration (Patton et al., 2007). In addition, high rates of dung and urine deposition, especially around

homesteads and waterholes, create high N concentrations toxic for many savanna grass species, potentially affecting vegetation cover. Given that all these factors affect soil properties, they also influence soil GHG emissions in such ecosystems (Wilsey et al., 2002).

In addition to increasing grazing and overstocking pressures, rapid population growth leads to more people migrating into savanna ecosystems (Pellikka et al., 2018), which has also led to cropland

expansion (Pellikka et al., 2018; Patton et al., 2007). Brink and Eva (2009) found that cropland increased by 57 % between 1975 and 2000 and natural vegetation decreased by 21 % in Africa. In the years between 1990 and 2010, cropland in the Horn of Africa increased by ca. 28 % (Brink et al., 2014), while wooded vegetation cover in East Africa decreased by 5.1 % in forests, 15.8 % in woodlands, and 19.4 % in shrublands (Pfeifer et al., 2013). In Taita Taveta County in Kenya, cropland

area increased from 30 % in 1987 to 43 % in 2011 (Pellikka et al., 2018). In the Taita Hills, located in the Taita Taveta County, this trend has slowed down in past years, but croplands are still being cleared from natural vegetation at an alarming rate in the foothills and savanna lowlands (Pellikka et al., 2013), which has also impacts on land surface temperatures, for example (Abera et al., 2019 submitted).

Croplands in Kenyan savanna ecosystems are mostly managed by smallholder farmers with land sizes less than two hectares (Waswa and Mburu, 2006). Due to high poverty levels in this region, inputs to improve crop yields, such as fertilizer and herbicide utilization and mechanized farming are limited (Waswa and Mburu, 2006; CIDP, 2014), thus an increase in productivity is mostly achieved via cropland expansion. Regardless of the challenges faced by savanna cropland farming, the diversity

and a large number of smallholder farms are likely to have a substantial effect on national GHG emissions (Pelster et al., 2017). Until now, only a few studies have investigated GHG emissions from agricultural soils in Africa (Rosenstock et al., 2016), and these studies are mostly from high-potential



areas for agriculture such as the Kenyan highlands. For example, Rosenstock et al. (2016) showed a large variation of $CO_2$ and $N_2O$ emissions both within and between crop types as affected by
environmental conditions and land management in four crop types. Studies measuring GHG emissions from low-productivity croplands in southern Kenya are still missing, to the best of our knowledge.

Given the vast area covered by savanna, enhanced C and N emissions from land-use and land cover changes, are likely to affect both global, regional, and national C and N cycles, hence quantification
of their contribution is paramount (Lal, 2004; Williams et al., 2007). Studies in Kenya have showed large variations of soil GHG emissions in various savanna ecosystems (Otieno et al., 2010; Oduor et al., 2018), as a result of land-use (Ondier et al., 2019) and management activities (K'Otuto et al. 2013), and have highlighted the importance of savannas in the regional C balance. Studies have also been undertaken in other savanna ecosystems (Castaldi et al., 2006; Grace et al., 2006; Livesley et
al., 2011). However, due to the high diversity and large area within these ecosystems, such studies may not be entirely representative (Ardö et al., 2008). In addition, few studies have compared GHG emissions from various land uses, such as between agricultural soils and natural soils, as Pelster et al. (2017) did for 59 smallholder farms in western Kenya.

The lack of reliable soil GHG emission data from natural savanna and cropland limits our
understanding of GHG emissions from African soils (Hickman et al., 2014; Valentini et al., 2014). Therefore, there is a crucial need for accurate quantification of GHG emissions from multiple land uses to provide well-defined and reliable baseline data to allow more accurate estimation of Kenya's national GHG inventory (IPCC, 2006). This is particularly important, as Kenya currently relies on a Tier-1 approach, i.e. using default emission factors (EFs) provided in the Guidelines for Greenhouse
Gas Inventories of the UN Intergovernmental Panel on Climate Change (IPCC) to estimate national GHG emission inventories. Following the Paris Climate Agreement (https://unfccc.int/process-and-meetings/the-paris-agreement/d2hhdC1pcy), most countries across the globe, including Kenya, have not only agreed to accurately report their GHG emissions at national scales following a Tier-2 approach but also to begin mitigating anthropogenic GHG emissions in the upcoming decades, as is
communicated via Nationally Determined Contributions (NDCs). To achieve both, accurate locally derived data are essential.

To address this limitation of GHG emissions from various land-use types (LUTs), our study aims at: (1) providing crucial baseline data on soil GHG emissions from four dominant land uses, namely conservation land, grazing land, bushland, and cropland, and (2) investigating abiotic and biotic



drivers of GHG emissions during different seasons. We hypothesized that GHG emissions in cropland
       will be higher compared to grazing land, bushland, and conservation land because of larger nutrient
       inputs (i.e. fertilization) in managed land. Further, we hypothesized that GHG emissions will differ
       between seasons; more precisely, we expected higher GHG emissions in the wet season than in the
       dry season because of higher soil moisture.

## 2.    Materials and Methods
### 2.1. Study area

       This study was conducted in the lowlands (800–1000 m a.s.l.) of Taita Taveta County, southern
       Kenya (Fig. 1), between latitude 3° 25´ S and longitude 38° 20´ E. Taita Taveta County is one of
       Kenya's ASAL regions, with 89 % of the county area characterized by semi-arid and arid conditions.
The county is divided into three major geographical regions, namely the mountainous zone of the
       Taita Hills (Dawida, Kasigau, Sagalla), Taita lowlands, and the foot slopes of Mt. Kilimanjaro around
       Taveta.

       In the lowlands, which is our study site, vegetation types include woodlands, bushlands, grasslands,
       and riverine forests/swamps. Tsavo East and Tsavo West National Parks cover ca. 62% of the county
area for wildlife conservation (CIDP, 2014). The parks are open savanna and bush woodland that
       support elephants, buffaloes, lions, antelopes, gazelles, giraffes, zebras, rhinoceroses, and a wealth of
       birdlife. There are 28 ranches designated for livestock production and two wildlife sanctuaries (Taita
       Hills Wildlife Sanctuary and LUMO Community Wildlife Sanctuary). Livestock in the region is
       mainly managed in the form of nomadic pastoralism and only limited ranching occurs (CIDP, 2014).
Other important land use includes croplands with dryland agriculture in small-scale farming
       operations with low farm inputs (CIDP, 2014), dry thickets and shrublands, and sisal farming
       (Pellikka et al., 2018). The main soil type is characterized by dark red, very deep, acid sandy clay soil
       (Ferralsols). Our study sites were located in four of these key land uses in the region, namely cropland,
       bushland, wildlife conservation land, and grazing land.

The lowland area has a bimodal rainfall pattern with two rainy seasons – a long rainy period between
       March and May and a short rainy period between October and December (CIDP, 2014). The hottest
       and driest months are January and February, while the dry season from June to October is cooler
       (Pellikka et al., 2018). Mean annual rainfall is 500 mm and the average annual air temperature is
       23 °C, with an average daily minimum temperature of 16.7 °C and a maximum temperature of 28.8 °C
(CIDP, 2014).

The cropland site is located in Maktau (1070 m a.s.l, Fig. 1). The farm measured approximately one and a half hectares, with maize (Zea mays L.) intercropped with beans as the main crops. The farm can be considered rain-fed smallholder agriculture. Crop growing closely follows the rainy seasons, with maize and beans sowed in March, and bean and maize harvesting occurring in June and August,

respectively. Other crops on the farm included cowpeas, pigeon peas, cassava, and sweet potatoes. Land preparation was performed by ploughing using animal traction before seeding, while weeding was performed by hand hoe. Small quantities of manure were used to improve soil fertility by applying approximately 20 kg of mixed dry and fresh animal manure (less than 1 kg of N) every month during the campaign period (Fig. 2a).

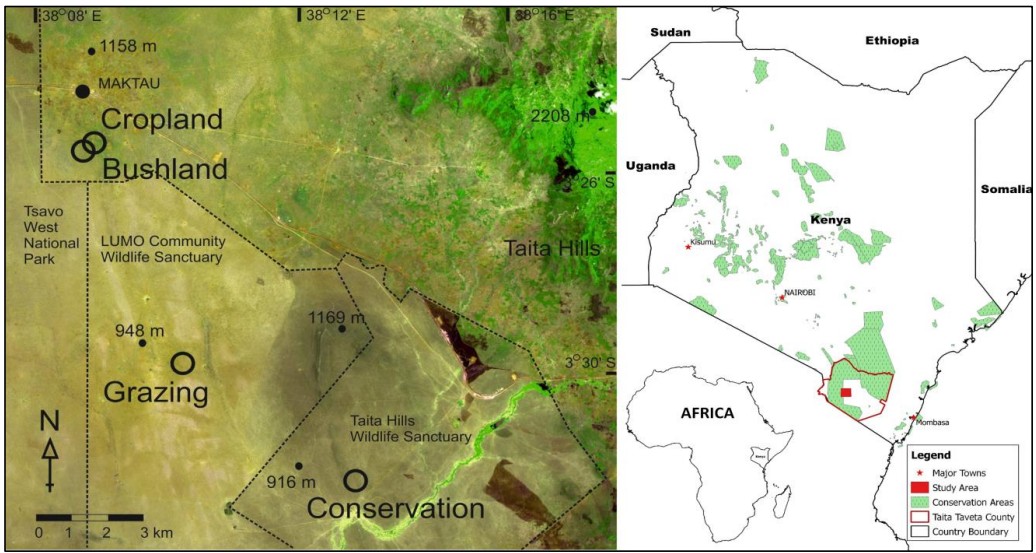


*Figure 1. Location of the study sites cropland, bushland, grazing land, and conservation land in the savanna area in the lowlands of Taita Taveta County in southern Kenya. Image showing the sites is Sentinel-2A acquired from Sentinel's Scientific DataHub (ESA, 2015)*

The second site is located in private bushland in Maktau next to the cropland (Fig. 1, Fig. 2c). In Taita
Taveta, bushland is found both within the conservation areas and under private ownership. In this region, bushland forms a cover with over 50 % of thorny shrubs and small trees, characterized by *Acacia spp and Commiphora ssp.* The bushes may vary in height from two to five metres. Herbs and savanna grasses (mostly annual or short-lived perennials) less than one metre tall form the ground cover. On private bushland similar to our study site, individual farmers usually maintain woodlots of
trees and shrubs as part of their farms, where they generate a small income from forest products such



as timber, poles, and firewood, and charcoal to some extent. Additionally, some grazing occurs on the bushland,  primarily by livestock owned by the farmer (CIDP, 2014).

The grazing land is located in the LUMO Community Wildlife Sanctuary (Fig. 1) next to Tsavo West National Park and Taita Hills Wildlife Sanctuary and covering approximately 460 km$^2$. The sanctuary was formed by merging three ranches, namely the Lualenyi and Mramba communal grazing areas and the Oza group ranch, which were given the name "LUMO". This sanctuary is designated for livestock grazing by local communities, with wildlife also present, as conservation areas are not necessarily fenced. No individual land ownership occurs, as the land is communally owned (GoK, 2013). Overgrazing is a major challenge, especially by herders who enter the conservancy illegally, leaving the soil bare for most of the year but especially during the dry season (CIDP, 2014). During this time, livestock is forced from LUMO into the Taita Hills Sanctuary conservation land because of the open boundary it shares with LUMO (Fig. 2d).

The conservation land is located within the Taita Hills Wildlife Sanctuary (Fig. 1). Covering an area of ca. 110 km$^2$ hectares, it is a private game sanctuary for wildlife conservation located between LUMO and communal land. The sanctuary is an open savanna grassland dominated by *Schmidtia bulbosa* and *Cenchrus ciliaris* grass species forming an open to closed ground cover, scrublands, and scattered woodlands with *Acacia spp.* as main tree species. However, most trees have been damaged by elephants, leaving the landscape more open. The sanctuary is well managed with the application of ecological management tools such as controlled fires. Through these and other conservation efforts, the sanctuary has attracted a higher diversity of large mammals, many of which remain within the unfenced sanctuary throughout the year. Wildlife are the predominant grazers and browsers, although livestock encroachment may be a problem especially during the dry season on the western and eastern borders of the sanctuary (GoK, 2013) (Fig. 2c).


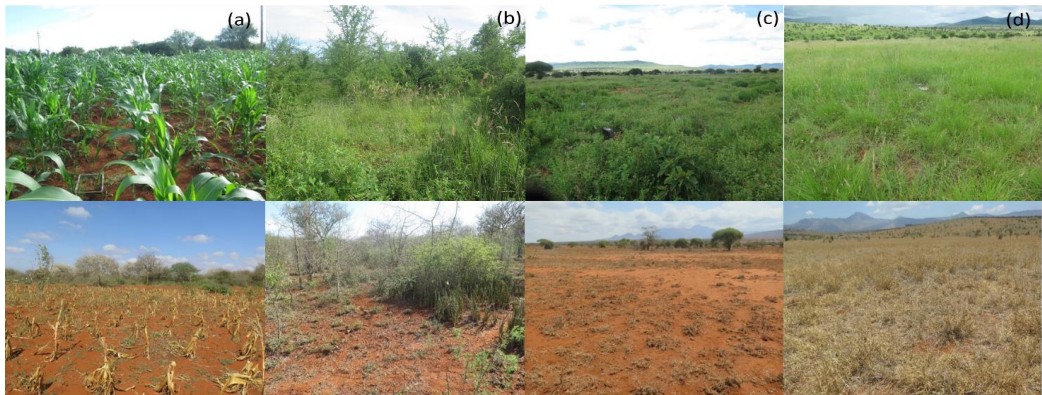


***Figure 2:*** *The four land-use types: (a) cropland, (b) bushland, (c) grazing land, and (d) conservation land. The upper panel shows the land-use types during the wet season, while the the lower panel depicts the situation during the dry season. The grey plastic collars visible in upper left photo are frames for the GHG flux chambers.*

### 2.2. Defining the Seasons


We divided the campaigns into dry and wet seasons based on an agro-climatic concept. Therefore, the onset date of the wet season was the first wet day of a 3 day wet spell receiving at least 20 mm without any 10 day dry spell (< 1 mm) in the following 20 days from 1st of March for the long wet season and 1st of September for the short wet season (Marteau et al., 2011). Equally, the end of the rainy season was the first of 10 consecutive days during which no rain occurred. Thus, for this study,

the long wet season (LW) was between 2 March to 4 June 2018, and the short wet season (SW) between 23 October and 26 December 2018. The two wet seasons were separated by two dry seasons, the short dry season (SD) from January to February 2018, and the long dry season (LD) from June to September 2018. We had three campaigns during each wet season: during the early days of wet

season onset (onset-SW, onset-LW), at the peak of the of the seasons (mid-SW, mid-LW), and towards the end of the seasons (end-SW, mid-LW).

### 2.3. Chamber measurements of greenhouse gas emission

Soil-atmosphere exchange of $CO_2$, $N_2O$, and $CH_4$ were measured in eight one-week campaigns between 29 November 2017 to 3 November 2018 using the static chamber method (Rochette, 2011;

Hutchinson et al., 1981). Within each of the four sites, three locations (clusters) were randomly selected and used as replicates for soil GHG concentration measurements. In each cluster, three plastic collars (27 cm × 37.2 cm × 10 cm) were inserted (5–8 cm) into the soil at least 24 hours before



the first sample was taken (see Pelster et al., 2017). The collars were left in the ground for the entire

measurement period to minimize soil disturbance before the time of measurements (Søe et al., 2004).

Only damaged or missing chamber bases (mostly due to livestock or wildlife activity) were replaced,

at least 24 hours before the next gas sampling. During each day of a campaign, gas sampling was

conducted daily between 7:00 am and 11:00 am, which represent the average flux of the diurnal cycle

(Shi et al., 2012; Davidson et al., 1998).

During each gas-sampling day, grey opaque PVC lids (27 cm × 37.2 cm × 12 cm) covered with

reflective tape were placed onto the collars for 30 mins. Lids were fitted with a fan for gas mixing

and a vent to avoid pressure differences between the chamber headspace and outside atmosphere

(Pelster et al.. 2017). A rubber seal was fitted along the edges of the chamber lid and paper clips were

used to hold the lid and collar in place to ensure airtightness. Four gas samples were then collected

every 10 minutes (time 0, 10, 20, 30 minutes) after lid deployment (Rochette, 2011). The height of

each chamber collar was measured on each sampling date to derive the total chamber volume (total

chamber height = height of chamber collar sticking out of the soil + height of the chamber lid). A

slightly modified version of the gas-pooling method was used to reduce overall sample size while

ensuring a good spatial representation of each LUT (see Arias-Navarro et al., 2013). Here, 20 ml of

headspace air were collected from each of the three chambers at each time interval with polypropylene

syringes, resulting in a composite gas sample of 60 ml. The first 40 ml were used to flush the vials,

and the remaining 20 ml were pushed into 10 ml glass vials, leading to a slight overpressure to

minimize contamination of the gas with ambient air during transportation (Rochette et al., 2003).

Following sample collection, gas samples were transported to the laboratory (Mazingira

Centre, mazingira.ilri.org) and analysed using a gas chromatograph (GC, model SRI 8610C gas

chromatograph). The GC was fitted with a $^{63}$Ni-Electron Capture Detector (ECD) for detecting $N_2O$

concentrations and a Flame Ionization Detector (FID) fitted with a methanizer for $CH_4$ and $CO_2$

analysis. The GC was operated with a Hayesep D packed column (3 m, 1/8″) at an oven temperature

of 70 °C, while ECD and FID detectors were operated at a temperature of 350 °C. Carrier gas ($N_2$)

flow rate was 25 mL min$^{-1}$ on both FID and ECD lines. In every 40 samples analysed with the GC

were eight calibration gases with known $CO_2$, $CH_4$, and $N_2O$ concentrations in synthetic air (levels of

calibration gases ranged from 400 to 2420 ppm for $CO_2$, 360 to 2530 ppb for $N_2O$, and 4.28 to 49.80

ppm for $CH_4$). Therefore, the gas concentrations of the samples were calculated from peak areas of

samples in relation to peak areas of standard gases with known concentrations using a linear model

for $CO_2$ and $CH_4$ and a power regression for $N_2O$.



### 2.4. Greenhouse gas flux calculations

Soil GHG emissions were determined by the rate of change in gas concentration in the chamber headspace over time by linear fitting. The goodness of fit was used to evaluate the linearity of concentration increases/decreases. The dynamics of the $CO_2$ concentrations over the 30 mins deployment period for each gas concentration was assessed to test for chamber leakage due to the typically more robust and continuous flux of $CO_2$ (Collier et al., 2014). If the linear model of $CO_2$ versus deployment time had an $R^2 > 0.95$ using all four time points (T1, T2, T3, and T4), the measurement was considered valid and four time points were used for analysing for $CO_2$, $N_2O$, and $CH_4$ emissions. However, if $R^2 < 0.95$ for $CO_2$ and one data point was a clear outlier, this point was discarded and the three remaining points used for the flux calculation if they showed a strong correlation of $CO_2$ versus time. Measurements that did not show a clear trend of $CO_2$ with time were considered faulty, and the entire data point was discarded. In addition, data points that showed a decrease in $CO_2$ concentration over time were assumed to indicate leakage and were thus discarded (chambers were opaque, i.e. photosynthesis was inactive during chamber deployment). However, if no leakage was found, negative $CH_4$ and $N_2O$ emissions were accepted as the uptake of the respective gas by the soil. Emissions were calculated according to Eq. (1):

$$F_{GHG} = \frac{(\frac{dc}{dt}) \times V_{ch} \times M_w}{A_{ch} \times Mv_{corr}} 60 \times 10^6 \qquad (1)$$

where $F_{GHG}$ = soil GHG flux ($CO_2$, $N_2O$, or $CH_4$), $\partial c/\partial t$ = change in chamber headspace gas concentration over time (i.e. slope of the linear regression), $V_{ch}$ = volume of the chamber headspace ($m^3$), $M_w$ = molar weight (g $mol^{-1}$) of C for $CO_2$ and $CH_4$ (12) or N for $N_2O$ (2x N = 28), $A_{ch}$ = area covered by the chamber ($m^2$) and $Mv_{corr}$ = pressure- and temperature-corrected molar volume (Brümmer et al., 2008) using Eq. (2). With 60 and $10^6$ being constants used to convert minutes into hours and microgramme respectively.

$$Mv_{corr} = 0.02241 \frac{273.15 + Temp(\text{℃})}{273.15} \times \frac{Atmospheric\ pressure\ at\ measurement\ (Pa)}{Atmospheric\ pressure\ at\ sea\ level\ (Pa)} \qquad (2)$$

The minimum limit of detection (LOD) for each gas was calculated following Parkin et al. (2012) and levels were $\pm 4.9$ mg $CO_2$-C $m^{-2} h^{-1}$ for $CO_2$, $\pm 0.04$ µg $N_2O$-N $m^{-2} h^{-1}$ for $N_2O$, and $\pm 0.03$ mg $CH_4$-C $m^{-2} h^{-1}$ for $CH_4$. However, we incorporated all the data in the analysis, including those below LOD in line with Croghan and Egeghy (2003), who noted that including such data





provides an insight on the distinct measurements, thus giving a clarification on the set of environmental observations.

### 2.5. Auxiliary measurements

During each gas-sampling day, we measured soil moisture and soil temperature (at a depth of 0-5 cm) adjacent to the collar using a ProCheck handheld data logger with a GS3 sensor (Decagon Devices Inc). Daily air temperature and precipitation data from November 2017 to November 2018 were obtained from a weather station in Maktau located within the cropland site (Tuure et al., 2019). A soil auger was used to collect soil samples (at a depth of 0-20 cm) during the wet season (22 May 2018) from each land-use type for soil chemical and physical property analysis. For bulk density, we collected a combination of three samples from each cluster close to each chamber collar at depths of 0–10 cm and 10–20 cm using a soil bulk density ring (Eijkelkamp Agrisearch Equipment, Giesbeek, The Netherlands). Samples were stored in airtight polyethylene bags and kept in a cooler box with ice packs before transportation to the laboratory for further analysis. In the laboratory, samples were stored in a refridgerator (4 °C) and analysed within 10 days.

The samples were sieved at < 2 mm before analysis. Soil water content was measured by drying soil at 105 ºC for 48 hours. Soil pH was determined in a 1:2.5 (soil:distilled water) suspension using an electrode pH meter (3540 pH and conductivity Meter, Bibby scientific Ltd, UK). We measured soil texture using the hydrometer technique (Scrimgeour, 2008; Reeuwijk, 2002). Total soil C and N content were analysed using a C/N elemental analyser as follows. A duplicate of 20 g of fresh sample was oven-dried at 40 ºC for 48 hours and ground into a fine powder using a ball mill (Retsch MM400). Approximately 200 mg of the dry sample was measured by elemental analysis (Vario MAX Cube Analyzer Version 05.03.2013).

### 2.6. Statistical Analysis

All statistical analyses were carried out using R 3.5.2 (R Core Team). Spearman correlation coefficients were performed among the variables followed by the Kruskal Wallis test to assess significant differences of soil GHG emissions between the land-use types and across seasons. A post-hoc analysis involving pairwise comparisons using the Nemenyi test was performed for gases where significant differences exist. Significance was set at $p < 0.05$.

We used several functions to assess the correlation between soil GHG ($CO_2$ and $N_2O$) with soil temperature and soil water content based on the coefficient of determination ($R^2$), root-mean-square





error (RMSE) and Akaike's information criterion (AIC). There being no difference in the outputs, we present results from the Gaussian function (O'Connell, 1990) for the correlation between soil GHG

($CO_2$ and $N_2O$) emissions and soil temperature using Eq. (3), and a quadratic function for correlation with soil water content using Eq. (4).

$$Rs = ae^{(bT + cT^2)} \tag{3}$$

$$Rs = a + bWC + cWC^2 \tag{4}$$

We also evaluated the combined effect of soil temperature (*T*) and soil water content (*WC*) on soil

GHG ($CO_2$ and $N_2O$) using several functions. Having also found no significant difference in the function result outputs, we combined Eq. (3) and Eq. (4) into Eq. (5) to assess the combined effect of these two predictors on soil emissions.

$$Rs = e^{(aT + bT^2)} \times (cWC + dWC^2) \tag{5}$$

After no correlation with soil water content and soil temperature was observed, we included

information on vegetation cover as a predictor. We used Normalized Difference Vegetation Index (NDVI) products (MOD13Q1) from MODIS (Moderate Resolution Imaging Spectroradiometer) from https://ladsweb.modaps.eosdis.nasa.gov. NDVI quantifies vegetation vigour by measuring the difference between reflectance in near-infrared (which green chlorophyll-rich vegetation strongly reflects) and red wavelength areas (which vegetation absorbs) computed using Eq. (6):

$$NDVI = \frac{NIR + Red}{NIR + Red} \tag{6}$$

MODIS NDVI is generated from a 16 day interval at a 250 m spatial resolution as a Level 3 product. To cover our study period, we selected NDVI data that were within the dates of campaign. However, if no NDVI data fitted within our dates, we used data that were from less than five days before or after the campaign dates, assuming that no significant increase or decrease would occur in the

vegetation. The pixels containing the study sites were extracted based on the latitude and longitude of each site. When comparing various functions, linear functions were applied to the seasonal datasets of soil GHG emissions with NDVI to assess the contribution of vegetation indices on soil GHG emissions using Eq. (7):

$$Rs = a + bNDVI \tag{7}$$





We also assessed the combined effect of WC and NDVI on soil $CO_2$ emissions using Eq. (8).

$$Rs = a + bNDVI + (cWC + dWC^2) \qquad (8)$$

where *Rs* is soil GHG ($CO_2$ and $N_2O$), *T* is soil temperature, and *WC* soil volumetric water content ($m^3 \, m^{-3}$) while *a, b, c,* and *d* are model coefficients.

### 3.  Results

**3.1. Meteorological data**

During the 12 month study period, long rainy periods were observed from early March to the end of May, while short rainy periods were observed between early September and October (Fig. 3). The total annual rainfall was 550 mm, which was within the average amount of rainfall expected in the area (CIDP, 2014). Mean annual air temperature was 22.7 °C (min=16.7 °C, max=30.5 °C). January

was the hottest month (min=17.4 °C, max=31.9 °C), while June and July (min=14.5± 0.2 °C, max=27± 0.1°C) were the coolest.

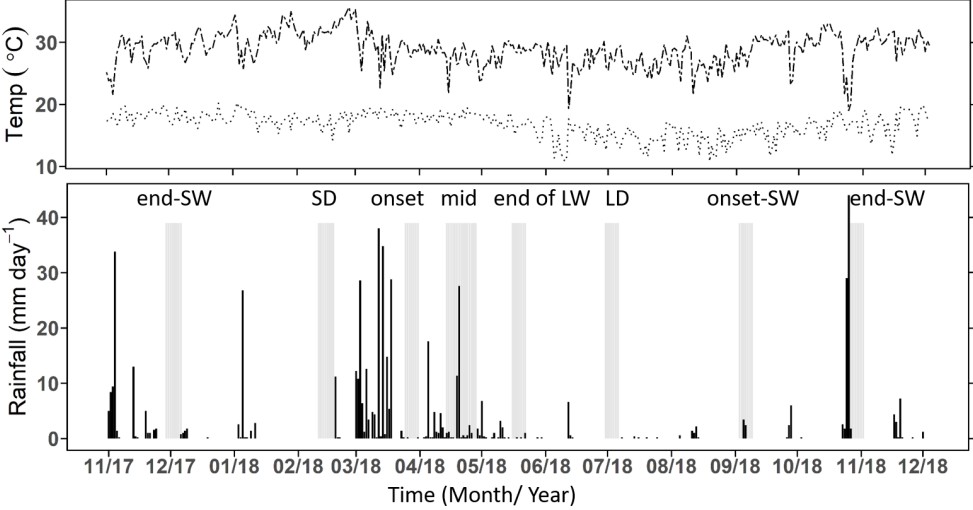

***Figure 3:*** *(a) Daily maximum and minimum air temperature and (b) daily rainfall from lowland in southern Kenya between November 2017 to October 2018 recorded at Maktau weather station. Total*

*annual rainfall recorded was 550 mm. Highlighted grey bars show the days of the sampling campaigns (the season above the grey bars denote SW and LW for the short and long wet season with corresponding onset, mid and end of the wet season, and SD for the short dry season and LD for the long dry season).*





### 3.2. Soil characteristics

Sand was the main component of the soils in all the sites (Table 2). Sand content was slightly higher in cropland (77±8 %) compared to the conservation area and bushland (ca. 72±1 %). Sand proportion was lower in the grazing land (64.3±0.4 %), while clay content was higher (31.7±0.5 %) than in the conservation area (26 ± 2 %), bushland (23.7±0.7 %), and cropland (19±2 %). Soil pH ranged between slightly acidic in the grazing land (6.3 ± 0.3), neutral in the bushland (7.2±0.4), and slightly alkaline

in the conservation area and cropland (7.5±0.1 and 7.9±0.2 respectively). Carbon content ranged from 0.93 % in the conservation land to 0.60 % in the cropland. Nitrogen content did not vary significantly between sites (mean=0.08±0.01 %).

**Table 1:** Soil characteristics of the topsoil (a depth of 0–20 cm) from the four land-use types investigated in this study. Values are given as mean ± SE.

| Land Use | % N | % C | Bulk Density (g cm$^{-3}$) | pH | Soil Texture | | |
| --- | --- | --- | --- | --- | --- | --- | --- |
| | | | | | % Clay | % Sand | % Silt |
| Bushland | 0.08 (0.03) | 0.77 (0.5) | 1.31 (0.2) | 7.2 (0.4) | 23.7 (0.7) | 71.6 (2.2) | 4.7 (2.3) |
| Conservation land | 0.09 (0.02) | 0.93 (0.7) | 1.27 (0.4) | 7.5 (0.1) | 26.4 (2.2) | 71.6 (0.5) | 2.0 (0.0) |
| Cropland | 0.07 (0.04) | 0.60 (0.2) | 1.26 (0.3) | 7.9 (0.2) | 19.1 (2.4) | 76.9 (8.1) | 4.0 (5.1) |
| Grazing land | 0.08 (0.02) | 0.83 (0.4) | 1.23 (0.5) | 6.3 (0.3) | 31.7 (0.5) | 64.3 (0.4) | 4.4 (0.4) |

### 3.3. Soil greenhouse gas emissions

#### 3.3.1. Soil carbon dioxide ($CO_2$) emissions

Mean annual soil $CO_2$ emissions were significantly higher in the conservation land (75±6 mg $CO_2$-C m$^{-2}$ h$^{-1}$) compared to the other three sites. Concurrently, no significant differences occurred in $CO_2$ emissions between grazing land (50±5 mg $CO_2$-C m$^{-2}$ h$^{-1}$), cropland

(47±3 mg $CO_2$-C m$^{-2}$ h$^{-1}$), and bushland (45±4 mg $CO_2$-C m$^{-2}$ h$^{-1}$). We observed no difference in the $CO_2$ emissions between the first three seasons, namely SD in February, and onset-LW in March and mid-LW in April. However, toward the end of the wet season (end-LW) in May, $CO_2$ emissions from the conservation land and grazing land were significantly higher than emissions from cropland and bushland ($p<0.05$). Through LD, onset-SW, and mid-SW, $CO_2$ emissions in the conservation land

remained significantly higher than emissions from the other three sites, while emissions from grazing land dropped during LD and were not different from bushland or cropland emissions thereafter.

Generally, $CO_2$ emissions were higher during the wet season than the dry season at all sites, showing a bimodal pattern (Fig. 4c). Just after onset of the rainy season in early March, $CO_2$ emissions



increased at all sites by over 200% from SD to LW and dropped during LD by approximately 70% in grazing land, bushland, and cropland. In the conservation land, the drop from LW to LD was about 20%. On average, the highest seasonal mean fluxes were observed during the wet season. In the bushland, the highest seasonal mean fluxes were reached in mid-LW ($98\pm6$ mg $CO_2$-C m$^{-2}$ h$^{-1}$) in early March. However, in the conservation land ($239\pm11$ mg $CO_2$-C m$^{-2}$ h$^{-1}$), grazing land ($160\pm16$ mg $CO_2$-C m$^{-2}$ h$^{-1}$), and cropland ($84\pm12$ mg $CO_2$-C m$^{-2}$ h$^{-1}$), the highest seasonal mean fluxes were observed during end-LW towards May. On the other hand, the lowest seasonal mean $CO_2$ emissions at all sites were observed during the SD campaign (below 20 mg $CO_2$-C m$^{-2}$ h$^{-1}$).

When comparing between the two wet seasons (LW and SW), $CO_2$ emissions were 45 % (bushland), 55 % (conservation land), 56 % (cropland), and 57 % (grazing land) higher during the long wet season than during the short wet season (Fig. 5a). For the two dry seasons, $CO_2$ emissions were also significantly higher in LD than SD across all the sites (in SD all sites recorded emission below 20 mg $CO_2$-C m$^{-2}$ h$^{-1}$). However, during the LD, $CO_2$ emission were 29 % (bushland), 38 % (cropland), 40 % (grazing land), and 77 % (conservation) higher than during SD (Fig. 5a). While $CO_2$ emissions in cropland, bushland, and grazing land had dropped below 30 mg $CO_2$-C m$^{-2}$ h$^{-1}$ during LD, $CO_2$ emissions were still high ($118\pm6$ mg $CO_2$-C m$^{-2}$ h$^{-1}$) in the conservation land, even more than during onset-LW ($63\pm9$ mg $CO_2$-C m$^{-2}$ h$^{-1}$) and mid-LW ($79\pm5$ mg $CO_2$-C m$^{-2}$ h$^{-1}$).

### 3.3.2. Soil nitrous oxide ($N_2O$) emissions

Mean annual $N_2O$ emissions were very low ($< 5$ µg $N_2O$-N m$^{-2}$ h$^{-1}$) at all four sites (Fig. 4d). Mean $N_2O$ emissions were higher in the cropland ($2.65\pm0.06$ µg $N_2O$-N m$^{-2}$ h$^{-1}$) than in the conservation land ($1.59\pm0.04$ µg $N_2O$-N m$^{-2}$ h$^{-1}$), grazing land ($1.49\pm0.04$ µg $N_2O$-N m$^{-2}$ h$^{-1}$), and bushland ($1.16\pm0.04$ µg $N_2O$-N m$^{-2}$ h$^{-1}$). $N_2O$ fluxes did not show a clear temporal pattern as observed for $CO_2$ emissions. Mean seasonal $N_2O$ emissions were very low during both the wet and dry seasons. Within each season, no significant differences were observed among the sites. At the onset of the rainy season (onset-LW), there were observable increases in $N_2O$ emissions from all the sites. During this period, mean $N_2O$ emissions at all the sites were ca. $2.6\pm0.4$ µg $N_2O$-N m$^{-2}$ h$^{-1}$). By mid-LW and end-LW periods, $N_2O$ emissions had dropped to nearly zero ($< 1$ µg $N_2O$-N m$^{-2}$ h$^{-1}$) at all sites. However, during LD in June, $N_2O$ emissions in cropland were significantly higher than at the other three sites ($2.35\pm0.03$ µg $N_2O$-N m$^{-2}$ h$^{-1}$, $p < 0.05$). During this period the farmer had just harvested his crops.

When comparing the two wet seasons, $N_2O$ emissions did not differ between LW and SW at all sites (Fig. 5b). However, short $N_2O$ emission pulses were observed during both seasons. A notable peak





of apprximately 70 µg $N_2O$-N $m^{-2}h^{-1}$ was observed in the cropland on 7 April 2018, which was after the application of livestock manure a week before the sampling campaign. It also rained the night before the sampling day. At the same site, we also recorded a peak of 55 µg $N_2O$-N $m^{-2}h^{-1}$ on 30 September 2018, likely also due to manure application (personal communication with the farmer Mwadime Mjomba). Other notable peaks were 29.9 µg $N_2O$-N $m^{-2}h^{-1}$ (in the bushland on 03

September 2018) and 26.6 µg $N_2O$-N $m^{-2}h^{-1}$ (in grazing land on 4 September 2018). These were observed during the SW from chambers with animal droppings within the chambers on these dates. During the dry season, $N_2O$ emissions did not differ between SD and LD in the bushland, conservation land, and grazing land, while $N_2O$ emissions in the cropland were significantly higher during LD than SD (Fig. 5b).

### 3.3.3. Soil methane ($CH_4$) emissions


Throughout the study period, $CH_4$ emissions did not vary significantly among sites and seasons (Fig. 4e and Fig. 5c). The studied sites were mostly $CH_4$ sinks rather than sources, and $CH_4$ emissions were very low, ranging from -0.03 to 0.9 mg $CH_4$-C $m^{-2}h^{-1}$ (Fig. 4e), and were often below the LOD (0.03 mg $CH_4$-C $m^{-2}h^{-1}$).




**Figure 4:** *Box plots showing differences in seasonal means for (a) soil moisture, (b) soil temperature, and soil emissions of (c) CO$_2$, (d) N$_2$O, and (e) CH$_4$ for each site from November 2017 to October 2018. Season abbreviations on the x-axis denote SW for the short wet season and LW for the long wet season with corresponding onset, mid and end of the wet season, along with SD for the short dry season and LD for the long dry season.*






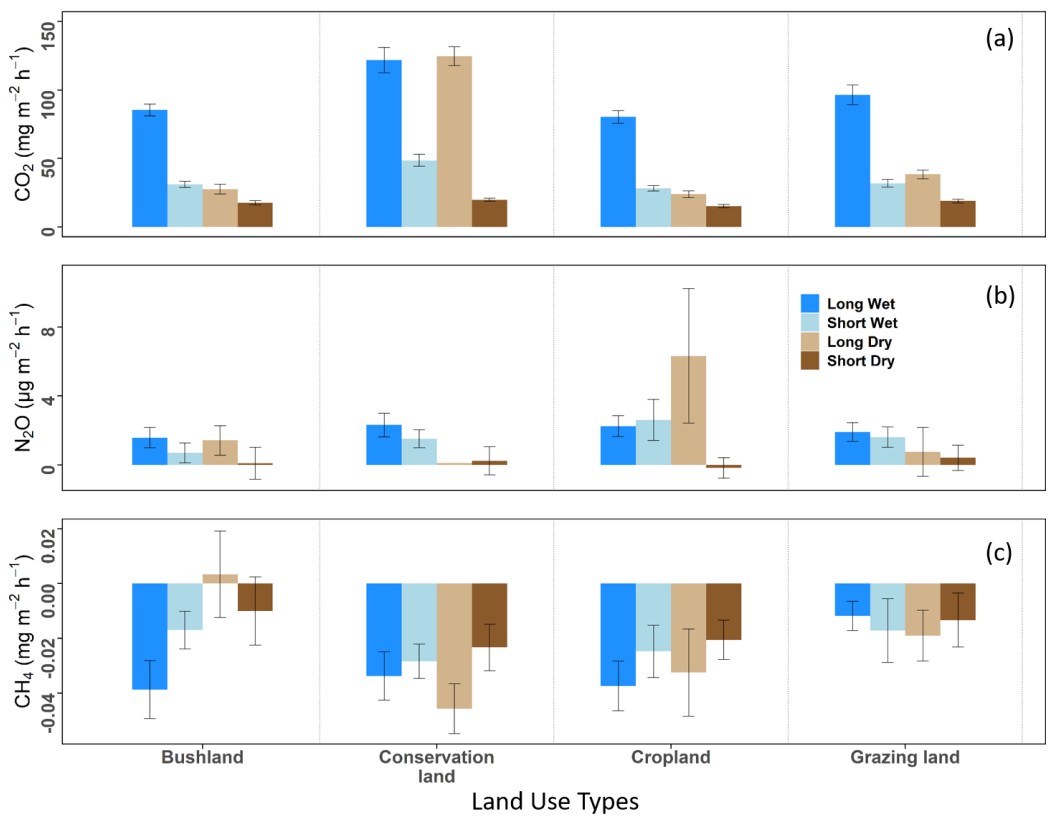

***Figure 5:*** *Seasonal differences in mean (a) CO$_2$, (b) N$_2$O, and (c) CH$_4$ emissions between the long and short wet seasons and the long and short dry season for the four land-use types.*

### 3.4. Effects of soil temperature, soil water content, and vegetation indices on soil GHG
emissions

Soil water content (WC) was highest during the wet season, ranging between 0.13 to 0.25 m$^3$ m$^{-3}$ and lowest during the dry season, ca. 0.07±0.02 m$^3$ m$^{-3}$ at all sites (see Fig. 4a). Soil temperature (T), on the other hand, were higher during the SD (36.7±2.1 $^\circ$C) at all the sites and lowest (24.5±0.6 $^\circ$C) during LD in July (Fig.4b). Throughout all the campaigns, mean WC and mean T were highest in the
conservation land, followed by grazing land, bushland, and lowest in the cropland.

Results from the nonlinear regression analyses on soil CO$_2$ and soil N$_2$O emission against T and WC are shown in Table 2. The results showed positive correlations between soil CO$_2$ emissions and WC (p <0.05). However, the R$^2$ was very weak at all sites. Conversely, CO$_2$ emissions showed no correlation with T (P < 0.05). There was no correlation between N$_2$O and CH$_4$ emissions with both





WC or T (p <0.05). Separating data into the wet and dry season did not improve any of the correlations as we had expected. However, we observed an increase in both $CO_2$ and $N_2O$ emissions at the onset of the rainy season in March.

**Table 2:** Soil water content (WC) and soil temperature (T) control on carbon dioxide ($CO_2$) and nitrous oxide ($N_2O$). Soil $CO_2$ and $N_2O$ emissions are denoted by *Rs*, while *a, b,* and *c* represent the

model coefficient

| Predictors | Land Use | $CO_2$-C mg m$^{-2}$ h$^{-1}$ | | $N_2O$-N ug m$^{-2}$ h$^{-1}$ | |
|---|---|---|---|---|---|
| Soil water | | $Rs = a + bWC + cWC^2$ | | | |
| content | Bushland | $6.12WC + 0.92WC^2$ | $R^2= 0.26$*** | $19.02WC - 64.11WC^2$ | $R^2= 0.008$ |
| (WC) | Conservation land | $135.27WC - 0.57WC^2$ | $R^2= 0.07$** | $11.63WC - 7.736WC^2$ | $R^2= 0.009$ |
| | Cropland | $17.83WC + 0.67WC^2$ | $R^2= 0.04$*** | $28.48WC - 66.63WC^2$ | $R^2= 0.005$ |
| | Grazing land | $15.03WC + 0.79WC^2$ | $R^2= 0.11$*** | $19.81WC - 53.56WC^2$ | $R^2= 0.002$ |
| Soil | | $R = ae^{(bT + cT^2)}$ | | | |
| Temperature | Bushland | $1.078e^{0.26T - 0.004T^2}$ | $R^2= 0.008$ | $360.25e^{-0.29T - 0.004T^2}$ | $R^2= 0.008$ |
| (T) | Conservation land | $0.001e^{0.81T - 0.014T^2}$ | $R^2= 0.015$** | $0.007e^{0.45T - 0.008T^2}$ | $R^2= 0.015$ |
| | Cropland | $4.568e^{-0.13T + 0.002T^2}$ | $R^2= 0.008$* | $0.007e^{-0.05T + 2.42T^2}$ | $R^2= 0.008$* |
| | Grazing land | $4.136e^{0.18T - 0.003T^2}$ | $R^2= 0.015$ | $2.366e^{0.05T - 0.001T^2}$ | $R^2= 0.015$ |

\*\*\*: p<0.0001, \*\*: p<0.001, \*: p<0.05

Results from combined soil water content and soil temperature on soil $CO_2$ and $N_2O$ emissions did not improve the correlation, as shown in Table 3. We concluded that soil emissions could be an overall effect of other factors than these two parameters. Thus, we chose to include vegetation indices in our

model.

**Table 3:** Combined effects of soil water content (WC) and soil temperature (T) control on soil $CO_2$ and $N_2O$ emissions. Soil $CO_2$ and $N_2O$ emissions denoted by *Rs*, while *a, b, d,* and *e* represent the model coefficient, ($R^2$) the coefficient of determination, and AIC the Akaike's information criterion

| Functions | Land use | a | b | d | e | $R^2$ | AIC |
|---|---|---|---|---|---|---|---|
| **$CO_2$ mg m$^{-2}$ h$^{-1}$** | | | | | | | |
| $Rs = e^{(aT + bT^2)} \times (dWC + eWC^2)$ | Bushland | -0.12 | 0.001 | 52.774 | -0.527 | 0.31*** | 1888 |
| | Conservation land | 0.90 | -0.016 | 0.0001 | 0.000 | 0.10*** | 2156 |
| | Cropland | -0.39 | 0.006 | 3701.901 | -84.001 | 0.08** | 1886 |
| | Grazing land | 0.14 | -0.003 | 0.842 | -0.008 | 0.12*** | 2024 |
| **$N_2O$ ug m$^{-2}$ h$^{-1}$** | Bushland | -0.50 | 0.007 | 2008.345 | -58.559 | 0.009 | 785 |
| $Rs = e^{(aT + bT^2)} \times (dWC + eWC^2)$ | Conservation land | 0.56 | -0.010 | 0.000 | 0.000 | 0.003 | 811 |
| | Cropland | 0.67 | -0.017 | 0.003 | -0.0001 | 0.089 | 911 |
| | Grazing land | 0.11 | -0.003 | 0.187 | -0.005 | 0.003 | 770 |

\*\*\*: p<0.0001, \*\*: p<0.001, \*: p<0.05





The annual change in vegetation cover and during each campaign in the study area are shown in Fig.

(6) and Fig (7) respectively. The highest NDVI values were observed in during the LW in April

(ranging from 0.58 to 0.76) and the lowest during the dry season (below 0.26). Vegetation greenness

in all the sites increased rapidly from mid-March, which coincided with the onset of the rainy season

and remained high. However, at end of the rainy season, NDVI gradually dropped. Highest NDVI

values occurred in the conservation land (0.51±0.05), followed by bushland (0.44±0.04), cropland

(0.41±0.05), and the lowest values were recorded in the grazing land (0.33±0.05).

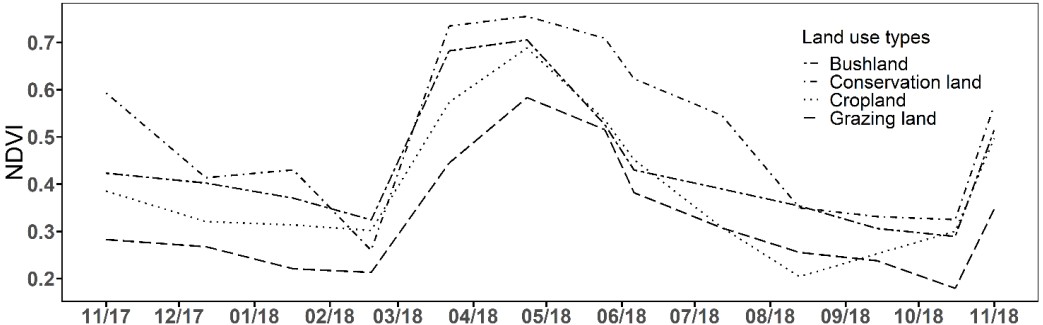

***Figure 6:*** *Monthly NDVI time series showing the annual trend in vegetation cover from November*
*2017 to November 2018 for the four land-use types.*




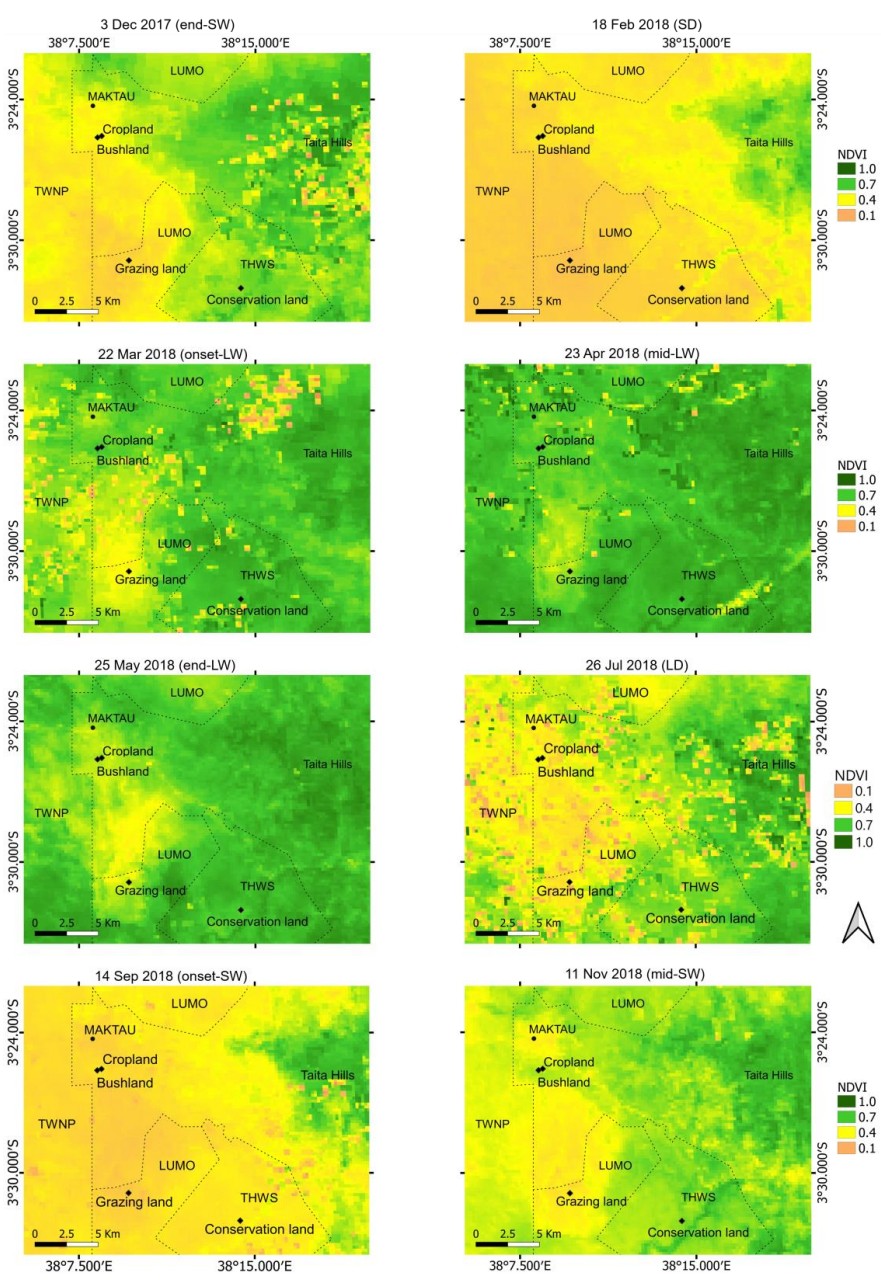

***Figure 7:*** *Normalized difference vegetation index (NDVI) maps for each campaign (SW and LW denotes short and long wet seasons with corresponding onset, mid and end of the season, and SD and LD for short and long dry season) from November 2017 to November 2018. LUMO is the LUMO community wildlife sanctuary, TWNP stands for the Tsavo west National Park and THWS for Taita Taita hills wildlife sanctuary*






Our results show that NDVI was capable of explaining a large degree in the variation of seasonal soil $CO_2$ emissions in the bushland, grazing land, and cropland (see Fig. 8). However, correlation in the conservation land was not significant. We also observed no significant correlation with $N_2O$ at all the

sites.

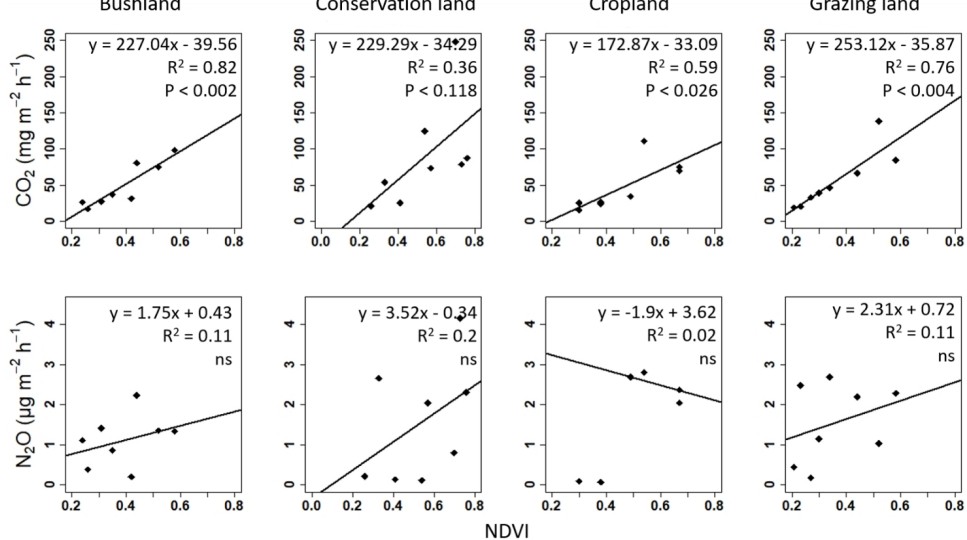

ns= not significant

***Figure 8:*** *Linear regression analyses of the measured seasonal means for soil $CO_2$ and soil $N_2O$ emissions during the campaign from November 2017 to November 2018 plotted against NDVI data*

*acquired during each campaign.*

Results from combined WC and NDVI improved the correlation outputs further (Table 4). These results highlight the importance of WC and vegetation cover (as represented by NDVI) on soil $CO_2$ emissions in the study areas.






**Table 4:** Combined effects of soil water content (WC) and NDVI on soil $CO_2$ emissions. Soil $CO_2$ emissions denoted by *Rs*, while *a, b, c,* and *d* represent the model coefficient and ($R^2$) the coefficient of determination.

| | Land use | a | b | c | d | $R^2$ |
|---|---|---|---|---|---|---|
| **$CO_2$-C mg m$^{-2}$ h$^{-1}$** | | | | | | |
| $Rs = a + bNDVI + (cWC + dWC^2)$ | Bushland | -29.69 | 196.47 | -83.74 | 781.29 | 0.86*** |
| | Conservation land | 6.48 | 382.96 | -861.98 | -256.66 | 0.82*** |
| | Cropland | 26.94 | 244.54 | -1250.22 | 3269.46 | 0.79*** |
| | Grazing land | -97.19 | 396.41 | 575.60 | -3440.80 | 0.96*** |

***: p<0.0001, **: p<0.001, *: p<0.05


## 4. Discussion

### 4.1. Soil $CO_2$ emissions

$CO_2$ emissions from the soil differed significantly between the land-use types. Higher $CO_2$ emissions were particularly observed in the conservation land followed by grazing land and bushland, while

lowest levels were recorded from the cropland. We observed the same trend with SOC, and thus we attributed the difference in $CO_2$ emissions between the land uses to SOC as observed (see Table 1). This is in line with a similar study by La Scala et al. (2000), which recognized SOC as a key driver of $CO_2$ emissions from the soil, as it is the primary source of energy for soil microorganisms (Lal, 2009).

The differences in $CO_2$ emissions and SOC among our land-use types can be linked to the difference in vegetation cover type, which can alter both biotic and abiotic factors that drives soil $CO_2$ emissions (Raich and Tufekcioglu, 2000; Pinto et al., 2002) and net carbon assimilation (La Scala et al., 2000). Vegetation types directly influence soil physicochemical properties, which modify soil microbial activities (Raich and Tufekcioglu, 2000). This also affects the quantity of plant carbon allocated

belowground (Metcalfe et al., 2011). Vegetation additionally affects root respiration by determining root biomass, and litter quality and quantity (Fanin et al., 2011; Rey et al., 2011). Root respiration and the associated microbial components are important in ecosystem soil respiration. Active roots add directly to soil respiration, while the dead roots and exudates from the roots provides carbon as a source of energy and nutrients for microbial biomass (Tufekcioglu et al., 2001).

In the conservation land, thick grasses formed a closed ground cover, especially during the wet seasons (also confirmed by NDVI values). This translated into higher soil respiration due to higher root respiration due to a more active root network and litter production compared to the other sites (Raich and Tufekcioglu, 2000). Besides this, grazing here was low and only occurred by random



elephants and other wild mammals. We observed less damage to grass cover in the conservation area

compared to grazing land (which was mostly bare due to overgrazing) and bushland. Abdalla et al. (2018) found higher grazing intensity to reduce SOC stocks due to the modification of vegetation cover, which affects litter accumulation and decomposition (Wilsey et al., 2002). This could be the case with the grazing land, which was under heavy grazing by livestock, and also by wildlife. This explains the difference in mean $CO_2$ emissions between conservation land and grazing land.

In the bushland, grazing can be considered moderate when compared to the conservation land and grazing land, as only the farmer's livestock grazed the land. However, the lower $CO_2$ emissions and lower SOC compared to the grazing land can be a consequence of the higher clay content observed in grazing land (see Table 1). The presence of polyvalent cations in clay forms organo-mineral complexes that protect SOC from microbial and enzymatic decay, which in turn increases SOC

storage (Amanuel et al., 2018).

Soil $CO_2$ emissions and mean SOC were lowest in cropland. Root respiration in cropland depends on periods of live roots in annual crop fields and on the biomass of roots during the initial growing season (Raich et al., 2000). Therefore, the continued removal of crop residues during harvesting and frequent tillage affected both root respiration and SOC. As much as crop residues contribute to carbon stocks

through their mineralization (Nandwa, 2001), most maize residues were used as livestock feed and sometimes as fuel, while bean residues were removed completely during the harvest and burned.

Manure inputs provide easily degradable substrates of C and N catalyzing soil emissions (Janssens et al., 2001; Davidson and Janssens, 2006). However, manure input in cropland was very low (approximately 20 kg in a 1.5 ha farm per month) and thus no measurable effects in $CO_2$ emissions

were detected. This was opposite to our hypothesis. Another reason could be that soil fertility was too low to have a detectable influence on $CO_2$ emissions (Pelster et al., 2017). Our results are in the same magnitude with those of Rosenstock et al. (2016); Farai Mapanda et al. (2011), and Pelster et al. (2017), who also did not detect any change in $CO_2$ emissions after manure application and attributed this to the low input of manure from the maize and sorghum plots.

On average, $CO_2$ emissions were higher during the wet season than during the dry season. At the start of both rainy seasons, $CO_2$ emissions increased significantly in all land-use types. The emission from the conservation land and grazing land are comparable with those in Brümmer et al. (2008), who observed $CO_2$ emissions ranging between 100 and 250 mg $CO_2$-C $m^{-2}\,h^{-1}$ in a natural savanna in Burkina Faso. Several other studies from similar ecosystem have also documented comparable





changes in $CO_2$ emissions with the onset of the rainy seasons (Castaldi et al., 2006; Livesley et al., 2011; Pinto et al., 2002).

In the cropland, our results during the wet season are similar with those that Rosenstock et al. (2016) measured during the wet season, which ranged between 50 to greater than 200 mg $CO_2$-C $m^{-2}\,h^{-1}$. We attributed the increase in $CO_2$ emissions during the wet season to the response of soil microbes to soil

moisture thus producing an increase in ecosystem respiration (Livesley et al., 2011; Otieno et al., 2010). Soil moisture promotes soluble substrates and oxygen availability, both needed by soil microbes activities such as decomposition and soil respiration (Davidson et al., 2006; 2009; Grover et al., 2012).

Consequently, higher soil $CO_2$ emissions during the wet season can be a result of increased root

respiration due to more active plant and root growth (Macdonald et al., 2006). Grass sprouts rapidly after the rains, increasing in root network density to maximize the use of available soil moisture on the soil surface (Merbold et al., 2009). This is one explanation for the higher $CO_2$ emissions in the grassy conservation land, grazing land, and bushland compared to cropland without the grasses. The lower $CO_2$ emissions in bushland compared to conservation land and grazing land were attributed to

the presence of more trees and shrubs, which according to Merbold et al. (2009), respond more slowly than grasses to changes in soil moisture. Therefore, grass production belowground in the conservation land and grazing land was probably more than in the bushland, leading to higher autotrophic respiration and also heterotrophic respiration (Janssens et al., 2001).

To our surprise, the highest seasonal mean $CO_2$ emissions in conservation land, grazing land, and

cropland were observed at the end rather than the at the peak of the wet season. During this time, both documented soil moisture and soil temperature had dropped in all land-use types. We note here that soil moisture and temperature we measured upto 5 cm deep. Thus, we credited the relatively high $CO_2$ emissions to root respiration that could tap moisture from deeper profiles than microbial activity in the soil (Carbone et al., 2011). According to Carbone et al. (2011), microbial respiration peaks first

with surface soil moisture, whereas root respiration continues to increase throughout the wet season, controlling emissions as temperatures increase and surface soil moisture gradually drops. Most microbial activities are found on the soil surface, which are first to wet-up and dry-down with rainfall, while roots are located deeper, with access to more water reserves that take longer to be exhausted (Carbone et al., 2011).




### 4.2. N₂O emissions

Soil N$_2$O emissions can vary highly over time, as regulated by factors such as soil moisture, temperature, aeration, ammonium and nitrate concentrations, pH, and mineralizable carbon (Butterbach-Bahl et al., 2013). However, we did not document any significant difference in N$_2$O

emissions between the four land uses. At all the sites, N$_2$O emissions were very low, and this we attributed to the observed low soil N content (see Table 1). According to Pinto et al. (2002) and Grover et al. (2012), savanna ecosystems have a very tight N cycling, which transcends to low N availability. Thus, available N is taken up by vegetation, leaving very little for denitrification (Castaldi et al., 2006; Mapanda et al., 2011). Our results are consistent with low N$_2$O emissions

observed in a Brazilian savanna (cerrado) by Wilcke et al. (2005), who also reported low N levels in their study area. Very low N$_2$O emissions due to poor nutrient availability have also been observed in other savanna landscapes (Scholes et al., 1997; Castaldi et al., 2016). Soil N$_2$O in cropland match those of Rosenstock et al. (2016), who also attributed the low soil N$_2$O emissions to poor nutrient availability in the soil.

We did not detect seasonal variations in N$_2$O emissions. The only exception to otherwise very low N$_2$O emissions was after the onset of the rainy season, when N$_2$O emissions slightly increased at all sites. Such a pattern has previously been shown by Scholes et al. (1997) in South Africa. Several other studies have also reported comparable increase in N$_2$O emissions after the rainy season (Scholes et al., 1997; Pinto et al., 2002; Castaldi et al., 2006; Livesley et al., 2011). Soil moisture in the savanna

ecosystem controls soil gas diffusion, oxygen (O$_2$) availability for microbial use and the availability of substrate for microbial communities (Davidson et al., 2000; Butterbach-Bahl et al., 2013). Therefore, the increase in N$_2$O emissions at the onset of the wet season is possibly a response of microbial communities to variation in soil moisture (Rees et al., 2006). In addition, the decomposition of litter and plant residue facilitated by soil moisture may have further increased N availability.

Rosenstock et al. (2016) also recorded low N$_2$O emissions with average fluxes typically less than 12 µg N$_2$O-N m$^{-2}$ hr$^{-1}$ in rainy Kenyan highlands. In cropland, mean N$_2$O emissions are close to those in Mapanda et al. (2010) in Zimbabwe, who reported an average of 3.3-3.4 µg N$_2$O-N m$^{-2}$ hr$^{-1}$. In June and July, the slight increase after the maize and bean harvests could be due to there being no plants to take up the available N and thus some was lost as N$_2$O.

Negative N$_2$O emissions were detected during the dry season. Such observations could result from the poor N levels observed at all sites. Soil denitrifiers may therefore use N$_2$O as an N substrate in the absence of NO$_2^-$ and NO$_3^-$ (Rosenkranz et al., 2006). Negative N$_2$O emissions have also been





reported in other tropical savanna soils under similarly dry conditions (Donoso et al., 1993; Castaldi et al., 2006; Livesley et al., 2011).

Application of manure in the cropland did not show any significant differences, which is the opposite of what we expected. Several studies have shown both organic and inorganic fertilisers in agricultural land to increase $N_2O$ emissions (Davidson, 2009; Butterbach-Bahl et al., 2013; Hickman et al., 2014). Due to low nutrient levels, as observed, the addition of manure to these soils may not have been sufficient to stimulate high $N_2O$ emissions. However, manure input by the farmer was also very low

(less than 12 kg N ha$^{-1}$ for the crop-growing season). Our conclusion here is that the maize took most of the available N that was added, thus diminishing the pool of N to be lost as $N_2O$. According to the Taita Development plan, this is a common scenario in the county, which translates to very low yields (CIDP, 2014). Our results are similar to those in Pelster et al. (2017) (generally $< 10\,\mu g\ N_2O\text{-}N\ m^{-2}\,hr^{-1}$), who also observed no detectable influence of manure application (which

they noted was low, between 1–25 kg N ha$^{-1}$) on $N_2O$ emissions. Increased nutrient inputs and improved management are required to improve yield and livelihoods, but this may simultaneously lead to increased soil $N_2O$ emissions. Nevertheless, Hickman et al. (2015) suggested that managed agricultural intensification in western Kenya could increase crop yields without immediate increases in $N_2O$ emissions, if application rates remained at or below 100 kg N ha$^{-1}$. We believe this could also

be the case in our study area.

Deposition of faeces and urine by animals while grazing in the grazing land and bushland did not show any statistical difference with cropland and conservation land, as we expected. We found faeces in and close to chambers for most of the sampling days in the grazing land, although we did not see any animal urine in the chambers. In addition, on each sampling day, we also observed animal

footprints in our chamber or close by in the same site in the bushland. In the conservation area, as much as animal were grazing, we never observed faeces in or close to our chambers throughout the campaign period. We only noted the presence of animals close to our chamber from the footprints and the destruction of our chamber from time to time.

### 4.3. CH$_4$ emissions

Methane emissions did not vary between the land-use types or with seasons. Most values were below the LOD at all the sites, which is similar to observations made by Rosenstock et al. (2016) and Wanyama et al. (2019).



### 4.4 Effects of soil moisture, soil temperature, and vegetation indices

Soil moisture and soil temperature are known to be important drivers of soil $CO_2$ production, and they may change across seasons. Seasonal variations in T were very minor and thus changes in WC were considered to be the main driver of $CO_2$ emissions in our study, as previously highlighted by Grover et al. (2012). Brümmer et al. (2009) and Livesley et al. (2011) also found that WC controlled $CO_2$ fluxes from savanna soils, rather than T. Soil moisture determines the rate of soil respiration, including heterotrophic and autotrophic processes that are highly moisture reliant ( Ardö et al., 2008;

Grover et al., 2012).

Soil temperature and soil moisture are also vital for nitrification and denitrification, as they control microbial activities and Oxygen diffusion in the soil. However, we did not observe a significant relationship between $N_2O$ emissions with both WC and T. Only in the cropland area did we observe a positive correlation with T (p <0.05). Some previous studies by Scholes et al., (1997) and Brümmer

et al. (2008) carried out in savannas were also unable to link $N_2O$ emissions to variations in T but others showed a positive relationship like in Castaldi et al., (2010). $N_2O$ emissions were very low during both the wet and dry seasons, as in ( Castaldi et al., 2004), Pelster et al. (2017), and Rosenstock et al. (2016**).** The most likely clarification for the lack of seasonality would be the low N levels observed at all the sites (Grover et al., 2012).

The vegetation cover and its status affects GHG emissions from soils. NDVI measures the status of vegetation (with a value range of -1 to 1). High NDVI values correspond to high vegetation cover, while low NDVI corresponds to less or no vegetation ( Gamon et al., 1995; Butt et al., 2011). Therefore, the drop in NDVI values at the end of the rainy season was a result of the reduction in vegetation cover in the land-use types. In the cropland area, this coincided with the harvesting of

beans and the drying of the maize plant occurring in June and July. The conservation land showed the highest mean NDVI, mainly due to the dense grassy vegetation, especially during the rainy season. Lowest NDVI values were observed in the grazing land, which was expected because the land is mostly bare caused by overgrazing. Results from the linear regression showed a positive correlation with soil $CO_2$ emissions with NDVI (p< 0.05), explaining between 35 % and 82 % of the variation in

soil $CO_2$ emissions at the four sites. We therefore observed high $CO_2$ emissions when NDVI was high, which is an indication of more vegetation cover. Several studies have shown that vegetation can affect soil respiration by intercepting radiation, modifying the soil moisture regime, adding litter onto the soil surface, and affecting both plant and root respiration (Myneni et al. 1995; Almagro et al., 2013;). Thus, the inclusion of both NDVI and WC is essential for predicting soil $CO_2$ emission from



savanna soils. Our results confirm the importance of vegetation variability in addition to WC as a key driver of productivity in savannas and is consistent with other studies (Reichstein et al., 2003; Anderson et al., 2008; Lees et al., 2018). Concurrently, the same relationship between NDVI and $N_2O$ emissions could not be proven. Our conclusion was that the low N level played a major impact on overall $N_2O$ emissions in this study.

## 5. Conclusion

Land-use management system plays an important role in soil GHG emissions due to changes in vegetation, soil, hydrology, and nutrient availability. However, the effects on GHG emissions change remain uncertain due to limited data being available in developing countries, especially in dry areas and ecosystems facing diverse land-use change. In our study, we quantified soil GHG emissions from
four dominant LUTs in the dry lowland of southern Kenya, namely bushland, conservation land, cropland, and grazing land. Our results showed significant variation between seasons and the respective land-use types. $CO_2$ emissions in particular were higher during the wet season than the dry season. The lowest seasonal mean $CO_2$ emissions were observed in the SD, when soil moisture was very low while soil temperature was very high. Most of the variation in $CO_2$ emissions was explained
by NDVI and soil moisture, highlighting the fact of including proxies of vegetation cover in soil GHG emission studies in savannas. $N_2O$ emissions and $CH_4$ emissions were of minor importance at all sites. However, we may have missed some incidents of the soil $N_2O$ emissions, as these are often episodic, i.e. after fertilization or precipitation events. These can easily be missed unless a continuous measurement and monitoring programme is in place. Following theseresults, there is still need for
more continuous studies to cover spatial and temporal variations in soil emissions from diverse land-use types across seasonal and management gradients. Furthermore, continuous measurements allow the detection changes in GHG emission patterns following intensification. Nevertheless, our results are useful for deriving reliable climate change mitigation interventions by informing the relevant policies.

## 6. Data availability

The data associated with the manuscript can be obtained from the corresponding author upon request.

**Acknowledgements:** We acknowledge the Schlumberger Foundation under the Faculty for the Future programme for funding. The work was conducted under Environmental sensing of ecosystem services for developing a climate-smart landscape framework to improve food security in East Africa,



funded by the Academy of Finland (318645). A research permit from NACOSTI (P/18/97336/26355)
is acknowledged. Taita Research Station of the University of Helsinki is acknowledged for technical
and fieldwork support and Mazingira Centre of the International Livestock Research Institute for
technical support in the laboratory work. Specifically, we would like to thank Mwadime Mjomba for
his help in collecting field samples and Paul Mutua, George Wanyama, Sheila Okoma, Francis
Njenga, and Margaret Philip for their help in the laboratory work. We acknowledge Taita Hills
Wildlife Sanctuary and LUMO Community Wildlife Sanctuary for providing us with access, and
especially Mr. Richard Obanda and Mr. Donart Mwambela Mwakio, and the team of wardens for
accompanying us when needed. Lutz Merbold and Sonja Leitner acknowledge the support provided
by the CGIAR Fund Council, Australia (ACIAR), Irish Aid, European Union, International Fund for
Agricultural Development (IFAD), the Netherlands, New Zealand, United Kingdom, USAID, and the
Kingdom of Thailand for funding the CGIAR Research Program on Livestock.

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
