# Peer review of "SOIL GREENHOUSE GAS EMISSIONS UNDER DIFFERENT LAND-USE TYPES IN SAVANNA ECOSYSTEMS OF KENYA"

_Biogeosciences, 2019_

## Referee Comment (RC1) · Anonymous Referee #1 · 15 Nov 2019

Overall comments:

The manuscript describes a study in four typical land use types in Kenya, Africa. Soil fluxes of CO2, N2O, and CH4 were measured manually 8 times over the course of a year. The main strength of the manuscript is that it produces flux estimates of these greenhouse gasses in under-represented ecosystems. Correlations with driving factors of moisture, soil C content, and vegetation activity (NDVI) were explored. The main weakness of the manuscript is the sampling campaign and methods are very limited and coarse, and thus interpretation of the driving factors of the fluxes are much more speculative than could be with greater initial and supporting data. My suggestion

would be to reduce the length of the manuscript to focus just on the data collected and acknowledge the weaknesses in the data set. A shorter, more concise, manuscript would be much more effective to get the data out there.

Abstract

Ln 25 – the N2O flux was more than double the cropland than bushland, why do you say is was not different between the four sites?

Ln 31 – Over the course of the measurement period or between sites, CO2 was correlated with soil moisture?

Ln 30-40 - The abstract does not have a clear message. Soil C is important, but soil moisture is driving fluxes, but NDVI is correlated. What is the take home point?

Introduction

The introductory paragraph never says what produces and consumes GHGs from the soil?

ASALs is an acronym that could be avoided by using drylands, or arid ecosystems. Overall there are many acronyms used that could be avoided.

Methods

Ln 187 – ssp

Ln 240 – This is a large assumption. Does the sampling really represent the average flux of the day for your ecosystem? At least one of those references is for a temperate forest where they did measure the 24 hour cycle, which likely has a very different cycle than these ecosystems due to differences and vegetation type and environmental variables.

Ln 252 – The pooling method reduces the sample # to 3 for each LUT time period instead of 9?

Ln 290 – was temperature measured in the chamber?

Results

Ln 385 – What are the errors on the fluxes? They are so small for soil CO2 fluxes. Report error and sample size.

Figures 4 and 5 are good. Keeping the color scheme the same would be helpful.

Figure 6 – put in same color scheme.

Figure 7 – this is at such a large scale, I don't find it very informative. Fig 6 shows the data used.

Discussion

Different terms are being used, soil respiration, soil Co2 emissions, ecosystem soil respiration (?) Make this consistent.

There is quite a bit of speculation in the discussion. It would be better shortened and more focused on the data collected, not the data lacking that could explain the patterns. This is true for CO2 and N2O sections. Interesting CH4 just gets one sentence because it is small. . . but this is important too!

---

## Short Comment (SC1) · 19 Nov 2019

I want so suggest some changes for that publication:

N2O fluxes are sometimes given in $\mu$g N2O-N m-2 h-1 (e.g. l. 295 or 420) and in some graphics they are given in N2O ($\mu$g m-2 h-1) (e.g. figure 4 or figure 5). This is the same for CO2. Units should be used consistent, so that you can compare the fluxes. l. 171: Height a.s.l. is only given for the cropland site and should be given as well for the other sites.

---

## Referee Comment (RC2) · D. Oteino (Referee) · 21 Nov 2019

This is interesting study conducted in semi-arid parts of Kenya, where similar data are quite scarce. The set-up is an area characterized by a series of activities. It is a surprised that there is some form of cultivation/farming in an area that looks more like Tsavo national Park. Nonetheless, the study provides valuable data that extend our knowledge of ecosystem gas fluxes in this part of the world. The study was conducted in a relatively poor soil. What the authors failed to mention, especially for the cropped and grazed sites was the slope of the field. I tend to imagine that erosion must be playing a critical role in mineralization processes in this place. It

looks like the organic/humus, top soil layer is completely gone and what remains is mainly the mineral soils. Unfortunately, the paper is already too long and I won't recommend inclusion of more information on land use history, which would have been helpful in understanding/interpreting these results. It's very surprising that temperature and soil moisture had no influence on soil $CO_2$ fluxes. Could it be the method of data collection, with significant data collection gaps that led to this? For future, the authors need to consider higher frequencies of data collection. In such arid ecosystems, evaporation is quite high and it is likely that critical information is lost by not collecting data more regularly. CH4 seems to contribute little to this paper, why not exclude it completely? I don't see the two lines of discussion on CH4 are of major benefit to the readers. The paper is already too long and probably removing all the descriptions on CH4 could reduce the number of pages. The word "Soil Organic Carbon SOC" is introduced in the introductory part of the Ms. In the methods, there is total soil carbon and in the results, I met Soil Carbon. In the discussions, SOC becomes the main discussions line. The authors need to be consistent in the use of these terms, otherwise the readers get confused. Ln 65. Not all savanna belongs to the ASALs. The humid savannas are relatively wet, with green vegetation almost throughout the year. It is therefore not right to make such a sweeping statement.

Please also note the supplement to this comment:
https://www.biogeosciences-discuss.net/bg-2019-407/bg-2019-407-RC2-supplement.pdf

———————————————————

[Figure]

**Supplement:**

Review report on Wachiye et al. *Soil greenhouse gas emissions under different land-use types in savanna ecosystems of Kenya*

General comments

This is interesting study conducted in semi-arid parts of Kenya, where similar data are quite scarce. The set-up is an area characterized by a series of activities. It is a surprised that there is some form of cultivation/farming in an area that looks more like Tsavo national Park. Nonetheless, the study provides valuable data that extend our knowledge of ecosystem gas fluxes in this part of the world. The study was conducted in a relatively poor soil. What the authors failed to mention, especially for the cropped and grazed sites was the slope of the field. I tend to imagine that erosion must be playing a critical role in mineralization processes in this place. It looks like the organic/humus, top soil layer is completely gone and what remains is mainly the mineral soils. Unfortunately, the paper is already too long and I won't recommend inclusion of more information on land use history, which would have been helpful in understanding/interpreting these results.

It's very surprising that temperature and soil moisture had no influence on soil $CO_2$ fluxes. Could it be the method of data collection, with significant data collection gaps that led to this? For future, the authors need to consider higher frequencies of data collection. In such arid ecosystems, evaporation is quite high and it is likely that critical information is lost by not collecting data more regularly.

$CH_4$ seems to contribute little to this paper, why not exclude it completely? I don't see the two lines of discussion on $CH_4$ are of major benefit to the readers. The paper is already too long and probably removing all the descriptions on $CH_4$ could reduce the number of pages.

The word "Soil Organic Carbon SOC" is introduced in the introductory part of the Ms. In the methods, there is total soil carbon and in the results, I met Soil Carbon. In the discussions, SOC becomes the main discussions line. The authors need to be consistent in the use of these terms, otherwise the readers get confused. Ln 65. Not all savanna belongs to the ASALs. The humid savannas are relatively wet, with green vegetation almost throughout the year. It is therefore not right to make such a sweeping statement.

Specific comments

Ln 67. Note that shrubs are woody vegetation

Ln 88. Revise the sentence. Overstocking leads to grazing pressure. The way the sentence is written is redundant.

Ln 96-7. ---Croplands are still being cleared from natural vegetation----re-write the sentence, it's not making the intended meaning.

Ln 104, what's "cropland farming"?

Ln 153. The authors need to be clear on the physiognomic characterization of the vegetation they are studying. Here you have woodlands, bushlands and on line 155 you have wood bushlands, which is which?

Ln 156 are Lions also grazers?

Ln 160 –other important land use(s)

Ln 173. Is the farm rain-fed or not? Are there other sources of moisture input apart from rain?

Ln 237, how deep was the collar inserted into the soil?

Result

Label Fig. 3 as a and b

Ln 377, Sand proportion was lower than what? In comparative sentences, learn also to use "lowest"or "highest" see ln 417.

Ln. 456 present data/results according to the chronology of the figures and avoid this back and forth.

Ln 481. Delete (in) before during.

**Discussions**

SOC is only mentioned in the introduction but not in the methodology or results, yet it becomes very prominent in the discussions. Be consistent in the use of terms.

Ln 525 is not correct. You cannot attribute the differences only to vegetation. It is definite that land use itself leads to the differences in soil C. Although this is argued correctly in the later sections, this section should be revised.

Ln 548. The argument with clays is a bit far-fetched anyway.

Ln 592. ---temperature was measured "down" to 5 cm. I would imagine that 5 cm depth is almost at the surface. What was the deciding factor for installing temp/moisture sensor at this depth? This depth, being close to the surface is associated with very strong temperature fluctuations. It may be one of the reasons why the authors found no temperature correlation with $CO_2$ efflux. Most grass rots, cereals included, have roots located within 10 cm, and may extend down to 30 cm. the woody vegetation in such dry places have their roots even deeper. Trying to establish relations with variables measured at 5 cm may not yield positive results.

Ln 593 check the sentence. How does root respiration tap moisture?

Ln 640. Consider soil erosion and volatilization also.

Ln 651. Use "dung" instead of faeces.

Ln 665 what's T? from nowhere, you introduce T.

---

## Author Comment (AC2) · 24 Dec 2019

Authors Response: Two reviewers have reviewed the manuscript and an additional comment was made in the open discussion. Before, responding to each reviewer and comment individually, we would like to thank for the constructive comments and informative feedback.

The document is structured as follows: each of the short comments (indicated by SC) is first repeated followed by our response (indicated as AC). Where relevant we either include a rephrased sentence already or explain how we intend to implement suggested changes.

[Figure]

Short comments Cornelius Oertel cornelius.oertel@thuenen.de I want to suggest some changes for that publication: SC: N2O fluxes are sometimes given in $\mu$g N2O-N m-2 h-1 (e.g. l. 295 or 420) and in some graphics, they are given in N2O ($\mu$g m-2 h-1) (e.g. figure 4 or figure 5). This is the same for CO2. Units should be used consistently so that you can compare the fluxes. AC: We thank Mr. Oertel for pointing this out and have made the necessary correction. CO2 (mg m-2 h-1) N2O ($\mu$g m-2 h-1) and CH4 (mg m-2 h-1).

SC: l. 171: Height a.s.l. is only given for the cropland site and should be given as well for the other sites. AC: This will be done in the revised manuscript. Cropland is at 1070 m a.s.l, bushland at 1076 m a.s.l, grazing land at 970 m a.s.l, and conservation land at 928 m a.s.l.

---

## Author Response (AR1)

**SOIL GREENHOUSE GAS EMISSIONS UNDER DIFFERENT LAND-USE TYPES IN SAVANNA ECOSYSTEMS OF KENYA**

**Authors:** *Sheila Wachiye[1,2,5], Lutz Merbold[3], Timo Vesala[2], Janne Rinne[4], Matti Räsänen[2], Sonja Leitner[3], and Petri Pellikka[1, 2]

1) Earth Change Observation Laboratory, Department of Geosciences and Geography, University of Helsinki, Finland
2) Institute for Atmosphere and Earth System Research, University of Helsinki, Finland
3) Mazingira Centre, International Livestock Research Institute (ILRI), Nairobi, Kenya
4) Department of Physical Geography and Ecosystem Science, Lund University, Sweden
5) School of Natural Resources and Environmental Management, University of Kabianga, Kenya

**Correspondence email:* *sheila.wachiye@helsinki.fi*

**Abstract**

Field measurement data on greenhouse gas (GHG) emissions are still scarce for many land-use types in Africa, causing a high uncertainty in GHG budgets. To address this  gap, we present *in situ* measurements of carbon dioxide ($CO_2$), nitrous oxide ($N_2O$), and methane ($CH_4$) emissions in the lowlands of southern Kenya. We conducted eight chamber  measurement campaigns on gas exchange from four dominant land-use types (LUTs) including (1) cropland, (2)  bushland, (3) grazing land, and (4) conservation land between 29 November 2017 to 3 November 2018,  accounting for regional seasonality (including wet and dry seasons and transitions periods). Mean $CO_2$ emissions for the whole observation period were significantly higher (p-value<0.05) in the conservation land (75±6 mg $CO_2$-C $m^{-2}$ $h^{-1}$) compared to the three other sites, which ranged from 45±4 mg $CO_2$-C $m^{-2}$ $h^{-1}$ (bushland) to 50±5 mg $CO_2$-C $m^{-2}$ $h^{-1}$ (grazing land). Furthermore, $CO_2$ emissions varied between seasons, with significantly higher emissions during the wet season than the dry season. Mean $N_2O$ emissions were  highest in cropland (2.7±0.6 µg $N_2O$-N $m^{-2}$ $h^{-1}$) and lowest in bushland (1.2±0.4 µg $N_2O$-N $m^{-2}$ $h^{-1}$ ) but did not vary with season. In fact, $N_2O$

**Commented [WS1]: RC1:** Ln 25 – the N2O flux was more than double the cropland than bushland, why do you say is was not different between the four sites?

**Commented [WS2R1]:** We have corrected

emissions were very low both in the wet and dry seasons, with slightly elevated values during the early days of the wet  seasons in all LUTs. On the other hand, $CH_4$ emissions did not show any significant differences between LUTs and seasons. Most $CH_4$ fluxes were below the limit of detection (LOD, ±0.03 mg $CH_4$-C $m^{-2}$ $h^{-1}$). We attributed the difference in soil $CO_2$ emissions between the four sites to soil C content, which differed between the sites and was highest in the conservation land. In addition, $CO_2$ and $N_2O$ emissions positively correlated with soil moisture, thus an increase in soil moisture led to an increase in emissions. Furthermore, vegetation cover explained the seasonal variation of soil $CO_2$ emissions as depicted by a strong positive correlation between NDVI and $CO_2$ emissions, most likely because with more green (active) vegetation cover higher $CO_2$ emissions occur due to enhanced root respiration compared to drier periods in the year. Soil temperature did not show a clear correlation with either $CO_2$ or $N_2O$ emissions, which is likely due to the low  variability in soil temperature between seasons and sites. Based on our results, soil C, active vegetation cover and soil moisture are key drivers of soil GHG emissions in all the tested LUTs in  South Kenya. Our results are within the range of previous GHG flux measurements from soils from various LUTs in other parts of Kenya and contribute to more accurate baseline GHG emission estimates from Africa, which are key to reduce uncertainties in global GHG budgets as well as for informing policymakers when discussing low-emission development strategies.

**Commented [WS3]:** Rephrased based on reviewer one on What is the take home point?

**KEYWORDS: Carbon Dioxide, Nitrous Oxide, Methane, Bushland, Conservation, Grazing land, Cropland.**

**1. Introduction**

Soil is a major source, and in many cases also a sink, of the atmospheric greenhouse gases (GHG) carbon dioxide ($CO_2$), nitrous oxide ($N_2O$), and methane ($CH_4$) (Oertel et al., 2016). The concentrations of these gases have increased since the onset of industrialization in 1970, leading to global warming (IPCC, 2013). GHGs trap the long-wave radiation emitted by the Earth's surface, thus increasing surface temperatures (Arrhenius, 1896). Soil $CO_2$ emissions originate from root respiration and heterotrophic decomposition of soil organic matter (Oertel et al. 2016). $N_2O$ can be produced from many pathways in the soil nitrogen (N) cycle, but is considered to result primarily

from nitrification and denitrification (Butterbach-Bahl et al. 2013). $CH_4$ is produced by methanogenesis under anaerobic conditions and consumed by methanotrophic microorganisms under aerobic conditions, with the latter being more important in well-aerated upland soils, which consequently show net $CH_4$ uptake (i.e. negative flux) (Serrano-silva et al. 2014; Hanson and Hanson 1996). The production and consumption of soil GHGs largely depends on soil physical and chemical properties (Davidson et al., 2006) (e.g. texture, soil organic matter and pH) and are further driven by environmental factors such as soil moisture and soil temperature (Davidson et al., 2006). Thus, soil GHG emissions and uptake along with their controlling factors differ between biomes based on the land use and land-use management.

Land-use changes are reportedly the largest source of anthropogenic GHG emissions in Africa (Valentini et al., 2014). However, *in situ* studies on GHG emissions from various ecosystems in remain scarce, particularly from savanna ecosystems (Castaldi et al., 2006). Savanna is an important land cover type in Africa, covering more than 40 % of its total area (Scholes et al., 1997). In Kenya, savanna and grassland ecosystems cover about 80 % of the total area, comprising various land-use types (LUTs) (GoK, 2013). These ecosystems are subject to accelerating land-use change (Grace et al., 2006) due to population growth (Meyer and Turner, 1992) and land-use management activities (Valentini et al., 2014). Conversion of savanna for small- and large-scale livestock production, crop cultivation, and human settlement is common in Africa (Bombelli et al., 2009). As a consequence, vegetation cover, net primary productivity, allocation of carbon and nutrients in plants and soil (Burke et al., 1998) as well as soil GHG emissions are affected (Abdalla et al., 2018; Carbone et al., 2008).

Overgrazing due to overstocking is a major cause of soil and vegetation degradation in large parts of African savannas (Patton et al., 2007; Abdalla et al., 2018). Factors associated with grazing include animal feeding preferences of certain plant species, thus creating higher pressure for certain species, which decline in numbers over time, leading to species loss and lower pasture nutritive value (Patton et al. 2007). In addition, soil trampling increases soil bulk density and decreases soil water infiltration (Patton et al., 2007). Furthermore, high rates of dung and urine deposition, especially around homesteads and waterholes, create high N concentrations that are toxic for many savanna grass species, affecting vegetation cover and composition (e.g. increase of encroaching species such as *Solanum incanum* L., which is toxic for livestock (van Vegten 1984)). Given that all these factors affect soil properties, soil GHG emissions are most likely similarly affected (Wilsey et al., 2002).

In addition to overgrazing, rapid human population growth leads to more people migrating into savanna ecosystems, which has led to the expansion of cropland (Pellikka et al., 2018; Patton et al.,

**Commented [WS4]: RC 1:** The introductory paragraph never says what produces and consumes GHGs from the soil?

**Commented [WS5R4]:** We have included in the introductory

**Commented [WS6]:** Rephrased

**Commented [WS7]: RC:** Ln 65. Not all savannah belongs to the ASALs. The humid savannas are relatively wet, with green vegetation almost throughout the year. It is therefore not right to make such a sweeping statement.
**AC:** *We revised to drylands instead of ASALs.*

**Commented [WS8]: RC 2:** Ln 88. Revise the sentence. Overstocking leads to grazing pressure. The way the sentence is written is redundant.

**Commented [WS9R8]:** Done

**Commented [WS10]:** Rephrased based on RC2

2007). Brink and Eva (2009) found that the area under cropland increased by 57 % between 1975 and 2000 in Africa. In the Horn of Africa, cropland areas increased by 28 % between 1990 and 2010 (Brink et al., 2014), while wooded vegetation in East Africa decreased by 5 % in forests, 16 % in woodlands, and 19 % in shrublands (Pfeifer et al., 2013). As an additional example, in our study area Taita Taveta County in Southern Kenya, the area under cropland increased from 30 % in 1987 to 43 % in 2011 (Pellikka et al., 2018). However, in the Taita Hills, located in the County, this trend has slowed down in recent years, while the savanna lowlands are still being cleared to make way for new cropland (Pellikka et al., 2013).

**Commented [WS11]: RC2:** Ln 96-7. ---Croplands are still being cleared from natural vegetation----re-write the sentence, it's not making the intended meaning.
**AC:** *Done. Revised --Natural vegetation is being cleared to make way for the expansion of cropland*

Croplands in the Kenyan savannas are mostly managed by smallholder farmers with small land sizes (Waswa and Mburu, 2006). Due to high poverty levels in this region, inputs to improve crop yields, such as the use of fertilizer and herbicide, and mechanized farming are minor (Waswa and Mburu, 2006; CIDP, 2014). Consequently, increase in productivity are mostly generated via cropland expansion. These smallholder farms are likely to have substantial effects on national GHG emission budgets (Pelster et al., 2017). Until now, only a few studies have investigated soil GHG emissions from such agricultural landscapes (Rosenstock et al., 2016), and these studies were mostly carried out in high-potential farming areas such as the Kenyan highlands, which receive >1000 mm rainfall (FAO, 1996). For example, Rosenstock et al. (2016) showed a large variation of $CO_2$ and $N_2O$ emissions both within and between four crop types as affected by environmental conditions and land management. However, studies measuring GHG emissions from low-productivity croplands in southern Kenya are to the best of our knowledge still missing. Thus, this study focused on soil GHG emissions from different LUTs relevant for the semi-arid region of Southern Kenya.

Given the vast area covered by savanna, land use and land-cover changes are likely to affect global, regional, and national C and N cycles, and hence the quantification of their role is vital (Lal, 2004; Williams et al., 2007). Studies in Kenya have shown large variations of soil GHG emissions in various savanna ecosystems (Otieno et al., 2010; Oduor et al., 2018), due to land-use (Ondier et al., 2019) and management activities (K'Otuto et al. 2013). Due to the high diversity of these savanna ecosystems, such studies may not be entirely representative for every region (Ardö et al., 2008).

The lack of reliable soil GHG flux data from natural savanna and cropland limits our understanding of GHG emissions from African soils (Hickman et al., 2014; Valentini et al., 2014). At the same time, accurate quantification of GHG emissions from multiple LUTs are essential to allow for reliable estimation of Kenya's national GHG inventory (IPCC, 2019). This is particularly important as Kenya currently relies on a Tier-1 approach by using default emission factors (EFs) provided in the

Guidelines for Greenhouse Gas Inventories of the UN Intergovernmental Panel on Climate Change (IPCC) to estimate national GHG emission budgets. Following the Paris Climate Agreement (https://unfccc.int/process-and-meetings/the-paris-agreement/d2hhdC1pcy), most countries across the globe, including Kenya, have not only agreed to accurately report their GHG emissions at national scales following a Tier-2 approach (i.e. using localized data) but also to mitigate anthropogenic GHG emissions in the upcoming decades, as is communicated via their Nationally Determined Contributions (NDCs). Both can only be achieved with locally derived data.

To address the lack of localized GHG emission data from different LUTs in Kenya, our study aims at: (1) providing crucial baseline data on soil GHG emissions from four dominant land uses, namely conservation land, grazing land, bushland, and cropland, and (2) investigating abiotic and biotic drivers of GHG emissions during different seasons. We hypothesized that GHG emissions in cropland would be higher compared to grazing land, bushland, and conservation land because of larger nutrient inputs (i.e. fertilization) in managed land. Further, we hypothesized that GHG emissions would differ between seasons; more precisely, we expected higher GHG emissions in the wet season than in the dry season caused by higher soil moisture.

**2. Materials and Methods**

**2.1. Study Area**

This study was conducted in the lowlands (800–1000 m a.s.l.) of Taita Taveta County (latitude 3° 25´ S and longitude 38° 20´ E) located in southern Kenya (Fig. 1). Taita Taveta County is one of Kenya's dryland areas, with 89 % of the  area characterized as arid and semi-arid area. The county is divided into three major geographical regions, namely the mountainous zone of the Taita Hills (Dawida, Kasigau, Sagalla), Taita lowlands, and the foot slopes of Mt. Kilimanjaro around Taveta.

In the lowlands,  vegetation types include woodlands, bushlands, grasslands, and riverine forests/swamps. Tsavo East and Tsavo West National Parks covers ca. 62 % of the county area  (CIDP, 2014). The parks are open savanna and bushland that support large herbivores, predators and a wealth of birdlife. There are 28 ranches designated for livestock production and two wildlife sanctuaries (Taita Hills Wildlife Sanctuary and LUMO Community Wildlife Sanctuary).

**Commented [WS12]:** *We harmonize this in the revised manuscript to bushlands as found in Tsavo East and West national park*

**Commented [WS13]:** Rephrased

pastoralism and only limited ranching occurs (CIDP, 2014). Other important land use includes croplands with dryland agriculture in small-scale farming operations with low farm inputs (CIDP, 2014), dry thickets and shrublands, and sisal farming (Pellikka et al., 2018). The main soilOther important land uses include cropland under small-scale farming (CIDP, 2014), shrublands, and sisal farming (Pellikka et al., 2018). Soil type is characterized by dark red, very deep, acid sandy clay soil (Ferralsols). Our study sites were located in four of these key land uses in the region, namely including cropland, bushland, wildlife conservation land, and grazing land.

The lowland area has a bimodal rainfall pattern with two rainy seasons – a long rainy period between March and May and a short rainy period between October and December (CIDP, 2014). The hottest and driest months are January and February, while the dry season from June to October is cooler (Pellikka et al., 2018). Mean annual rainfall is 500 mm and the average annual air temperature is 23 °C, with an average daily minimum temperature of 16.7 °C and a maximum temperature of 28.8 °C (CIDP, 2014).

The croplandThe lowland has a bimodal rainfall pattern with two rainy seasons – a long rain season between March and May and a short rain season between October and December (CIDP, 2014). The hot and dry months are January and February while the dry season from June to October is cooler (Pellikka et al., 2018). Mean annual rainfall is 500 mm and the mean annual air temperature is 23 °C, with an average daily minimum and maximum temperature of 16.7 °C and 28.8 °C respectively (CIDP, 2014).

The first site isinvestigated is a cropland located in Maktau (1070 m a.s.l;., Fig. 1). The farm measured approximately1, Fig. 2a) with a size of about one and a half hectares, cultivated with maize (Zea mays L.) intercropped with beans as the main crops.. The farm can be considered as a typical rain-fed smallholder agriculture. Cropfarm and crop growing closely follows the rainy seasons, with maize and beans sowedsowing in March, and bean and maize harvesting occurring in June for beans and August, respectively. Other crops on the farm included cowpeas, pigeon peas, cassava, and sweet potatoes. Land preparation was performed by for maize. Animal ploughing using animal tractionis done to prepare land before seeding, while and weeding was performed is by hand hoehoeing. Small quantities of fresh and dry manure (roughly 20 kg, accounting for less than 1 kg of N) were used every month to improve soil fertility by applying approximately 20 kg of mixed dry and fresh animal manure (less than 1 kg of N) every month during the campaign period (Fig. 2a). .

**Commented [WS14]:** Corrected

**Commented [WS15]: SC:** l. 171: Height a.s.l. is only given for the cropland site and should be given as well for the other sites.
**AC:** *This is done in the revised manuscript for all the sites*

[Figure]

The second site is located in a private bushland in Maktau next to the cropland (1076 m a.s.l., Fig. 1, Fig.  2b). In this region, bushland is found both within the conservation areas and under private ownership. Bushland forms a cover with over 50 % of thorny shrubs and small trees, characterized by *Acacia spp* and *Commiphora* *spp*. The bushes may vary in height ranging from two to five metres. Herbs and savanna grasses (mostly annual or short-lived perennials) less than one metre tall form the ground cover. Private bushland similar to our study site  is used by the farmers to generate small income from forest products such as timber, poles, and firewood, and charcoal to some extent. Additionally, some grazing occurs  primarily by livestock owned by the farmer (CIDP, 2014).

The third site, grazing land (covering approximately 460 km$^2$) is located in the LUMO Community Wildlife Sanctuary (970 m a.s.l, Fig. 1, Fig. 2c) next to Tsavo West National Park and Taita Hills Wildlife Sanctuary. The sanctuary was formed by merging three ranches, namely  Lualenyi and Mramba communal grazing areas and  Oza group ranch  thus the name "LUMO". This sanctuary is communally owned (GoK, 2013) designated for community livestock grazing, where wildlife is also present, as conservation areas are not necessarily fenced. However, overgrazing is a major challenge,

**Commented [WS16]: RC1:** Ln 187 – ssp
**AC:** *Corrected to spp*

[revised manuscript text omitted]
 micrograms respectively. Temperature in Eq. (2) represent air temperature in the chamber headspace measured during each sampling.

$$Mv_{corr} = 0.02241\ \frac{273.15+Temp(℃)}{273.15} \times \frac{Atmospheric\ pressure\ at\ measurement\ (Pa)}{Atmospheric\ pressure\ at\ sea\ level\ (Pa)} \quad (2)$$

The minimum limit of detection (LOD) for each gas was calculated following Parkin et al. (2012)Parkin et al. (2012) and levels were ± 4.9 mg $CO_2$-C $m^{-2}$ $h^{-1}$ for $CO_2$, ± 0.04 µg µg $N_2O$-N m $^{-2}$ $h^{-1}$ for $N_2O$, and ± 0.03 mg $CH_4$-C $m^{-2}$ $h^{-1}$ for $CH_4$. However, we included all data in the analysis, including those below LOD in line with Croghan and Egeghy (2003), who noted that including such data provides an insight on the distinct measurements, thus giving clarifying the set of environmental observations.

Field Code Changed

**2.5. Auxiliary measurements**

During each gas-sampling day, we measured soil moisturewater content (WC) and soil temperature (T) (at a depth of 0–5 cm) adjacent to the collar using a ProCheck handheld data logger with a GS3 sensor (Decagon DevicesProCheck METER Group, Inc., USA). Daily air temperature and precipitation data from November 2017 to November 2018 were obtained from a weather station in Maktau located within the cropland site (Tuure et al., 2019).(Tuure et al., 2019). A soil auger was used to collect soil samples (at a depth of 0–20 cm) during the wet season (22 May 2018) fromin each land-use typesite for soil chemical and physical property analysis. For bulk density, we collected a combination of three samples from each cluster close to each chamber collar at depths of 0–10 cm and 10–20 cm using a soil bulk density ring (Eijkelkamp Agrisearch Equipment, Giesbeek, The Netherlands). Samples were stored in airtight polyethylene bags and kept in a cooler box with ice packs before transportation to the laboratory for further analysis. In the laboratory, samples were stored in a refridgeratorrefrigerator (4 °C) and analysed within 10 days.

The samples were sieved at < 2 mm before analysis. Soil water content was measured by drying soil at 105 °C for 48 h. Soil pH was determined in a 1:2.5 (soil : distilled water) suspension using an electrode pH meter (3540 pH and conductivity Meter, Bibby Scientific Ltd, UK) and soil texture using the hydrometer technique (Scrimgeour, 2008; Reeuwijk, 2002). Total soil C and N content, a duplicate of 20 g of fresh sample was oven-dried at 40 °C for 48 hours and ground into a fine powder using a ball mill (Retsch MM400). Approximately 200 mg of the dry sample were measured by elemental analysis (Vario MAX Cube Analyzer Version 05.03.2013).

**2.6. Statistical Analysis**

Statistical analyses were carried out using R Statistical Software (R 3.5.2, R Core Team). Spearman correlation coefficients were performed among the variables followed by the Kruskal Wallis test to assess significant differences of  GHG emissions between the LUTs and across seasons. A post-hoc analysis involving pairwise comparisons using the Nemenyi test was performed  where significant differences existed. Significance level was set at $p < 0.05$.

We assessed the correlation between soil GHG  emissions with T and WC using several functions based on the coefficient of determination ($R^2$), root-mean-square error (RMSE) and Akaike's information criterion (AIC). There being no difference in the outputs, we present results from the Gaussian function (O'Connell, 1990) for the correlation between soil  emissions and T using Eq. (3), and a quadratic function for correlation with WC using Eq. (4). We also evaluated the combined effect of T and WC on soil GHG emissions by combining Eq. (3) and Eq. (4) into Eq. (5) to assess the effect of these two variables on the emissions.

$$Rs = ae^{(bT+cT^2)} \quad (3)$$

$$Rs = a + bWC + cWC^2 \quad (4)$$

$$Rs = e^{(aT+bT^2)} \times (cWC + dWC^2) \qquad (5)$$

Where *Rs* is soil GHG ($CO_2$ and $N_2O$) emissions, *T* is soil temperature (°C), and *WC* is soil water content ($m^3\,m^{-3}$), while *a*, *b*, *c*, and *d* represent the model coefficients.

After no correlation with T and a weak correlation with WC were observed, we included  vegetation cover. For this, we used Normalized Difference Vegetation Index (NDVI)  from  Moderate Resolution Imaging Spectroradiometer (MODIS) from https://ladsweb.modaps.eosdis.nasa.gov. NDVI quantifies vegetation vigour by measuring the difference between reflectance in near-infrared (which green chlorophyll-rich vegetation strongly reflects) and red wavelength areas (which vegetation absorbs) computed using Eq. (6). MOD13Q1 products from MODIS are NDVI data generated from a 16-day interval at a 250 m spatial resolution as a Level 3 product (Didan, K, 2015).

$$NDVI = \frac{NIR+Red}{NIR+Red} \qquad (6)$$

 To cover our study period, we selected NDVI data  within the  campaign dates. If no data fitted within our dates, we used data that were less than five days before or after the campaign dates, assuming that no significant increase or decrease would occur in the vegetation. The pixels containing the study sites were extracted based on the latitude and longitude of each site. Linear functions were applied to the seasonal datasets of Rs with NDVI to assess the contribution of vegetation on soil emissions using Eq. (7) and a combined effect of WC and NDVI on soil $CO_2$ emissions (Rs) using Eq. ( 8).

$$Rs = a + bNDVI \qquad (7)$$

$$Rs = a + bNDVI + (cWC + dWC^2) \qquad (8)$$

where *Rs* is soil GHG ($CO_2$ and $N_2O$), *T* is soil temperature, and *WC* soil volumetric water content ($m^3$ $m^{-3}$) while *a*, *b*, *c*, and *d* are model coefficients.

**3. Results**

**3.2. Meteorological data**

During the 12--month study period, the long rainy periods were observedrains lasted from early March to the end of May, while short rainy periods were observedrains occurred between early September and OctoberDecember (Fig. 3). The total annual rainfall was 550 mm, which wasis within the average rainfall quantity expected in the area (CIDP, 2014). MeanThe mean annual air temperature was 22.7 °C (min=16.7 °C, max=30.5 °C). January was the hottest month (min=17.4 °C, max=31.9 °C), while June and July (min=14.5± 0.2 °C, max=27± 0.1 °C) were the coolest.

[Figure]

[Figure]

***Figure 3:*** *(a) Daily maximum and minimum air temperature and (b) daily rainfall from lowland in southern Kenya between November 2017 to October 2018 recorded at Maktau weather station. Total annual recorded rainfall was 550 mm. Highlighted grey bars show the days of the sampling campaigns (the season above the grey bars denote SW and LW for the short and long wet season with corresponding onset, mid and end of the wet season, and SD for the short dry season and LD for the long dry season).*

**3.3. Soil characteristics**

Sand content was highest in cropland (77±8%) compared to the conservation land and bushland (ca. 72±1 %) and lowest in grazing land (64.3 ±0.4%) (See Table 2). Grazing land had the highest clay content (31.7±0.5%) while cropland (19±2%) had the lowest. Soil pH ranged between slightly acidic in the grazing land (6.3 ± 0.3), neutral in the bushland (7.2±0.4), and slightly alkaline in the conservation land and cropland (7.5±0.1 and 7.9±0.2 respectively). Carbon content ranged from 0.93 % in the conservation land to 0.60 % in the cropland. Nitrogen content did not vary significantly between sites (mean=0.08±0.01 %).

**Table 1:** Soil characteristics of the topsoil (a depth of 0–20 cm) from the four land-use types investigated in this study. Values are given as mean ± SE.

**Commented [WS17]: RC:** Label Fig. 3 as a and b
**AC:** *Done*

**Commented [WS18]:** Rephrased based on RC2

| Land Use | % N | % C | Bulk Density (g cm$^{-3}$) | pH | Soil Texture % Clay | % Sand | % Silt |
|---|---|---|---|---|---|---|---|
| Bushland | 0.08 (0.03) | 0.77 (0.5) | 1.31 (0.2) | 7.2 (0.4) | 23.7 (0.7) | 71.6 (2.2) | 4.7 (2.3) |
| Conservation land | 0.09 (0.02) | 0.93 (0.7) | 1.27 (0.4) | 7.5 (0.1) | 26.4 (2.2) | 71.6 (0.5) | 2.0 (0.0) |
| Cropland | 0.07 (0.04) | 0.60 (0.2) | 1.26 (0.3) | 7.9 (0.2) | 19.1 (2.4) | 76.9 (8.1) | 4.0 (5.1) |
| Grazing land | 0.08 (0.02) | 0.83 (0.4) | 1.23 (0.5) | 6.3 (0.3) | 31.7 (0.5) | 64.3 (0.4) | 4.4 (0.4) |

**3.4. Soil greenhouse gas emissions**

**3.4.1. Soil carbon dioxide ($CO_2$) emissions**

Mean annual soil $CO_2$ emissions were highest in the conservation land (75±6 mg $CO_2$-C m$^{-2}$ h$^{-1}$). Concurrently, no significant differences occurred  between grazing land (50±5 mg $CO_2$-C m$^{-2}$ h$^{-1}$), cropland (47±3 mg $CO_2$-C m$^{-2}$ h$^{-1}$), and bushland (45±4 mg $CO_2$-C m$^{-2}$ h$^{-1}$). We observed no significant difference in  $CO_2$ emissions between the first three seasons, namely SD , onset-LW  and mid-LW . However, towards the end of the wet season (end-LW) in May, $CO_2$ emissions in the conservation land and grazing land were significantly higher than  cropland and bushland ($p<0.05$). Through LD, onset-SW, and mid-SW, $CO_2$ emissions in the conservation land remained significantly higher , while those from grazing land dropped during LD and were not different from bushland or cropland emissions thereafter.

Generally, $CO_2$ emissions were higher in the wet seasons than in the dry seasons at all sites (Fig. 4c). At the onset of the rainy season in early March, $CO_2$ emissions increased at all sites by over 200% from SD to LW and dropped during LD by approximately 70% in grazing land, bushland, and cropland. In the conservation land, the drop from LW to LD was about 20%. In the bushland, the highest seasonal mean fluxes were reached in mid-LW (98±6 mg $CO_2$-C m$^{-2}$ h$^{-1}$) while in the conservation land (239±11 mg $CO_2$-C m$^{-2}$ h$^{-1}$), grazing land (160±16 mg $CO_2$-C m$^{-2}$ h$^{-1}$), and cropland (84±12 mg $CO_2$-C m$^{-2}$ h$^{-1}$), the highest  were observed during end-LW towards the end May. The lowest seasonal mean $CO_2$ emissions at all sites were observed during the SD campaign (below 20 mg $CO_2$-C m$^{-2}$ h$^{-1}$, Fig 4).

When comparing  the two wet seasons (LW and SW), $CO_2$ emissions were 45 % (bushland), 55 % (conservation land), 56 % (cropland), and 57 % (grazing land) higher

**Commented [WS19]: RC:** Different terms are being used, soil respiration; soil $CO_2$ emissions, ecosystem soil respiration (?) Make this consistent.
**AC:** *We only use the term "soil $CO_2$ emission".*

**Commented [WS20]: SC:** N2O fluxes are sometimes given in µg N2O-N m$^{-2}$ h$^{-1}$ (e.g. l. 295 or 420) and in some graphics they are given in N2O (µg m$^{-2}$ h$^{-1}$) (e.g. figure 4 or figure 5). This is the same for $CO_2$. Units should be used consistent, so that you can compare the fluxes.
**AC:** *We made the necessary correction. $CO_2$ (mg $CO_2$-C m$^{-2}$ h$^{-1}$) $N_2O$ (µg $N_2O$-N m$^{-2}$ h$^{-1}$) and $CH_4$ (mg $CH_4$-C m$^{-2}$ h$^{-1}$).*

**Commented [WS21]: RC:** Ln 365 – What are the errors on the fluxes? They are so small for soil $CO_2$ fluxes. Report error and sample size.
**AC:** *The sample size per season is seven daily average values derived from three flux values per day from each land use type. The error bar presented here is the standard error of the three flux values per day.*

season in LW than during the short wet seasonSW (Fig. 5a). For the two dry seasons, $CO_2$ emissions were also significantly higher in LD than SD across all the sites (in SD all sites recorded emission below 20 mg $CO_2$-C $m^{-2}$ $h^{-1}$). However, duringDuring the LD, $CO_2$ emissionemissions were 29 % (bushland), 38 % (cropland), 40 % (grazing land), and 77 % (conservation) higher than during SD (Fig. 5a). While. As much as $CO_2$ emissions in cropland, bushland, and grazing land had dropped belowto less than 30 mg $CO_2$-C $m^{-2}$ $h^{-1}$ during LD, $CO_2$ emissions were still high (118±6 mg $CO_2$-C m $^{-2}$ $h^{-1}$) in the conservation land, even more than during onset LW (63±9 (118±6 mg $CO_2$-C $m^{-2}$ $h^{-1}$) and mid LW (79±5 mg $CO_2$-C m $^{-2}$ $h^{-1}$the emissions remained high (Fig 4c).

**3.4.2. Soil nitrous oxide ($N_2O$) emissions**

Mean annual $N_2O$ emissions were very low (< 5 µg $N_2O$-N $m^{-2}$ $h^{-1}$) at all four sites (Fig. 4d). Mean $N_2O$ emissions were higher in the croplandCropland (2.657±0.066 µg $N_2O$-N $m^{-2}$ $h^{-1}$) recorded the highest mean $N_2O$ emissions than in the conservation land (1.596±0.044 µg $N_2O$-N $m^{-2}$ $h^{-1}$), grazing land (1.495±0.044 µg $N_2O$-N $m^{-2}$ $h^{-1}$), and bushland (1.162±0.044 µg $N_2O$-N $m^{-2}$ $h^{-1}$). $N_2O$ fluxesemissions did not show a clear temporal pattern as observed for $CO_2$ emissions. Mean seasonal $N_2O$ emissions were very low during both the wet and dry seasons. Within each season, no significant differences in $N_2O$ emissions were observed among the sites. AtHowever, at the onset of the rainy season (onset-LW), there were observable increases in $N_2O$ emissions from all the sites. During this period, mean $N_2O$ emissions at all the sites were ca. 2.6±0.4 µg $N_2O$-N $m^{-2}$ $h^{-1}$). By mid-LW and end-LW periods, $N_2O$ emissions had dropped to nearly zero (< (≤1 µg $N_2O$-N $m^{-2}$ $h^{-1}$) at all sites. However,In June during LD in June, $N_2O$ emissions infrom the cropland were significantly higher than at the other three sites (2.35±0.03 µg $N_2O$-N $m^{-2}$ $h^{-1}$, $p < 0.05$). During this period, the farmer had just harvested his crops.

When comparing the two wet seasons, $N_2O$ emissions did not differ between LW and SW at all sites (Fig. 5b). However, short $N_2O$ emission pulses were observed during both seasons. A notable peak of apprximatelyabout 70 µg $N_2O$-N $m^{-2}$ $h^{-1}$ was observed in the cropland on 7 April 2018, which wasa week after the application of livestock manure a week before the sampling campaign.application. It had also rained the night before the sampling day. At the same site, we also recorded a peak of 55.2 µg $N_2O$-N $m^{-2}$ $h^{-1}$ on 30 September 2018, likely also due to manure application (personal communication from the farmer Mwadime Mjomba). Other notable peaks were 29.9 µg $N_2O$-N $m^{-2}$ $h^{-1}$ (in the bushland on 033 September 2018) and 26.59 6 µg $N_2O$-N $m^{-2}$ $h^{-1}$ (in grazing land on 4 September 2018). These were observed during the SW from chambers with animal

droppings. For the dry seasons, $N_2O$ emissions did not differ between SD and LD in the bushland, conservation land, and grazing land, while $N_2O$ emissions in the cropland were significantly higher during LD than SD (Fig. 5b).

**3.4.3. Soil methane ($CH_4$) emissions**

Throughout the study period, $CH_4$ emissions did not vary significantly among sites and seasons (Fig. 4e and Fig. 5c). The studied sites were mostly $CH_4$ sinks rather than sources, and $CH_4$ fluxes were very low, ranging from -0.03 to 0.9 mg $CH_4$-C $m^{-2}$ $h^{-1}$ (Fig. 4e), often below the limit of detection (0.03 mg $CH_4$-C $m^{-2}$ $h^{-1}$).

[Figure]

(a)
(b)
(c)
(d)
(e)
Soil Moisture (m$^3$ m$^{-3}$)
Soil Temp (°C)
CO$_2$ (mg m$^{-2}$ h$^{-1}$)
N$_2$O (µg m$^{-2}$ h$^{-1}$)
CH$_4$ (mg m$^{-2}$ h$^{-1}$)
End-SW
SD
Onset-LW
Mid-LW
End-LW
LD
Onset-SW
Mid-SW
Seasons
Land Use Types
Bushland
Conservation Land
Cropland
Grazing Land

[Figure]

515 ***Figure 4:*** *Box plots showing differences in seasonal means for (a) soil moisture, (b) soil temperature, and soil emissions of (c)* $CO_2$*, (d)* $N_2O$*, and (e)* $CH_4$ *for each site from November 2017 to October 2018. Season abbreviations on the x-axis denote SW for the short wet season and LW for*

**Commented [WS22]: RC**: Figures 4 and 5 are good. Keeping the color scheme, the same would be helpful.
**AC:** *The two figures provide different information and having them in the same color scheme might be misleading. Figure 4 shows the difference in emissions between the land uses types while figure 5 gives the differences in emission between the wet and dry season.*

*the long wet season with corresponding onset, mid and end of the wet season, along with SD for the short dry season and LD for the long dry season.*

520

[Figure]

[Figure]

***Figure 5:*** *Seasonal differences in mean (a) $CO_2$, (b) $N_2O$, and (c) $CH_4$ emissions between the long and short wet seasons and the long and short dry season for the four land-use types.*

**3.5. Effects of soil temperature, soil water content, and vegetation indices on soil GHG emissions**

Soil water content (WC) was highest during the wet season (mean 0. 19±0. 06 m$^3$ m$^{-3}$) and lowest during the dry season (mean. 0.07±0.02 m$^3$ m$^{-3}$) at all sites (see Fig. 4a). Soil temperature (T),  were highest during the SD (36.7±2.1 °C)  and lowest (24.5±0.6 °C) in LD  (Fig.4b). Throughout all the campaigns, mean WC and mean T were highest in the conservation land, followed by grazing land, bushland, and  lowest in the cropland.

 Regression results on soil $CO_2$ and soil $N_2O$ emissions against T and WC are shown in Table 2. The results showed positive correlations between soil $CO_2$ emissions and WC ($p < 0.05$). However, the $R^2$ was very weak at all sites. Conversely, $CO_2$ emissions showed no correlation with T ($P < 0.05$). We observed no correlation between $N_2O$ and $CH_4$ emissions with either WC or T ($p < 0.05$). Separating data into the wet and dry season did not improve  the correlations.

**Table 2:** Soil water content (WC) and soil temperature (T) control on carbon dioxide ($CO_2$) and nitrous oxide ($N_2O$) denoted by *Rs*, while *a*, *b*, and *c* represent the model coefficient

| **Predictors** | **Land Use** | **$CO_2$-C mg $m^{-2}$ $h^{-1}$** | | **$N_2O$-N ug $m^{-2}$ $h^{-1}$** | |
|---|---|---|---|---|---|
| Soil water content (WC) | | $Rs = a + bWC + cWC^2$ | | | |
| | Bushland | $6.12WC + 0.92WC^2$ | $R^2$= 0.26*** | $19.02WC - 64.11WC^2$ | $R^2$= 0.008 |
| | Conservation land | $135.27WC - 0.57WC^2$ | $R^2$= 0.07** | $11.63WC - 7.736WC^2$ | $R^2$= 0.009 |
| | Cropland | $17.83WC + 0.67WC^2$ | $R^2$= 0.04*** | $28.48WC - 66.63WC^2$ | $R^2$= 0.005 |
| | Grazing land | $15.03WC + 0.79WC^2$ | $R^2$= 0.11*** | $19.81WC - 53.56WC^2$ | $R^2$= 0.002 |
| Soil Temperature (T) | | $R = ae^{(bT+cT^2)}$ | | | |
| | Bushland | $1.078e^{0.26T-0.004T^2}$ | $R^2$= 0.008 | $360.25e^{-0.29T-0.004T^2}$ | $R^2$= 0.008 |
| | Conservation land | $0.001e^{0.81T-0.014T^2}$ | $R^2$= 0.015** | $0.007e^{0.45T-0.008T^2}$ | $R^2$= 0.015 |
| | Cropland | $4.568e^{-0.13T+0.002T^2}$ | $R^2$= 0.008* | $0.007e^{-0.05T+2.42T^2}$ | $R^2$= 0.008* |
| | Grazing land | $4.136e^{0.18T-0.003T^2}$ | $R^2$= 0.015 | $2.366e^{0.05T-0.001T^2}$ | $R^2$= 0.015 |

***: p<0.0001, **: p<0.001, *: p<0.05

Results from combined WC and T on soil $CO_2$ and $N_2O$ emissions did not improve the correlation, as shown in Table 3.  Thus, we included vegetation indices in our model.

**Table 3:** Combined effects of soil water content (WC) and soil temperature (T) control on soil $CO_2$ and $N_2O$ emissions. Soil $CO_2$ and $N_2O$ emissions denoted by *Rs*, while *a*, *b*, *d*, and *e* represent the model coefficient, ($R^2$) the coefficient of determination, and AIC the Akaike's information criterion

| Functions | Land use | a | b | d | e | $R^2$ | AIC |
|---|---|---|---|---|---|---|---|
| **$CO_2$-C mg m$^{-2}$ h$^{-1}$** | | | | | | | |
| $Rs = e^{(aT+bT^2)} \times (dWC + eWC^2)$ | Bushland | -0.12 | 0.001 | 52.774 | -0.527 | 0.31*** | 1888 |
| | Conservation land | 0.90 | -0.016 | 0.0001 | 0.000 | 0.10*** | 2156 |
| | Cropland | -0.39 | 0.006 | 3701.901 | -84.001 | 0.08** | 1886 |
| | Grazing land | 0.14 | -0.003 | 0.842 | -0.008 | 0.12*** | 2024 |
| **$N_2O$-N ug m$^{-2}$ h$^{-1}$** | Bushland | -0.50 | 0.007 | 2008.345 | -58.559 | 0.009 | 785 |
| $Rs = e^{(aT+bT^2)} \times (dWC + eWC^2)$ | Conservation land | 0.56 | -0.010 | 0.000 | 0.000 | 0.003 | 811 |
| | Cropland | 0.67 | -0.017 | 0.003 | -0.0001 | 0.089 | 911 |
| | Grazing land | 0.11 | -0.003 | 0.187 | -0.005 | 0.003 | 770 |

***: p<0.0001, **: p<0.001, *: p<0.05

The annual change in vegetation cover at each site are shown in Fig. (6). The highest NDVI values were observed during the LW in April (ranging from 0.58 to 0.76) and the lowest during the SD (< 0.26). Vegetation greenness increased rapidly from mid-March at all sites coinciding with the onset of the rainy season and remained high (Fig.6). At end of the rainy season, NDVI gradually dropped. Highest NDVI values occurred in the conservation land (0.51±0.05); followed by bushland (0.44±0.05), cropland (0.41±0.05), and the lowest  were recorded in the grazing land (0.33±0.05).

[Figure]

[Figure]

**Commented [WS23]: RC2:** Ln 481. Delete (in) before during.
**AC:** *Done*

**Commented [WS24]: RC1:** Figure 6 – put in same color scheme.
**AC:** *This is a relevant point and we tried to have them in the same color scheme of figure 4. However, this then made it very difficult to differentiate between the LUC types***.**

***Figure 6:*** *Monthly NDVI time series showing the annual trend in vegetation cover from November 2017 to November 2018 for the four land-use types.*

[Figure]

***Figure 7:*** *Normalized difference vegetation index (NDVI) maps for each campaign (SW and LW denotes short and long wet seasons with corresponding onset, mid and end of the season, and SD and LD for short and long dry season) from November 2017 to November 2018. LUMO is the LUMO community wildlife sanctuary, TWNP stands for the Tsavo west National Park and THWS for Taita Taita hills wildlife sanctuary*

OurRegression analysis results show that NDVI was capable of explaining a large degree in the variation of shows a positive correlation between NDVI and seasonal soil $CO_2$ emissions inat all the bushland, grazing land, and croplandsites (see Fig. 8). However, 7). Combined WC and NDVI improved the correlation in the conservation land was noteven further as shown in Table 4. No significant. We also correlation was observed no significant correlation withbetween $N_2O$ at all the sites.emissions and NDVI (Fig. 7).

[Figure]

[Figure]

[revised manuscript text omitted]
 that contribute to soil $CO_2$ emissions (Lal, 2009) also showed the same trend (conservation land > grazing land > bushland > cropland). Therefore, the difference in land use and land-use management activities between our sites played a vital role in modifying both biotic and abiotic factors that drive both soil C content and soil $CO_2$ emissions (Pinto et al., 2002).

Due to the difference in land use and management, vegetation type and cover differed between our sites. The dense grass network in the conservation land formed an almost closed ground cover, especially in the wet seasons (further confirmed by NDVI values). Being a private sanctuary, only wild mammals (no livestock allowed) grazed and browsed there, and thus we observed less damage on the grass cover throughout all the campaigns as compared to the grazing land (which had large patches of bare soil due to overgrazing) and bushland. This provides a good explanation for the difference in mean $CO_2$ emissions between these three LUTs, as vegetation is known to affect soil C concentration and root and microbial respiration that directly contribute to soil $CO_2$ emissions (Fanin et al., 2011; Rey et al., 2011).

With the lowest $CO_2$ emissions being measured in the cropland, we attribute this observation to the continued tillage and removal of crops and crop residues during land preparation, weeding and harvesting, which affects both root respiration and soil C content (Raich et al., 2000; Nandwa, 2001). In East Africa and especially in smallholder farming systems, most of the crop residues are used as livestock feed and fuel. In addition, manure inputs in cropland are very low (about 20 kg per month on a 1.5 ha farm) and thus no measurable difference in $CO_2$ emissions was detected before and after manure input, and with the other LUTs. Several other studies observed the same scenario from low

**Commented [WS25]:** Ln 548. The argument with clays is a bit far-fetched anyway.
**AC:** *We removed this argument in the revised manuscript.*

**Commented [WS26]:** Rephrased

manure input in maize and sorghum plots (Rosenstock et al., 2016; Mapanda et al., 2011, and Pelster et al. 2017).

**Commented [WS27]:** Rephrased

On average, $CO_2$ emissions were higher during the wet season than during the dry season. At the start of both rainy seasons (SW, LW), $CO_2$ emissions increased significantly in all LUTs. Emissions from the conservation land and grazing land are comparable to those in Brümmer et al. (2008), who observed $CO_2$ emissions ranging between 100 and 250 mg $CO_2$-C $m^{-2}$ $h^{-1}$ in a natural savanna in Burkina Faso. Several other studies from similar ecosystem have also documented comparable changes in $CO_2$ emissions with the onset of the rainy seasons (Castaldi et al., 2006; Livesley et al., 2011; Pinto et al., 2002). In the cropland, results in the wet season are similar to those measured by Rosenstock et al. (2016), ranging from 50 to > 200 mg $m^{-2}$ $h^{-1}$. We attributed the increase in $CO_2$ emissions in the wet season to the response of soil microbes and vegetation to soil moisture (Livesley et al., 2011; Otieno et al., 2010). Soil moisture connects microorganisms with soluble substrates (Moyano et al., 2013) and increases microbial activity (Davidson et al., 2006; 2009; Grover et al., 2012) and thereby soil $CO_2$ emissions.

Furthermore, an increase in soil $CO_2$ emissions during the wet season can also be a result of increased root respiration due to more active plant and root growth (Macdonald et al., 2006). Grasses sprout more rapidly than trees and shrubs with the first rains (Merbold et al., 2009). This provides a possible explanation for the higher $CO_2$ emissions in the grassy conservation land, grazing land, and bushland compared to cropland during the rainy season. However, grazing land recorded higher $CO_2$ emissions than bushland (only the farmer's livestock grazed here). The main difference between these two sites – apart from grazing intensity – was that bushland had more trees (*Acacia spp.*) and shrubs (*Commiphora spp.*) and less herbaceous undergrowth than the grazing land, thus providing shade that might have interfered with growth and regrowth of plants below the canopy. Therefore, grass root production in the open conservation land and grazing land was likely higher than in the bushland (Janssens et al., 2001), although we cannot confirm this because root biomass was not determined in this study. In cropland, all grasses and weeds were cleared during regular weeding and therefore did not play a role in root respiration.

To our surprise, the highest mean seasonal $CO_2$ emissions in conservation land, grazing land, and cropland were observed at the end rather than at the peak of the wet season. During this time, both soil moisture and soil temperature had dropped in all LUTs. However, our data was only recorded up to a depth of 5 cm, but roots of perennial grasses, shrubs and trees can tap moisture from greater soil depths (Carbone et al., 2011). According to Carbone et al. (2011), while microbial activity is highest

**Commented [WS28]: RC2**: Ln 593 check the sentence. How does root respiration tap moisture?
**AC**: deleted and rephrased

and most variable in the upper soil layers, which are first to wet-up and dry-down, roots can access water reserves in deeper soil layers that take longer to be exhausted, and therefore remain active at the end of the wet and into the dry season.

**4.2. Soil $N_2O$ emissions**

Our results showed very low $N_2O$ emissions from all LUTs, which we attributed to low soil N content observed in all the sites (see Table 1). Savanna ecosystems are characterized by very tight N cycling, which transcends to low N availability (Pinto et al., 2002 and Grover et al., 2012), and most of this N is rapidly taken up by vegetation, leaving very little for denitrification (Castaldi et al., 2006; Mapanda et al., 2011). The $N_2O$ flux results observed from conservation land, grazing land and bushland are consistent with those observed in a Brazilian savanna by Wilcke et al. (2005), and other studies from similar ecosystems reported comparable $N_2O$ flux magnitudes (Scholes et al., 1997; Castaldi et al., 2016; Mapanda et al., 2010). The higher $N_2O$ emissions observed in June and July from our cropland site after the maize and bean harvests likely occurred due to the disturbance and following absence of live plants, which led to higher soil N availability because of less N uptake by plants and increased root decomposition.

**Commented [WS29]:** Rephrased

In contrast to the patterns observed for $CO_2$ emissions, we did not detect any seasonal variations in $N_2O$ emissions. The only exception to the otherwise very low $N_2O$ emissions was after the onset of the rainy season, when $N_2O$ emissions slightly increased at all sites. Such patterns have previously been shown by several similar studies (Scholes et al., 1997; Pinto et al., 2002; Castaldi et al., 2006; Livesley et al., 2011). The increase in $N_2O$ flux at the onset of the rains has been attributed to an increase in microbial activity and therefore faster decomposition of litter and plant residue facilitated by an increase in soil moisture, thus increasing N availability (Rees et al., 2006). Furthermore, according to Davidson et al., (2000) and Butterbach-Bahl et al., (2013), soil moisture affects soil gas diffusion, oxygen ($O_2$) availability, and the movement of substrate necessary for microbial growth and metabolism.

Negative $N_2O$ emissions were detected during the dry season. Such observations could result from the low N contents observed at all sites coupled with low soil moisture in the dry season, which facilitates diffusion of atmospheric $N_2O$ into the soil. Soil denitrifiers may, therefore, use $N_2O$ as an N substrate in the absence of $NO_2^-$ and $NO_3^-$ (Rosenkranz et al., 2006). Negative $N_2O$ emissions have also been reported in other tropical savanna soils under similarly dry conditions (Castaldi et al., 2006; Livesley et al., 2011).

Manure application in the cropland was very low (< 12 kg of N in 1.5ha for the crop-growing season), and thus $N_2O$ emissions from cropland were low and not different from the other LUTs, which was in contrast to what we had hypothesized. Due to low soil N levels in the cropland, the low amount of manure added was not sufficient to stimulate $N_2O$ emissions, likely because soil N availability was still limiting for plant and microbial growth (Castaldi et al., 2006). Traditional farming systems in smallholder farms in Africa involve repeated cropping with no or very low N inputs that leads to soil N mining over time (Chianu et al., 2012). In line with this, in our cropland site maize and beans are grown during every wet season with no fallow in between years. In addition, the farmer did not use any chemical fertilizer to increase soil N, and the N input from biological N fixation into the soil was likely small because beans were harvested for consumption and bean plant residues were used as livestock feed and not incorporated into the soil. Therefore, the small quantities of manure applied and legume N fixation may have likely been insufficient to compensate for N loss through leaching and crop harvests. According to the Taita Development plan, this is a common scenario in the county, which translates to very low crop yields in this region (CIDP, 2014). Another possible explanation for not detecting the influence of manure on $N_2O$ emissions could be the fact that we did not manage to sample immediately after manure application and therefore might have missed the instant impact of manure application on $N_2O$ emissions. However, similar studies by Pelster et al. (2017) and Rosenstock et al. (2016) also did not see any influence of manure application on soil $N_2O$ emissions and reported $N_2O$ emission values that were generally < 10 µg $N_2O$-N $m^{-2}$ $h^{-1}$). Equally, the deposition of dung and urine by animals in the grazing land and bushland did not have any measurable influence on soil $N_2O$ emissions.

**Commented [WS30]: RC2:** Ln 651. Use "dung" instead of faeces.
AC: Corrected

**4.3.Soil $CH_4$ emissions**

Methane emissions did not vary between the land-use types or with seasons. Most values were below the LOD at all the sites. Soil water content in our study is clearly the limiting factor for methanogenesis, which needs anoxic conditions for a certain period until methanogenic archaea are established (Serrano-silva et al. 2014). Furthermore, soil compaction by animal trampling may have limited $CH_4$ diffusion into the soil thus limiting $CH_4$ consumption by oxidation (Ball et al. 1997). In cropland, continuous tillage interferes with soil structure thus affecting the microenvironment that favours methanotrophic (Jacinthe et al. 2014). Additionally, low soil C as observed in all the sites generally leads to low abundance of soil microorganisms and consequently also methane oxidisers (Serrano-silva et al. 2014). Nevertheless, soils around lakes, waterholes and rivers can be $CH_4$ sources in semi-arid savanna ecosystems, but those were not investigated during this study.

**4.2.4.4. Effects of soil moisture, soil temperature, and vegetation indices on GHG emissions**

Soil moisture and soil temperature are known to be important drivers of soil $CO_2$ production, and they may change across seasons. Seasonal variations in T were very minor and thus changes in WC were considered to be the main driver of $CO_2$ emissions in our study, as previously highlighted by Grover et al. (2012). Brümmer et al. (2009) and Livesley et al. (2011) also found that WC controlled $CO_2$ fluxes from savanna soils, rather than T. Soil moisture determines the rate of soil respiration, including heterotrophic and autotrophic processes that are highly moisture reliant ( Ardö et al., 2008; Grover et al., 2012).

Soil temperature and soil moisture are also vital for nitrification and denitrification, as they control the activities of soil microbes and $O_2$. However, we did not observe a significant relationship between $N_2O$ emissions with both WC and T. Only in the cropland area did we observe a positive correlation with T ($p < 0.05$). As much as previous results have shown a positive relationship between T and $N_2O$ emissions (Castaldi et al., 2010), we did not observe such a relationship in our study. Other studies carried out in savannas by Scholes et al., (1997) and Brümmer et al. (2008) were also unable to link $N_2O$ emissions to variations in T. In fact, $N_2O$ emissions were very low during both the wet and dry seasons, as in ( Castaldi et al., 2004), Pelster et al. (2017), and Rosenstock et al. (2016**)**. The most likely clarification for the lack of seasonality would be the low N levels observed at all the sites (Grover et al., 2012).

The vegetation and its status are other important drivers of GHG emissions from soils. NDVI measures the status of vegetation (with a value range of -1 to 1). High NDVI values correspond to high vegetation cover, while low NDVI corresponds to less or no vegetation ( Gamon et al., 1995; Butt et al., 2011). Therefore, the drop in NDVI values at the end of the rainy season was a result of the reduction in vegetation cover in the land-use types. In the cropland area, this coincided with the harvesting of beans and the drying of the maize plant occurring in June and July. The conservation land showed the highest mean NDVI, mainly due to the dense grassy vegetation, especially during the rainy season. Lowest NDVI values were observed in the grazing land, which was expected because the land is mostly bare caused by overgrazing. Results from the linear regression showed a positive correlation with soil $CO_2$ emissions with NDVI ($p < 0.05$), explaining between 35 % and 82 % of the variation in soil $CO_2$ emissions at the four sites. We therefore observed high $CO_2$ emissions when NDVI was high, which is an indication of more vegetation cover. Several studies have shown that vegetation can affect soil respiration by intercepting radiation, modifying the soil moisture

regime, adding litter onto the soil surface, and affecting both plant and root respiration (Myneni et al. 1995; Almagro et al., 2013;). Thus, the inclusion of both NDVI and WC is essential for predicting soil $CO_2$ emission from savanna soils. Our results confirm the importance of vegetation variability in addition to WC as a key driver of productivity in savannas and is consistent with other studies (Reichstein et al., 2003; Anderson et al., 2008; Lees et al., 2018). Concurrently, the same relationship between NDVI and $N_2O$ emissions could not be proven. Our conclusion was that the low N level played a major impact on overall $N_2O$ emissions in this study.

As is common for sub-tropical regions, seasonal variation in soil temperature was small in the study region and therefore soil temperature did not play a big role in modifying soil GHG emissions. Instead, changes in soil moisture were considered to be the main driver of $CO_2$ emissions in our study, as has previously been highlighted also by other studies (Grover et al. (2012), Brümmer et al. (2009) and Livesley et al. (2011)). However, we did not observe any significant relationship between $N_2O$ emissions with either soil moisture or temperature apart from in the cropland, where we found a positive correlation between $N_2O$ and soil temperature ($p < 0.05$). As much as previous results have sometimes shown a positive relationship between temperature and $N_2O$ emissions (Castaldi et al., 2010), our results are in line with others (Scholes et al., (1997), Brümmer et al. (2008) who were also unable to link soil $N_2O$ emissions to variations in soil temperature. In fact, $N_2O$ emissions were very low during both the wet and dry seasons, which is similar to the findings of Castaldi et al., (2004). The most likely explanation for the lack of seasonality effects on $N_2O$ emissions would be the low soil N levels observed at all the sites, which was probably the most limiting factor for $N_2O$ emissions and thus overruled all other potential controlling factors (Grover et al., 2012).

The vegetation cover as depicted by NDVI represents the status of the vegetation (value range from -1 to 1). High NDVI values correspond to high vegetation cover, while low NDVI correspond to little or no vegetation (Gamon et al., 1995; Butt et al., 2011). Therefore, the increase in NDVI that we observed at the onset of the rainy season indicates sprouting and regrowth of vegetation at that time, while the drop in NDVI values at the end of the rainy season indicates reduction in vegetation cover due to plant senescence and grazing. In the cropland area, low NDVI coincided with the harvesting of beans and the drying of the maize plants in June and July. Highest mean NDVI values were observed in the conservation land, mainly due to the dense grassy vegetation, while the lowest NDVI values were found in the grazing land, which we had expected because this area has large spots without vegetation due to overgrazing. Results from linear regression analysis showed a strong positive correlation of soil $CO_2$ emissions with NDVI ($p < 0.05$), explaining between 35 % and 82 %

of the variation in soil $CO_2$ emissions at the four sites. This means that $CO_2$ emissions were highest when NDVI (i.e. vegetation cover) was high Thus, the inclusion of both NDVI and soil moisture measurements is essential for reliably predicting soil $CO_2$ emission from savanna soils, which is consistent with other studies (Reichstein et al., 2003; Anderson et al., 2008; Lees et al., 2018). Concurrently, the same relationship between NDVI and $N_2O$ emissions could not be proven in our study.

**5. Conclusion**

The  magnitude and temporal and spatial variability of soil GHG emissions in most developing countries have large uncertainties due to a lack of data, especially in dry areas and ecosystems facing  land-use change. In our study, we quantified soil GHG emissions from four dominant LUTs in the dry lowlands of southern Kenya, namely bushland, conservation land, cropland, and grazing land. Our results showed significant variation between seasons and the respective LUTs. $CO_2$ emissions, in particular, were higher during the wet season , when soil moisture was  high, compared to the dry season. Most of the variation in $CO_2$ emissions could be explained by soil moisture and NDVI , highlighting the importance of including proxies for vegetation cover in soil GHG emissions studies in savannas. $N_2O$ emissions and $CH_4$ emissions were of minor importance at all sites. However, we acknowledge that we might have missed some episodes of elevated soil $N_2O$ emissions, as these are often episodic and of short duration, for examples after fertilization or precipitation events.  Following these results, there is still need for more continuous studies to cover spatial and temporal variation in soil GHG emissions, as well as the inclusion of other LUTs than the ones examined in this study (e.g. wetlands). Nevertheless, we believe that our results are important to reduce uncertainties in GHG emission baselines and to identify reliable and meaningful climate change mitigation interventions by informing the relevant policies.

**6. Data availability**

The data associated with the manuscript can be obtained from https://figshare.com/articles/Final_data_for_Soil_Greenhouse_Gas_Emissions_under_Different_Land-Use_Types_in_Savanna_Ecosystems_of_Kenya_/11673579

**Acknowledgments:** We acknowledge the Schlumberger Foundation under the Faculty for the Future programme for funding. The work was conducted under the Environmental sensing of ecosystem services for developing a climate-smart landscape framework to improve food security in East Africa, funded by the Academy of Finland (318645). A research permit from NACOSTI (P/18/97336/26355) is acknowledged. Taita Research Station of the University of Helsinki is acknowledged for technical and fieldwork support and Mazingira Centre of the International Livestock Research Institute for technical support in the laboratory work. Specifically, we would like to thank Mwadime Mjomba for his help in collecting field samples and Paul Mutua, George Wanyama, Sheila Okoma, Francis Njenga, and Margaret Philip for their help with the laboratory work. We acknowledge Taita Hills Wildlife Sanctuary and LUMO Community Wildlife Sanctuary for providing us with access, and especially Mr. Richard Obanda and Mr. Donart Mwambela Mwakio, and the team of wardens for accompanying us when needed. Lutz Merbold and Sonja Leitner acknowledge the support provided by the CGIAR Fund Council, Australia (ACIAR), Irish Aid, European Union, and International Fund for Agricultural Development (IFAD), the Netherlands, New Zealand, United Kingdom, USAID, and the Kingdom of Thailand for funding the CGIAR Research Program on Livestock. Janne Rinne and Lutz Merbold were further supported by the European Commission through the project ‘Supporting EU-African Cooperation on Research Infrastructures for Food Security and Greenhouse Gas Observations’ (SEACRIFOG; project ID 730995).

[revised manuscript text omitted]

**REVIEWERS COMMENTS**

**Soil Greenhouse Gas Emissions under Different Land-Use Types in Savannah Ecosystems of Kenya**

**Authors Response:**

The manuscript has been reviewed by 2 reviewers and an additional comment was made in the open discussion. Before, responding to each reviewer and comment individually, we would like to thank for the constructive comments and informative feedback.

The document is structured as follows: each of the reviewer's comment (indicated by RC) is first repeated followed by our response (indicated as AC and in italic). Where relevant we either include a rephrased sentence already or explain on how we intent to implement suggested changes.

**Anonymous Referee #1**

**Overall comments:**

**RC:** The manuscript describes a study in four typical land use types in Kenya, Africa. Soil fluxes of $CO_2$, $N_2O$, and $CH_4$ were measured manually eight times over the course of a year. The main strength of the manuscript is that it produces flux estimates of these greenhouse gasses in under-represented ecosystems. Correlations with driving factors of moisture, soil C content, and vegetation activity (NDVI) were explored. The main weakness of the manuscript is the sampling campaign and methods are very limited and coarse, and thus interpretation of the driving factors of the fluxes are much more speculative than could be with greater initial and supporting data. My suggestion would be to reduce the length of the manuscript to focus just on the data collected and acknowledge the weaknesses in the data set. A shorter, more concise, manuscript would be much more effective to get the data out there.

**AC:** *Once again, we thank the reviewer for pointing out both, the strengths and the limitations in the originally submitted manuscript. Following the concern made on the length of the paper, we will reduce the revised manuscript, focus only on the important issues, and not compromise the quality of the manuscript.*

**Abstract**

**RC:** Ln 25 – the N2O flux was more than double the cropland than bushland, why do you say is was not different between the four sites?

**AC:** *We thank the reviewer for pointing this out. Actually the difference in annual mean $N_2O$ emissions were significantly higher in the cropland (2.7±0.6 $\mu g\ m^{-2}\ h^{-1}$) than in the conservation land (1.6±0.4 $\mu g\ m^{-2}\ h^{-1}$), grazing land (1.5±0.4 $\mu g\ m^{-2}\ h^{-1}$), and bushland (1.2±0.4 $\mu g\ m^{-2}\ h^{-1}$)(Kruskal-Willis rank test). In contrast, no difference was observed between the other three sites.*

**RC:** Ln 31 – Over the course of the measurement period or between sites, $CO_2$ was correlated with soil moisture?

**AC:** *In all our study sites and across the seasons, soil $CO_2$ emissions were positively correlated with soil moisture.*

**RC:** Ln 30-40 - The abstract does not have a clear message. Soil C is important, but soil moisture is driving fluxes, but NDVI is correlated. What is the take home point?

**AC:** *We will rephrase the abstract as follows in order to provide a clearer message.*

*"Based on our results, soil C and soil moisture are key drivers of soil GHG emissions in all land use types. In addition, vegetation cover explained the seasonal variation of soil $CO_2$ emissions as depicted by the strong positive correlation of between NDVI and $CO_2$ emissions. We conclude that with more green (active) vegetation cover higher $CO_2$ emissions occur due to enhanced root respiration compared to drier periods in the year.*

**Introduction**

**RC:** The introductory paragraph never says what produces and consumes GHGs from the soil?

**AC:** *We will include the following information into the introduction of the revised manuscript. Each of the GHGs observed is either consumed or produced via biogeochemical processes. For instance, methane is produced by methanogenesis process under anaerobic conditions and consumed by methanotrophic microorganisms under aerobic conditions (Serrano-Silva et al., 2014). $CO_2$, is in our case (dark chambers) produced in the soil during decomposition of organic matter and root respiration which produces $CO_2$. Similarly, $CO_2$ is produced via plant respiration in the case plants were present in the chamber (Oertel et al. 2016). $N_2O$ on the other hand is produced as an intermediate product of denitrification and nitrification processes amongst others (Butterbach-Bahl et al., 2013).*

**RC:** ASALs is an acronym that could be avoided by using drylands, or arid ecosystems. Overall there are many acronyms used that could be avoided.

**AC:** *We thank the reviewer for this suggestion. Overall, we will have a look at the use of abbreviations. In the case of ASAL, we will use drylands in the revised manuscript. A similar point was made by reviewer 2.*

**Methods**

**RC:** Ln 187 – ssp

**AC:** *Corrected to spp*

**RC**: Ln 240 – This is a large assumption. Does the sampling really represent the average flux of the day for your ecosystem? At least one of those references is for a temperate forest where they did measure the 24-hour cycle, which likely has a very different cycle than these ecosystems due to differences and vegetation type and environmental variables.

**AC:** *We agree that this is partially an assumption, and yet one has to compromise, as a more frequent sampling was impossible. Overall, the theory of sampling in the respective morning hours is sound (Parkin and Venterea, 2010), as temperatures increase during the day and thus microbial processes similarly are enhanced. Of course, the fact that moisture and other factors are essential drivers following our results this may be misleading and yet again the drivers suggested are for the whole dataset, i.e. across seasons and not necessarily aiming at explaining diurnal variations. We have done diurnal GHG measurements in other regions in Kenya with a portable laser absorption spectrometer where we found clear temperature dependencies during the course of the day (data not shown here, as this is part of another study, Butterbach-Bahl in prep.).*

**RC:** Ln 252 – The pooling method reduces the sample # to 3 for each LUT time period instead of 9?

**AC:** *This is correct, the three gas flux estimates that we have at the end of the day for each LUC type is thanks to the pooling method derived from 9 distinct locations (chambers). With this approach, one can still account for spatial heterogeneity while reducing the overall number of GHG samples to be analysed in the laboratory* (Arias-Navarro et al., 2013)*.*

**RC:** Ln 290 – was temperature measured in the chamber?

**AC:** *We recorded the air temperature in the headspace of each chamber at the same interval as gas pooling (recorded at T1, T2, T3 and T4). This temperature is then used to correct the gas flux during flux calculation.*

**Results**

**RC:** Ln 385 – What are the errors on the fluxes? They are so small for soil $CO_2$ fluxes. Report error and sample size.

**AC:** *The sample size per season is seven daily average values derived from three flux values per day from each land use type. The error bar presented here is the standard error of the three flux values per day.*

**RC**: Figures 4 and 5 are good. Keeping the color scheme, the same would be helpful.

**AC:** *The two figures provide different information and having them in the same color scheme might be misleading. Figure 4 shows the difference in emissions between the land uses types while figure 5 gives the differences in emission between the wet and dry season.*

**RC:** Figure 6 – put in same color scheme.

**AC:** *This is a relevant point and we tried to have them in the same color scheme of figure 4. However, this then made it very difficult to differentiate between the LUC types***.**

**RC:** Figure 7 – this is at such a large scale, I don't find it very informative. Fig 6 shows the data used.

**AC:** *We agree and we will move Figure 7 into the appendix and keep Figure 6 in the revised manuscript.*

**Discussion**

**RC:** Different terms are being used, soil respiration; soil $CO_2$ emissions, ecosystem soil respiration (?) Make this consistent.

**AC:** *We thank the reviewer for pointing this out. In the revised manuscript, we will only use the term "soil* $CO_2$ *emission".*

**RC**: There is quite a bit of speculation in the discussion. It would be better shortened and more focused on the data collected, not the data lacking that could explain the patterns. This is true for CO2 and N2O sections.

**AC:** *Following this suggestion and a similar point being raised by Reviewer 2, we will make the discussion shorter and more concise – some examples are:*

***Example 1***

***Original*** *-* $CO_2$ *emissions from the soil differed significantly between the land-use types. Higher* $CO_2$ *emissions were particularly observed in the conservation land followed by grazing land and bushland, while lowest levels were recorded from the cropland. We observed the same trend with SOC, and thus we attributed the difference in* $CO_2$ *emissions between the land uses to SOC as observed (see Table 1). This is in line with a similar study by La Scala et al. (2000), which recognized SOC as a key driver of* $CO_2$ *emissions from the soil, as it is the primary source of energy for soil microorganisms (Lal, 2009).*

***Revised*** *- Highest mean* $CO_2$ *fluxes were observed from the conservation land followed by grazing land and bushland, and the lowest from cropland. Soil C content also showed the same trend (conservation land > grazing land > bushland > cropland), which is the primary source of energy for soil microorganisms (Lal 2009) and thus affecting* $CO_2$ *emissions. Therefore, we attributed this variance to the difference in land use and management activities playing a major role in modifying both biotic and abiotic factors (Pinto et al., 2002).*

***Example 2***

***Original*** *- Soil* $CO_2$ *emissions and mean soil C content were lowest in cropland. Root respiration in cropland depends on periods of live roots in annual crop fields and on the biomass of roots during the initial growing season (Raich et al., 2000). Therefore, the continued removal of crop residues*

*during harvesting and frequent tillage affected both root respiration and SOC. As much as crop residues contribute to carbon stocks through their mineralization (Nandwa, 2001), most maize residues were used as livestock feed and sometimes as fuel, while bean residues were removed completely during the harvest and burned.*

***Revised*** *- With the lowest $CO_2$ emissions being reported from the cropland, we attribute this observation to the continued tillage and removal of crop and crop residues during land preparation, weeding and harvesting, affecting both root respiration and soil C content (Raich et al., 2000; Nandwa, 2001). In East Africa and in smallholder farming systems, most of the crop residues are used as livestock feed and fuel.*

***Example 3***

***Original*** *- Manure inputs provide easily degradable substrates of C and N leading to enhanced soil $CO_2$ emissions (Janssens et al., 2001; Davidson and Janssens, 2006). However, manure input in cropland was very low (approximately 20 kg in a 1.5 ha farm per month) and thus no measurable effects in $CO_2$ emissions were detected. This was opposite to our hypothesis. Another reason could be that soil fertility was too low to have a detectable influence on $CO_2$ emissions (Pelster et al., 2017). Our results are in the same magnitude with those of Rosenstock et al. (2016); Farai Mapanda et al. (2011), and Pelster et al. (2017), who also did not detect any change in $CO_2$ emissions after manure application and attributed this to the low input of manure to the maize and sorghum plots.*

***Revised*** *- Manure inputs in cropland were very low (about 20 kg per month on a 1.5 ha farm) and thus no measurable difference on $CO_2$ emissions were detected. Several other studies observed the same scenario from low manure input in maize and sorghum plots (Rosenstock et al., 2016; Farai Mapanda et al., 2011, and Pelster et al. 2017).*

***Example4***

***Original*** *- Soil $N_2O$ emissions can vary highly over time, as regulated by factors such as soil moisture, temperature, aeration, ammonium and nitrate concentrations, pH, and mineralizable carbon (Butterbach-Bahl et al., 2013). However, we did not document any significant difference in $N_2O$ emissions between the four land uses. At all the sites, $N_2O$ emissions were very low, and this we attributed to the observed low soil N content (see Table 1). According to Pinto et al. (2002) and Grover et al. (2012), savanna ecosystems have a very tight N cycling, which transcends to low N availability. Thus, available N is taken up by vegetation, leaving very little for denitrification (Castaldi et al., 2006; Mapanda et al., 2011). Our results are consistent with low $N_2O$ emissions 610 observed in a Brazilian savanna (cerrado) by Wilcke et al. (2005), who also reported low N levels in their study area. Very low $N_2O$ emissions due to poor nutrient availability have also been observed in other savanna landscapes (Scholes et al., 1997; Castaldi et al., 2016). Soil $N_2O$ in cropland match those of Rosenstock et al. (2016), who also attributed the low soil $N_2O$ emissions to poor nutrient availability in the soil*

***Revised*** *- Our results showed low $N_2O$ emissions from all the LUTs, which we attributed to the low soil N content observed (see Table 1). Savanna ecosystems are characterized by very tight N cycling,*

*which transcends to low N availability (Pinto et al., 2002 and Grover et al.,2012) and most of this N is taken up by vegetation, leaving very little for denitrification (Castaldi et al., 2006; Mapanda et al., 2011). The flux results observed from the conservation land, grazing land and bushland are consistent with those observed in a Brazilian savanna by Wilcke et al. (2005). Many other studies from similar ecosystem reported comparable $N_2O$ flux magnitudes (Scholes et al., 1997; Castaldi et al., 2016; Mapanda et al., 2010). The higher $N_2O$ emissions observed in June and July from our cropland site after the maize and bean harvests are likely occurring due to the absence of plants.*

**RC:** Interesting $CH_4$ just gets one sentence because it is small: : : but this is important too!

**AC:** *For the completeness of the paper, methane emissions have to be mentioned. Even though Reviewer 2 suggested to remove everything related to $CH_4$, we decided to keep this information in the manuscript for 2 reasons: (1) completeness in terms of greenhouse gases, and (2) even if the contribution of $CH_4$ is low, this is an important results and similar measurements may not have to be repeated.*

**Soil Greenhouse Gas Emissions under Different Land-Use Types in Savannah Ecosystems of Kenya**

**Authors Response:**

Two reviewers have reviewed the manuscript and an additional comment was made in the open discussion. Before, responding to each reviewer and comment individually, we would like to thank for the constructive comments and informative feedback.

The document is structured as follows: each of the reviewer's comment (indicated by RC) is first repeated followed by our response (indicated as AC and in italic). Where relevant we either include a rephrased sentence already or explain on how we intent to implement suggested changes.

**RC 2**

**D. Otieno (Referee)**

denotieno@yahoo.com

**RC:** This is interesting study conducted in semi-arid parts of Kenya, where similar data are quite scarce. The set-up is an area characterized by a series of activities. It is a surprised that there is some form of cultivation/farming in an area that looks more like Tsavo national Park. Nonetheless, the study provides valuable data that extend our knowledge of ecosystem gas fluxes in this part of the world.

**AC:** *We would like to thank Mr. Otieno for this review and his valuable comments.*

**RC:** The study was conducted in a relatively poor soil. What the authors failed to mention, especially for the cropped and grazed sites was the slope of the field. I tend to imagine that erosion must be playing a critical role in mineralization processes in this place. It looks like the organic/humus, top soil layer is completely gone and what remains is mainly the mineral soils. Unfortunately, the paper is already too long and I will not recommend inclusion of more information on land use history, which would have been helpful in understanding/interpreting these results.

**AC:** *The study area are located is the lowland of the Taita Taveta county, which is very flat. The cropland is at 1070 m a.s.l, bushland at 1076 m a.s.l, grazing land at 970 m a.s.l, and conservation land at 928 m a.s.l. The cropland is received very small quantities of manure and no chemical fertilizer inputs and thus no significant difference in soil C content with the other land use types. In the grazing area, overgrazing was evident as most of the soil was bare especially in dry season. This contributes to soil erosion and compaction of the land by wind and rain and even the livestock while grazing. We will add this information briefly in the revised manuscript.*

**RC:** It's very surprising that temperature and soil moisture had no influence on soil CO2 fluxes. Could it be the method of data collection, with significant data collection gaps that led to this?

**AC:** *Soil $CO_2$ emissions were positively correlated to soil moisture. However, variation in soil temperature for the time of measurements during the day in both dry and the wet season were minor, and thus we found no statistically significant effect of soil temperature upon $CO_2$ emissions for the dataset. Other studies by Brümmer et al. (2009) and Livesley et al. (2011) also found that soil moisture controlled $CO_2$ emissions from savanna soils, rather than soil temperature. However, if we have had the opportunity to measure more frequently – i.e. following a diurnal course – we are confident that an effect of temperature exists. For instance, we found such diurnal course in GHG emissions in a similar ecosystem in Kenya. This was part of another project and is consequently not shown here.*

**RC:** For future, the authors need to consider higher frequencies of data collection. In such arid ecosystems, evaporation is quite high and it is likely that critical information is lost by not collecting data more regularly.

**AC:** *For this study, the sampling frequency was based on seasonal variation, thus the campaigns were targeting the wet, transition and dry season and when moisture and/or management practices are likely to impact GHG emissions. Certainly, we would have preferred more frequent measurements, though given the research question asked and the available resource for this project, we had to make a compromise. However, there is a follow-up study in other Land Use Types with measurements that are more frequent.*

**RC:** $CH_4$ seems to contribute little to this paper, why not exclude it completely? I do not see the two lines of discussion on $CH_4$ are of major benefit to the readers. The paper is already too long and probably removing all the descriptions on $CH_4$ could reduce the number of pages.

**AC:** *We agree that the importance of methane emissions is negligible when compared to the other gases. However, our aim was to look at all three GHGs in this study and due to the lack of available GHG emissions data from such land cover types in this region of the world we still think its beneficial to report these here. Certainly, in order to not further lengthen the paper, we decided to keep this information as short as possible.*

**RC:** The word "Soil Organic Carbon SOC" is introduced in the introductory part of the Ms. In the methods, there is total soil carbon and in the results, I met Soil Carbon. In the discussions, SOC becomes the main discussions line. The authors need to be consistent in the use of these terms, otherwise the readers get confused.

**AC:** *We thank the reviewer for pointing this out and we will harmonize in the revised manuscript accordingly.*

**RC:** Ln 65. Not all savannah belongs to the ASALs. The humid savannas are relatively wet, with green vegetation almost throughout the year. It is therefore not right to make such a sweeping statement.

**AC:** *Noted with thanks and we adjust the revised manuscript accordingly and use drylands instead of ASALs.*

**Specific comments**

**RC:** Ln 67. Note that shrubs are woody vegetation

**AC:** *Noted with thanks.*

**RC:** Ln 88. Revise the sentence. Overstocking leads to grazing pressure. The way the sentence is written is redundant.

**AC:** *Done*

**RC:** Ln 96-7. ---Croplands are still being cleared from natural vegetation----re-write the sentence, it's not making the intended meaning.

**AC:** *Done. Revised --Natural vegetation is being cleared to make way for the expansion of cropland*

**RC:** Ln 104, what's "cropland farming"?

**AC:** *Here we refer to cropping agriculture in the savanna.*

**RC:** Ln 153. The authors need to be clear on the physiognomic characterization of the vegetation they are studying. Here you have woodlands, bushlands and on line 155 you have wood bushlands, which is which?

**AC:** *We harmonize this in the revised manuscript to bushland as found in Tsavo East and West national park.*

**RC:** Ln 156 are Lions also grazers?

**AC:** *Lion are not grazers. On this line, we were mentioning the fauna that the ecosystem supports in general to highlight the importance and the functions of the park.*

**RC:** Ln 160 –other important land use(s)

**AC:** *Corrected*

**RC:** Ln 173. Is the farm rain-fed or not? Are there other sources of moisture input apart from rain?

**AC:** *The farm is totally rain fed.*

**RC:** Ln 237, how deep was the collar inserted into the soil?

**AC:** *The collars we inserted between 5cm to more than 8cm into the soil. We ensured the collars were inserted so the extend above the surface did not hold water during the rainy season and the collars were less likely to be trampled on and broken by large animals*

**Result**

**RC:** Label Fig. 3 as a and b

**AC:** *Done*

**RC:** Ln 377, Sand proportion was lower than what? In comparative sentences, learn also to use "lowest" or "highest" see ln 417.

**AC:** *Sand proportion was lowest in the grazing land (64.3±0.4 %) than in the other study three sites. We will adjust the phrasing in the revised manuscript.*

**RC:** Ln. 456 present data/results according to the chronology of the figures and avoid this back and forth.

**AC:** *Done*

**RC:** Ln 481. Delete (in) before during.

**AC:** *Done*

**Discussion**

**RC:** SOC is only mentioned in the introduction but not in the methodology or results, yet it becomes very prominent in the discussions. Be consistent in the use of terms.

**AC:** *Noted with thanks.*

**RC:** Ln 525 is not correct. You cannot attribute the differences only to vegetation. It is definite that land use itself leads to the differences in soil C. Although this is argued correctly in the later sections, this section should be revised.

**AC:** *Corrected*

Ln 548. The argument with clays is a bit far-fetched anyway.

**AC:** *We removed this argument in the revised manuscript.*

**RC:** Ln 592. ---temperature was measured "down" to 5 cm. I would imagine that 5 cm depth is almost at the surface. What was the deciding factor for installing temp/moisture sensor at this depth? This depth, being close to the surface is associated with very strong temperature fluctuations. It may be one of the reasons why the authors found no temperature correlation with $CO_2$ efflux. Most grass

roots, cereals included, have roots located within 10 cm, and may extend down to 30 cm. the woody vegetation in such dry places have their roots even deeper. Trying to establish relations with variables measured at 5 cm may not yield positive results.

**AC:** *According to a study by Pavelka et al. (2007), daily dynamic of soil $CO_2$ fluxes are affected by soil temperature near the soil surface and hence for correlation between soil $CO_2$ emissions and soil temperature, the measurement of soil temperature at the soil surface, is highly recommended to avoid the inaccuracies. Coupled with this, the ProCheck handheld GS3 sensor (Decagon Devices Inc) for soil moisture and temperature that we were using could only measure up to 5cm. Because, we were taking measurement within or close to the chamber collars, we did not want to cause any soil disturbance. This is also recommended by the GRACENet protocol we were using as our reference protocol.*

**RC**: Ln 593 check the sentence. How does root respiration tap moisture?

**AC:** *Noted. Here we mean roots can still tap moisture from deeper profile and thus root respiration can continue even after the surface moisture has dried up.*

**RC:** Ln 640. Consider soil erosion and volatilization also.

**AC:** *This is an important point and we thank the reviewer for pointing this out.*

**RC:** Ln 651. Use “dung” instead of faeces.

**AC:** *Corrected*

**RC:** Ln 665 what’s T? From nowhere, you introduce T.

**AC:** *T stands for soil temperature.*

**Soil Greenhouse Gas Emissions under Different Land-Use Types in Savannah Ecosystems of Kenya**

**Authors Response:**

Two reviewers have reviewed the manuscript and an additional comment was made in the open discussion. Before, responding to each reviewer and comment individually, we would like to thank for the constructive comments and informative feedback.

The document is structured as follows: each of the short comment (indicated by SC) is first repeated followed by our response (indicated as AC and in italic). Where relevant we either include a rephrased sentence already or explain on how we intent to implement suggested changes.

**Short comments**

Cornelius Oertel

cornelius.oertel@thuenen.de

I want so suggest some changes for that publication:

**SC:** N2O fluxes are sometimes given in µg N2O-N $m^{-2}$ $h^{-1}$ (e.g. l. 295 or 420) and in some graphics they are given in N2O (µg $m^{-2}$ $h^{-1}$) (e.g. figure 4 or figure 5). This is the same for $CO_2$. Units should be used consistent, so that you can compare the fluxes.

**AC:** *We thank Mr. Oertel for pointing this out and have made the necessary correction. $CO_2$ (mg $m^{-2}$ $h^{-1}$) $N_2O$ (µg $m^{-2}$ $h^{-1}$) and $CH_4$ (mg $m^{-2}$ $h^{-1}$).*

**SC:** l. 171: Height a.s.l. is only given for the cropland site and should be given as well for the other sites.

**AC:** *This will be done in the revised manuscript. Cropland is at 1070 m a.s.l, bushland at 1076 m a.s.l, grazing land at 970 m a.s.l, and conservation land at 928 m a.s.l.*

---

## Author Response (AR2)

Sheila Wachiye

University of Helsinki

Gustaf Hällström Street 2

P.O Box 64, 00014

Helsinki. Finland

To the Editor

Biogeosciences Journal

Dear Sir

**Ref: Minor Revision on manuscript "Soil Greenhouse Gas Emissions under Different Land-Use Types in Savannah Ecosystems of Kenya"**

We would like to thank you for the positive feedback we received from you on the above-mentioned manuscript. We also would like to thank our reviewers for their positive feedback and the constructive comments. Based on the comments, we have made all the corrections for areas we agree with the reviewer and given justifications where otherwise in the document attached.

Looking forward to your positive response.

Sincerely

Sheila Wachiye

**Minor Revision on manuscript "Soil Greenhouse Gas Emissions under Different Land-Use Types in Savannah Ecosystems of Kenya"**

This document is therefore structured with each of the reviewer's comment (indicated by RC) followed by our response (indicated as AC and in italic).

**RC: Line 35: could also be microbial/mycorrhizal respiration (especially if respiring root exudates)**

*AC: We thank the reviewer for pointing this out and we agree. In the abstract, we limited our conclusion to what we observed during the campaigns, while in the discussion we have provided an explanation on the importance of soil microbial/mycorrhizal activity in seasonal variation of soil GHG emissions.*

**RC: 50(ish): describe how soil can be a sink for $N_2O$ and $CO_2$ as well; the text as written only describes their source dynamics, which contrasts the first sentence of the introduction.**

*AC: We have now included this in the first paragraph of the introduction, which reads as below:*

Soil is a major source, and in many cases also a sink, of the atmospheric greenhouse gases (GHG) carbon dioxide ($CO_2$), nitrous oxide ($N_2O$), and methane ($CH_4$) (Oertel et al., 2016). The concentrations of these gases have increased since the onset of the industrial revolution (from about1750), leading to global warming (IPCC, 2013). GHGs trap the long-wave radiation emitted by the Earth's surface, thus increasing surface temperatures (Arrhenius, 1896). Soil $CO_2$ emissions originate from root & mycorrhiza respiration and heterotrophic decomposition of soil organic matter (Oertel et al., 2016). In addition to being a $CO_2$ source, by increasing the soil organic carbon (SOC) content soils can also act as a sink for $CO_2$. $N_2O$ on the other hand can be produced from many pathways in the soil nitrogen (N) cycle, but is considered to result primarily from nitrification and denitrification (Butterbach-Bahl et al., 2013). $N_2O$ uptake into soils is also possible as observed previously (e.g.(Butterbach-Bahl et al., 2002; Rosenkranz 2006; Flechard et al., 2005), and it depends on the complete reduction of $N_2O$ to $N_2$, the ease of $N_2O$ diffusion within the soil profile, and its dissolution in soil water (Chapuis-Lardy et al. 2007). $CH_4$ is produced by methanogenesis under anaerobic conditions and consumed by methanotrophic microorganisms under aerobic conditions, with the latter being more important in well-aerated upland soils, which consequently show net $CH_4$ uptake (i.e. negative flux) (Serrano-silva et al., 2014; Hanson and Hanson 1996).

**RC: Line 57: I recommend also adding the critical role of plants to soil GHG fluxes**

*AC: We have now included the role of vegetation in the second paragraph of our introduction as follows below:*

The production and consumption of soil GHGs largely depend on soil physical and chemical properties (Davidson et al., 2006) (e.g. texture, soil organic matter and pH) and are further driven by environmental factors such as soil moisture and soil temperature (Davidson et al., 2006). In addition, vegetation affects both biotic and abiotic factors that drive soil emissions (Raich and Tufekcioglu, 2000; Pinto et al., 2002) and net carbon assimilation (La Scala et al., 2000). Vegetation type directly influences soil physicochemical properties, which in turn modify soil microbial activities (Raich and Tufekcioglu, 2000). It also controls the quantity of plant carbon allocated belowground (Metcalfe et al., 2011) by determining root biomass and litter quality and quantity (Fanin et al., 2011; Rey et al., 2011). Vegetation composition additionally affects root respiration and the associated microbial components. Active roots add directly to soil respiration, while dead roots and root exudates provides carbon as a source of energy and nutrients for soil microbial biomass (Tufekcioglu et al., 2001). Hence, changes in vegetation types and cover due to land-use system and land-use management activities have the potential to modify the soil-to-atmosphere GHG exchange (Raich and Tufekcioglu 2000). Thus, soil GHG emissions and uptake along with their controlling factors differ between biomes based on land use and land-use management.

**RC: Line 241: A brief description of the power regression for $N_2O$ would be forthcoming.**

*AC: We thank the reviewer for this comment. Therefore, we have included both the functions for linear model and the power model as follows.*

$$Conc_{CO_2, \ CH_4} = ax + \beta \qquad\qquad (1)$$

$$Conc_{N_2O} = ax^{\beta} \qquad\qquad (2)$$

*Where $Conc_{CO_2, \ CH_4}$ are the carbon dioxide and methane concentrations in ppm, $F_{N_2O}$ is the nitrous oxide concentration in ppb, a and β are model coefficients, and x is the peak area derived from the GC. Both equations are based on peak area measurements of known standards with our GCs, and while the FID ($CO_2$ & $CH_4$ detection) is linear over the entire concentration range, the ECD ($N_2O$ detection) behaves non-linearly and therefore a power function leads to better fits.*

**RC: Line 250: what if the outlier was at the beginning or end of measurement indicating a potential nonlinearity?**

*AC: When an outlier was found, it was investigated to establish whether it was genuine or error. Most outliers found were discarded because they were found to be erroneous (i.e. leaky gas vials, etc.). For our study, the majority of the data collected during each campaign fulfilled the conditions for linearity, which is why we fitted linear models.*

**RC: Line 264: the SI unit for time is seconds, but I am ok with presenting values per hour. (And 'represents'; minor usage errors should be reviewed throughout the manuscript, e.g. a missing period on p. 607 and 'value' on 595).**

*AC: We thank the reviewer for this observation. We have corrected this throughout the manuscript.*

**RC: Equation 2: what is the 0.02241?**

*AC: We apologise for not including the meaning of this value in our equation. The value 0.02241 is the molar volume of an ideal gas in $m^3 \, mol^{-1}$ at standard temperature and pressure following the ideal gas law. We have made this correction in the manuscript.*

**SOIL GREENHOUSE GAS EMISSIONS UNDER DIFFERENT LAND-USE TYPES IN SAVANNA ECOSYSTEMS OF KENYA**

**Authors:** *Sheila Wachiye[1, 2, 5], Lutz Merbold[3], Timo Vesala[2], Janne Rinne[4], Matti Räsänen[2], Sonja Leitner[3], and Petri Pellikka[1, 2]

1) Earth Change Observation Laboratory, Department of Geosciences and Geography, University of Helsinki, Finland
2) Institute for Atmosphere and Earth System Research, University of Helsinki, Finland
3) Mazingira Centre, International Livestock Research Institute (ILRI), Nairobi, Kenya
4) Department of Physical Geography and Ecosystem Science, Lund University, Sweden
5) School of Natural Resources and Environmental Management, University of Kabianga, Kenya

*Correspondence email: sheila.wachiye@helsinki.fi

**Abstract**

Field measurement data on greenhouse gas (GHG) emissions are still scarce for many land-use types in Africa, causing a high uncertainty in GHG budgets. To address this gap, we present *in situ* measurements of carbon dioxide ($CO_2$), nitrous oxide ($N_2O$), and methane ($CH_4$) emissions from the lowlands of southern Kenya. We conducted eight chamber measurement campaigns on gas exchange from four dominant land-use types (LUTs) including (1) cropland, (2) bushland, (3) grazing land, and (4) conservation land between 29 November 2017 to 3 November 2018, accounting for regional seasonality (wet and dry seasons, and transitions periods). Mean $CO_2$ emissions for the whole observation period were significantly highest ($p$-value$<0.05$) in the conservation land ($75\pm6$ mg $CO_2$-C m$^{-2}$ h$^{-1}$) compared to the three other sites, which ranged from $45\pm4$ mg $CO_2$-C m$^{-2}$ h$^{-1}$ (bushland) to $50\pm5$ mg $CO_2$-C m$^{-2}$ h$^{-1}$ (grazing land). Furthermore, $CO_2$ emissions varied between seasons, with significantly higher emissions in the wet season than the dry season. Mean $N_2O$ emissions were highest in cropland ($2.7\pm0.6$ μg $N_2O$-N m$^{-2}$ h$^{-1}$) and lowest in bushland ($1.2\pm0.4$ μg $N_2O$-N m$^{-2}$ h$^{-1}$) but did not vary with season. In fact, $N_2O$ emissions were very low both in the wet and dry seasons, with slightly elevated values during the early days of the wet seasons in all LUTs. On the other hand, $CH_4$ emissions did not show any significant differences between LUTs and seasons. Most $CH_4$ fluxes were below the limit of detection (LOD, $\pm0.03$ mg $CH_4$-C m$^{-2}$ h$^{-1}$). We attributed the difference in soil $CO_2$ emissions between the four sites to soil C content, which differed between the sites and was highest in the conservation land. In

addition, $CO_2$ and $N_2O$ emissions positively correlated with soil moisture, thus an increase in soil moisture led to an increase in emissions. Furthermore, vegetation cover explained the seasonal variation of soil $CO_2$ emissions as depicted by a strong positive correlation between NDVI and $CO_2$ emissions, most likely because with more green (active) vegetation cover, higher $CO_2$ emissions

35  occur due to enhanced root respiration compared to drier periods. Soil temperature did not show a clear correlation with either $CO_2$ or $N_2O$ emissions, which is likely due to the low variability in soil temperature between seasons and sites. Based on our results, soil C, active vegetation cover and soil moisture are key drivers of soil GHG emissions in all the tested LUTs in South Kenya. Our results are within the range of previous GHG flux measurements from soils from various LUTs in other parts

40  of Kenya and contribute to more accurate baseline GHG emission estimates from Africa, which are key to reduce uncertainties in global GHG budgets as well as for informing policymakers when discussing low-emission development strategies.

**KEYWORDS: Carbon Dioxide, Nitrous Oxide, Methane, Bushland, Conservation, Grazing land, Cropland.**

45  **1. Introduction**

Soil is a major source, and in many cases also a sink, of the atmospheric greenhouse gases (GHG) carbon dioxide ($CO_2$), nitrous oxide ($N_2O$), and methane ($CH_4$) (Oertel et al., 2016). The concentrations of these gases have increased since the onset of the industrial revolution (from about1750), leading to global warming (IPCC, 2013). GHGs trap the long-wave

50  radiation emitted by the Earth's surface, thus increasing surface temperatures (Arrhenius, 1896). Soil $CO_2$ emissions originate from root & mycorrhiza respiration and heterotrophic decomposition of soil organic matter (Oertel et al., 2016). In addition to being a $CO_2$ source, by increasing the soil organic carbon (SOC) content soils can also act as a sink for $CO_2$. $N_2O$ on the other hand can be produced from many pathways in the soil nitrogen (N) cycle, but is considered to result primarily

55  from nitrification and denitrification (Butterbach-Bahl et al., 2013). $N_2O$ uptake into soils is also possible as observed previously (e.g.Butterbach-Bahl et al., 2002; Rosenkranz 2006; Flechard et al., 2005), and it depends on the complete reduction of $N_2O$ to $N_2$, the ease of $N_2O$ diffusion within the soil profile, and its dissolution in soil water (Chapuis-Lardy et al. 2007). $CH_4$ is produced by methanogenesis under anaerobic conditions and consumed by methanotrophic microorganisms under

60  aerobic conditions, with the latter being more important in well-aerated upland soils, which consequently show net $CH_4$ uptake (i.e. negative flux) (Serrano-silva et al., 2014; Hanson and Hanson 1996).

**Commented [WS1]:** RC: 35: could also be microbial/mycorrhizal respiration (especially if respiring root exudates)

**Commented [WS2R1]:** *AC: We thank the reviewer for pointing this out and we agree. In the abstract we limited our conclusion to what we observed during the campaigns, while in the discussion we have provided an explanation on the importance of soil microbial/mycorrhizal activity in seasonal variation of soil GHG emissions*

**Commented [WS3]:** RC: 50(ish): describe how soil can be a sink for N2O and CO2 as well; the text as written only describes their source dynamics which contrasts the first sentence of the introduction.

**Commented [WS4R3]:** AC: Done

The production and consumption of soil GHGs largely depend on soil physical and chemical properties (Davidson et al., 2006) (e.g. texture, soil organic matter and pH) and are further driven by environmental factors such as soil moisture and soil temperature (Davidson et al., 2006). In addition, vegetation affects both biotic and abiotic factors that drive soil emissions (Raich and Tufekcioglu, 2000; Pinto et al., 2002) and net carbon assimilation (La Scala et al., 2000). Vegetation type directly influences soil physicochemical properties, which in turn modify soil microbial activities (Raich and Tufekcioglu, 2000). It also controls the quantity of plant carbon allocated belowground (Metcalfe et al., 2011) by determining root biomass and litter quality and quantity (Fanin et al., 2011; Rey et al., 2011). Vegetation composition additionally affects root respiration and the associated microbial components. Active roots add directly to soil respiration, while dead roots and root exudates provides carbon as a source of energy and nutrients for soil microbial biomass (Tufekcioglu et al., 2001). Hence, changes in vegetation types and cover due to land-use system and land-use management activities have the potential to modify the soil-to-atmosphere GHG exchange (Raich and Tufekcioglu 2000). Thus, soil GHG emissions and uptake along with their controlling factors differ between biomes based on land use and land-use management.

Land-use changes are reportedly the largest source of anthropogenic GHG emissions in Africa (Valentini et al., 2014). However, *in situ* studies on GHG emissions from various ecosystems in remain scarce, particularly from savanna ecosystems (Castaldi et al., 2006). Savanna is an important land cover type in Africa, covering more than 40 % of its total area (Scholes et al., 1997). In Kenya, savanna and grassland ecosystems cover about 80 % of the total area, comprising various land-use types (LUTs) (GoK, 2013). These ecosystems are subject to accelerating land-use change (Grace et al., 2006) due to population growth (Meyer and Turner, 1992) and land-use management activities (Valentini et al., 2014). Conversion of savanna for small- and large-scale livestock production, crop cultivation, and human settlement is common in Africa (Bombelli et al., 2009). As a consequence, vegetation cover, net primary productivity, allocation of carbon and nutrients in plants and soil (Burke et al., 1998) as well as soil GHG emissions are affected (Abdalla et al., 2018; Carbone et al., 2008).

Overgrazing due to overstocking is a major cause of soil and vegetation degradation in large parts of African savannas (Patton et al., 2007; Abdalla et al., 2018). Factors associated with grazing include animal feeding preferences to specific plant species, thus creating higher pressure for those species, which decline in numbers over time, leading to species loss and lower pasture nutritive value (Patton et al., 2007). In addition, soil trampling increases soil bulk density and reduces soil water infiltration

Commented [WS5]: RC: 57: I recommend also adding the critical role of plants to soil GHG fluxes

Commented [WS6R5]: AC: Done

95  (Patton et al., 2007). Furthermore, high rates of dung and urine deposition, especially around homesteads and waterholes, create high N concentrations that are toxic for many savanna grass species, affecting vegetation cover and composition (e.g. increase of encroaching species such as *Solanum incanum* L., which is toxic for livestock (van Vegten, 1984)). Given that all these factors affect soil properties, soil GHG emissions are most likely similarly affected (Wilsey et al., 2002).

100 In addition to overgrazing, rapid human population growth leads to more people migrating into savanna ecosystems, which has led to the expansion of cropland (Pellikka et al., 2018; Patton et al., 2007). Brink and Eva (2009) found that the area under cropland increased by 57 % between 1975 and 2000 in Africa. In the Horn of Africa, cropland areas increased by  28 % between 1990 and 2010 (Brink et al., 2014), while wooded vegetation in East Africa decreased by 5 % in forests, 16 % in
105 woodlands, and 19 % in shrublands (Pfeifer et al., 2013). As an additional example, in our study area Taita Taveta County in Southern Kenya, the area under cropland increased from 30 % in 1987 to 43 % in 2011 (Pellikka et al., 2018). However, in the Taita Hills, located in the County, this trend has slowed down in recent years, while the savanna lowlands are still being cleared to make way for new cropland (Pellikka et al., 2013).

110 Croplands in the Kenyan savannas are mostly managed by smallholder farmers (Waswa and Mburu, 2006). Due to high poverty levels in this region, inputs to improve crop yields, such as the use of fertilizer and herbicide, and mechanized farming are minor (Waswa and Mburu, 2006; CIDP, 2014). Thus, an increase in productivity are mostly via cropland expansion. In spite of this, these smallholder farms are likely to have substantial effects on national GHG emission budgets (Pelster et al., 2017).
115 Until now, only a few studies have investigated soil GHG emissions from such agricultural landscapes (Rosenstock et al., 2016), and these studies were mostly carried out in high-potential farming areas such as the Kenyan highlands, which receive >1000 mm rainfall (FAO, 1996). For example, Rosenstock et al. (2016) showed a large variation of $CO_2$ and $N_2O$ emissions both within and between four crop types as affected by environmental conditions and land management. However, studies
120 measuring GHG emissions from low-productivity croplands in southern Kenya are to the best of our knowledge still missing. Thus, this study focused on soil GHG emissions from different LUTs relevant for the semi-arid region of Southern Kenya.

Given the vast area covered by savanna, land use and land-cover changes are likely to affect global, regional, and national C and N cycles, and hence the quantification of their role is vital (Lal, 2004;
125 Williams et al., 2007). Studies in Kenya have shown large variations of soil GHG emissions in various savanna ecosystems (Otieno et al., 2010; Oduor et al., 2018), due to land-use (Ondier et al., 2019)

and management activities (K'Otuto et al., 2013). Owing to the high diversity of these savanna ecosystems, such studies may not be entirely representative for every region (Ardö et al., 2008).

The lack of reliable soil GHG flux data from natural savanna and cropland limits our understanding of GHG emissions from African soils (Hickman et al., 2014; Valentini et al., 2014). At the same time, accurate quantification of GHG emissions from multiple LUTs are essential to allow for reliable estimation of Kenya's national GHG inventory (IPCC, 2019). This is particularly important as Kenya currently relies on a Tier-1 approach by using default emission factors (EFs) provided in the Guidelines for Greenhouse Gas Inventories of the UN Intergovernmental Panel on Climate Change (IPCC) to estimate national GHG emission budgets. Following the Paris Climate Agreement (https://unfccc.int/process-and-meetings/the-paris-agreement/d2hhdC1pcy), most countries across the globe, including Kenya, have not only agreed to accurately report their GHG emissions at national scales following a Tier-2 approach (i.e. using localized data) but also to mitigate anthropogenic GHG emissions in the upcoming decades, as is communicated via their Nationally Determined Contributions (NDCs). Both can only be achieved with locally derived data.

To address the lack of localized GHG emission data from different LUTs in Kenya, our study aims at: (1) providing crucial baseline data on soil GHG emissions from four dominant land uses, namely conservation land, grazing land, bushland, and cropland, and (2) investigating abiotic and biotic drivers of GHG emissions during different seasons. We hypothesized that GHG emissions in cropland would be higher compared to grazing land, bushland, and conservation land because of larger nutrient inputs (i.e. fertilization) in managed land. Further, we hypothesized that GHG emissions would differ between seasons; more precisely, we expected higher GHG emissions in the wet season than in the dry season caused by higher soil moisture.

**2. Materials and Methods**

**2.1. Study Area**

This study was conducted in the lowlands (800–1000 m a.s.l.) of Taita Taveta County (latitude 3° 25´ S and longitude 38° 20´ E) located in southern Kenya (Fig. 1). Taita Taveta County is one of Kenya's dryland areas, with 89 % of the area characterized as arid and semi-arid area. The county is divided into three major geographical regions, namely the mountainous zone of the Taita Hills (Dawida, Kasigau, Sagalla), Taita lowlands, and the foot slopes of Mt. Kilimanjaro around Taveta. In the lowlands, vegetation types include woodlands, bushlands, grasslands, and riverine forests/swamps.

Tsavo East and Tsavo West National Parks covers ca. 62 % of the county area (CIDP, 2014). The parks are open savanna and bushland that support large herbivores, predators and a wealth of birdlife. There are 28 ranches designated for livestock production and two wildlife sanctuaries (Taita Hills Wildlife Sanctuary and LUMO Community Wildlife Sanctuary). Other important land uses include cropland under small-scale farming (CIDP, 2014), shrublands, and sisal farming (Pellikka et al., 2018). Soil type is characterized by dark red, very deep, acid sandy clay soil (Ferralsols). Our study sites were located in four of these key land uses in the region including cropland, bushland, wildlife conservation land, and grazing land.

The lowland has a bimodal rainfall pattern with two rainy seasons – a long rain season between March and May and a short rain season between October and December (CIDP, 2014). The hot and dry months are January and February while the dry season from June to October is cooler (Pellikka et al., 2018). Mean annual rainfall is 500 mm and the mean annual air temperature is 23 °C, with an average daily minimum and maximum temperature of 16.7 °C and 28.8 °C respectively (CIDP, 2014).

The first site investigated is cropland located in Maktau (1070 m a.s.l., Fig. 1, Fig. 2a) about one and a half hectares, cultivated with maize (Zea mays L.) intercropped with beans. The farm is a typical rain-fed smallholder farm and crop growing closely follows the rainy seasons, with sowing in March, and harvesting in June for beans and August for maize. Animal ploughing is done to prepare land before seeding and weeding is by hand hoeing. Small quantities of fresh and dry manure (roughly 20 kg, accounting for less than 1 kg of N) were used every month to improve soil fertility.

The second site is located in a private bushland in Maktau next to the cropland (1076 m a.s.l., Fig. 1, Fig. 2b). In this region, bushland is found both within the conservation areas and under private ownership. Bushland forms a cover with over 50 % of thorny shrubs and small trees, characterized by *Acacia spp* and *Commiphora spp*. The bushes may vary in height ranging from two to five metres. Herbs and savanna grasses (mostly annual or short-lived perennials) less than one-metre tall form the ground cover. Private bushland similar to our study site is used by the farmers to generate small income from forest products such as timber, poles, and firewood, and charcoal to some extent. Additionally, some grazing occurs primarily by livestock owned by the farmer (CIDP, 2014).

The third site, grazing land (covering approximately 460 km$^2$) is located in the LUMO Community Wildlife Sanctuary (970 m a.s.l, Fig. 1, Fig. 2c) next to Tsavo West National Park and Taita Hills Wildlife Sanctuary. The sanctuary was formed by merging three ranches, namely Lualenyi and Mramba communal grazing areas and Oza group ranch thus the name "LUMO". This sanctuary is

communally owned (GoK, 2013) designated for community livestock grazing, where wildlife is also present, as conservation areas are not necessarily fenced. However, overgrazing is a major challenge, caused by herders who enter the conservancy illegally especially in the dry season (CIDP, 2014).

The forth site is the conservation land located within the Taita Hills Wildlife Sanctuary (928 m a.s.l., Fig. 1, Fig. 2d) covering an area of ca. 110 km$^2$. This is a private game sanctuary for wildlife conservation located between LUMO and communal land. The sanctuary is an open savanna grassland dominated by *Schmidtia bulbosa* and *Cenchrus ciliaris* grass species forming an open to closed ground cover, shrublands, and scattered woodlands with *Acacia spp.* as main tree species. However, most trees have been damaged by elephants, leaving the landscape open. The sanctuary is well managed with the application of ecological management tools such as controlled fires. Through these and other conservation efforts, the sanctuary has attracted a higher diversity of large mammals, many of which remain within the unfenced sanctuary throughout the year. Wildlife are the predominant grazers and browsers, although livestock encroachment may be a problem especially during the dry season on the western and eastern borders of the sanctuary (GoK, 2013).

[Figure]

***Figure 1.*** *Location of the study sites cropland, bushland, grazing land, and conservation land in the savanna area in the lowlands of Taita Taveta County in southern Kenya. Image showing the sites is Sentinel-2A acquired from Sentinel's Scientific DataHub (ESA, 2015). The Kenyan and African boundary ©World Resources Institute (retrieved from https://www.wri.org/resources/data-sets/kenya-gis-data)*

[Figure]

***Figure 2:*** *The four land-use types: (a) cropland, (b) bushland, (c) grazing land, and (d) conservation
land. The upper panel shows the land-use types during the wet season, while the lower panel depicts
the situation during the dry season. The grey plastic collars visible in upper left photo are frames for
the GHG flux chambers.*

**2.2. Defining the seasons**

We divided our campaigns into dry and wet seasons, based on the agro-climatic concept. The onset
of the wet season was the first wet day of a 3 day wet spell receiving at least 20 mm without any
10 day dry spell (< 1 mm) in the next 20 days from 1 March for the long wet season and 1 September
for the short wet season (Marteau et al., 2011). Equally, the end of the rainy season was the first of
10 consecutive days with no rain. Thus, for this study, the long wet season (LW) was between 2
March to 4 June 2018, and the short wet season (SW) between 23 October and 26 December 2018.
The two wet seasons were separated by two dry seasons, the short dry season (SD) from January to
February 2018, and the long dry season (LD) from June to September 2018. We had three campaigns
in each of the wet season: the early days of the wet seasons onset (onset-SW, onset-LW), the peak of
the seasons (mid-SW, mid-LW), as well as at the end of the seasons (end-SW, end-LW).

**2.3. Chamber measurements of greenhouse gas emission**

Soil-atmosphere exchange of $CO_2$, $N_2O$, and $CH_4$ were measured in eight one-week campaigns from
29 November 2017 to 3 November 2018 using the static chamber method (Rochette, 2011;
Hutchinson et al., 1981). Within each of the four sites (LUTs), three clusters were randomly selected
as replicates for soil GHG concentration measurements. In each cluster, three plastic collars (27cm ×
37.2 cm × 10 cm) were inserted (5–8 cm) into the soil at least 24 hours before the first sample was
taken (see Pelster et al., 2017 for further details). The collars were left in the ground for the entire

measurement period to minimize soil disturbance during measurements (Søe et al., 2004). Any damaged or missing collars (mostly due to livestock or wildlife activity) were replaced, at least 24 hours before the next gas sampling. During each day of a campaign, gas sampling was conducted daily between 7:00 and 11:00 am, which is about the average flux of the diurnal cycle ( Parkin and
235    Venterea, 2010).

During each gas-sampling day, grey opaque PVC lids (27 cm × 37.2 cm × 12 cm) covered with reflective tape were placed onto the collars for 30 mins. Lids were fitted with a fan for gas mixing and a vent to avoid pressure differences between the chamber headspace and outside atmosphere (Pelster et al., 2017). A rubber seal was fitted along the edges of the chamber lid and paper clips used
240    to hold the lid and collar in place to ensure airtightness. Four gas samples were then collected every 10 mins (time 0, 10, 20, 30 mins) after lid deployment (Rochette, 2011). The height of each chamber collar was measured on each sampling day to derive the total chamber volume (total chamber height = height of chamber collar sticking out of the soil + height of the chamber lid). A slightly modified version of the gas-pooling method was used to reduce the overall sample size while ensuring a good
245    spatial representation of each LUTs (see Arias-Navarro et al., 2013). Here, 20 ml of headspace air were collected from each of the three chambers at each time interval with polypropylene syringes (60 ml capacity), resulting in a composite gas sample of 60 ml. The first 40 ml were used to flush the vials, and the remaining 20 ml over-pressured into 10 ml glass vials to minimize contamination of the gas with ambient air during transportation (Rochette et al., 2003).

250    Gas samples were transported to the laboratory (Mazingira Centre, mazingira.ilri.org) and analysed using a gas chromatograph (GC, model SRI 8610C). The GC was fitted with a $^{63}$Ni-Electron Capture Detector (ECD) for detecting $N_2O$ concentrations and a Flame Ionization Detector (FID) fitted with a methanizer for $CH_4$ and $CO_2$ analysis. The GC was operated with a Hayesep D packed column (3 m, 1/8″) at an oven temperature of 70 °C, while ECD and FID detectors were operated at a
255    temperature of 350 °C. Carrier gas ($N_2$) flow rate was 25 mL min$^{-1}$ on both FID and ECD lines. In every 40 samples analysed with the GC were eight calibration gases with known $CO_2$, $CH_4$, and $N_2O$ concentrations in synthetic air (levels of calibration gases ranged from 400 to 2420 ppm for $CO_2$, 360 to 2530 ppb for $N_2O$, and 4.28 to 49.80 ppm for $CH_4$). Therefore, the gas concentrations of the samples were calculated from peak areas of samples in relation to peak areas of standard gases using

260 a linear model for $CO_2$ and $CH_4$ and a power regression for $N_2O$, using Eq (1) that follows.

Commented [WS7]: 241: A brief description of the power regression for N2O would be forthcoming.

Commented [WS8R7]: Include both models

$$Conc_{CO_2, \ CH_4} = ax + \beta \qquad\qquad (1)$$

$$Conc_{N_2O} = ax^\beta \qquad\qquad (2)$$

Where $Conc_{CO_2, \ CH_4}$ are the carbon dioxide and methane concentrations in ppm, $F_{N_2O}$ is the nitrous oxide concentration in ppb, $a$ and $\beta$ are model coefficients, and x is the peak area derived from the

265 GC. Both equations are based on peak area measurements of known standards with our GCs, and while the FID ($CO_2$ & $CH_4$ detection) is linear over the entire concentration range, the ECD ($N_2O$ detection) behaves non-linearly and therefore a power function leads to better fits.

**2.4. Greenhouse gas flux calculations**

Soil GHG emissions were determined by the rate of change in gas concentration in the chamber
270 headspace over time by linear fitting. The goodness of fit was used to evaluate the linearity of concentration increases/decreases. The dynamics of the $CO_2$ concentrations over the 30 min deployment period for each gas concentration was assessed to test for chamber leakage due to the typically more robust and continuous flux of $CO_2$ (Collier et al., 2014). If the linear model of $CO_2$ versus deployment time had an $R^2 > 0.95$ using all four-time points (T1, T2, T3, and T4), the
275 measurement was considered valid and four-time points were used for analysing the $CO_2$, $N_2O$, and $CH_4$ emissions. However, if $R^2 < 0.95$ for $CO_2$ and one data point was a clear outlier, this point was discarded and the three remaining points used for the flux calculation if they showed a strong correlation of $CO_2$ versus time. Measurements that did not show a clear trend of $CO_2$ with time were considered faulty, and the entire data point series was discarded. In addition, data points that showed
280 a decrease in $CO_2$ concentration over time were assumed to indicate leakage and were similarly discarded (chambers were opaque, i.e. photosynthesis was inactive during chamber deployment). However, if no leakage was found, negative $CH_4$ and $N_2O$ emissions were accepted as the uptake of the respective gas by the soil. Emissions were calculated according to Eq. (1):

Commented [WS9]: RC: 250: what if the outlier was at the beginning or end of measurement indicating a potential nonlinearity?

Commented [WS10R9]: AC: When an outlier was found, it was investigated to establish whether it was genuine or error. Most outliers found were discarded because they were found to be erroneous (i.e. leaky gas vials, etc.). For our study, the majority of the data collected during each campaign fulfilled the conditions for linearity, which is why we fitted linear models.

$$F_{GHG} = \frac{(\frac{dc}{dt}) \times V_{ch} \times M_w}{A_{ch} \times Mv_{corr}} \, 60 \times 10^6, \qquad\qquad (3)$$

285 Where $F_{GHG}$ = soil GHG flux ($CO_2$, $N_2O$, or $CH_4$), $\partial c/\partial t$ = change in chamber headspace gas concentration over time (i.e. slope of the linear regression), $V_{ch}$ = volume of the chamber headspace ($m^3$), $M_w$ = molar weight (g mol$^{-1}$) of C for $CO_2$ and $CH_4$ (12) or N for $N_2O$ (2x N = 28), $A_{ch}$ = area

covered by the chamber ($m^2$) and $Mv_{corr}$ = pressure- and temperature-corrected molar volume (Brümmer et al., 2008) using Eq. (2). With 60 and $10^6$ being, constants used to convert minutes into

290   hours and micrograms respectively. Temperature in Eq. (2) is the air temperature in the chamber headspace measured during each sampling, and 0.02241 the molar volume of a gas at standard temperature and pressure ($m^3$ $mol^{-1}$).

$$Mv_{corr} = 0.02241 \cdot \frac{273.15 + Temp(°C)}{273.15} \times \frac{Atmospheric\ pressure\ at\ measurement\ (Pa)}{Atmospheric\ pressure\ at\ sea\ level\ (Pa)}$$

(4)

295   The minimum limit of detection (LOD) for each gas was calculated following Parkin et al. (2012) and levels were $\pm 4.9$ mg $CO_2$-C $m^{-2}$ $h^{-1}$ for $CO_2$, $\pm 0.04$ µg $N_2O$-N $m^{-2}$ $h^{-1}$ for $N_2O$, and $\pm 0.03$ mg $CH_4$-C $m^{-2}$ $h^{-1}$ for $CH_4$. However, we included all data in the analysis, including those below LOD in line with Croghan and Egeghy (2003), who noted that including such data provides an insight on the distinct measurements, thus  clarifying the set of environmental observations.

300   **2.5. Auxiliary measurements**

During each gas-sampling day, we measured soil water content (WC) and soil temperature (T) (at a depth of 0–5 cm) adjacent to the collar using a handheld data logger with a GS3 sensor (ProCheck METER Group, Inc. USA). Daily air temperature and precipitation data from November 2017 to November 2018 were obtained from a weather station in Maktau located within the cropland site

305   (Tuure et al., 2019). A soil auger was used to collect soil samples (at a depth of 0–20 cm) during the wet season (22 May 2018) in each site for soil chemical and physical property analysis. For bulk density, we collected a combination of three samples from each cluster close to each chamber collar at depths of 0–10 cm and 10–20 cm using a soil bulk density ring (Eijkelkamp Agrisearch Equipment, Giesbeek, The Netherlands). Samples were stored in airtight polyethylene bags and kept in a cooler

310   box with ice packs before transportation to the laboratory for further analysis. In the laboratory, samples were stored in a refrigerator (4 °C) and analysed within 10 days.

The samples were sieved at < 2 mm before analysis. Soil water content was measured by drying soil at 105 °C for 48 h. Soil pH was determined in a 1:2.5 (soil : distilled water) suspension using an electrode pH meter (3540 pH and conductivity Meter, Bibby Scientific Ltd, UK) and soil texture

315   using the hydrometer technique (Scrimgeour, 2008; Reeuwijk, 2002). Total soil C and N content, a duplicate of 20 g of fresh sample was oven-dried at 40 °C for 48 hours and ground into a fine powder

**Commented [WS11]:** 264: the SI unit for time is seconds, but I am ok with presenting values per hour. (and 'represents'; minor usage errors should be reviewed throughout the manuscript, e.g. a missing period on p. 607 and 'value' on 595).

**Commented [WS12R11]:** CorrecteD

**Commented [WS13]:** RC: Equation 2: what is the 0.02241?

**Commented [WS14R13]:** *AC: We apologise for not including the meaning of this value in our equation. The value 0.02241 is the molar volume of an ideal gas in $m^3$ $mol^{-1}$ at standard temperature and pressure following the ideal gas law. We have made this correction .*

using a ball mill (Retsch MM400). Approximately 200 mg of the dry sample were measured by elemental analysis (Vario MAX Cube Analyzer Version 05.03.2013).

**2.6. Statistical Analysis**

320 Statistical analyses were carried out using R Statistical Software (R 3.5.2, R Core Team). Spearman correlation coefficients were performed among the variables followed by the Kruskal Wallis test to assess significant differences of GHG emissions between the LUTs and across seasons. A post-hoc analysis involving pairwise comparisons using the Nemenyi test was performed where significant differences existed. Significance level was set at $p < 0.05$. We assessed the correlation between soil

325 GHG emissions with T and WC using several functions based on the coefficient of determination ($R^2$), root-mean-square error (RMSE) and Akaike's information criterion (AIC). There being no difference in the outputs, we present results from the Gaussian function (O'Connell, 1990) for the correlation between soil emissions and T using Eq. (3), and a quadratic function for correlation with WC using Eq. (4). We also evaluated the combined effect of T and WC on soil GHG emissions by

330 combining Eq. (3) and Eq. (4) into Eq. (5) to assess the effect of these two variables on the emissions.

$$Rs = ae^{(bT+cT^2)} \tag{3\underline{5}}$$

$$Rs = a + bWC + cWC^2 \tag{4\underline{6}}$$

$$Rs = e^{(aT+bT^2)} \times (cWC + dWC^2) \tag{5\underline{7}}$$

Where *Rs* is soil GHG ($CO_2$ and $N_2O$) emissions, *T* is soil temperature ($^o$C), and *WC* is soil water

335 content ($m^3\,m^{-3}$), while *a, b, c,* and *d* *are* the model coefficients.

After no correlation with T and a weak correlation with WC were observed, we included vegetation cover. For this, we used Normalized Difference Vegetation Index (NDVI) from Moderate Resolution Imaging Spectroradiometer (MODIS) from https://ladsweb.modaps.eosdis.nasa.gov. NDVI quantifies vegetation vigour by measuring the difference between reflectance in near infrared (which

340 green chlorophyll-rich vegetation strongly reflects) and red wavelength areas (which vegetation absorbs) computed using Eq. (6). MOD13Q1 products from MODIS are NDVI data generated from a 16-day interval at a 250 m spatial resolution as a Level 3 product (Didan, K, 2015).

$$NDVI = \frac{NIR+Red}{NIR+Red} \tag{6\underline{8}}$$

To cover our study period, we selected NDVI data within the campaign dates. If no data fitted within
our dates, we used data that were less than five days before or after the campaign dates, assuming that
no significant increase or decrease would occur in the vegetation. The pixels containing the study
sites were extracted based on the latitude and longitude of each site. Linear functions were applied to
the seasonal datasets of Rs with NDVI to assess the contribution of vegetation on soil emissions using
Eq. (7) and a combined effect of WC and NDVI on soil $CO_2$ emissions (Rs) using Eq. (8).

$$Rs = a + bNDVI \tag{79}$$

$$Rs = a + bNDVI + (cWC + dWC^2) \tag{810}$$

**3. Results**

**3.2. Meteorological data**

During the 12-month study period, the long rains lasted from early March to the end of May, while
short rains occurred between early September and December (Fig. 3). The total annual rainfall was
550 mm, which is within the average rainfall expected in the area (CIDP, 2014). The mean annual air
temperature was 22.7 °C (min=16.7 °C, max=30.5 °C). January was the hottest month (min=17.4 °C,
max=31.9 °C), while June and July (min=14.5± 0.2 °C, max=27± 0.1 °C) were the coolest.

[Figure]

*Figure 3: (a) Daily maximum and minimum air temperature and (b) daily rainfall from lowland in
southern Kenya between November 2017 to October 2018 recorded at Maktau weather station. Total
annual recorded rainfall was 550 mm. Highlighted grey bars show the days of the sampling*

365    *long dry season).*

**3.3. Soil characteristics**

Sand content was highest in cropland (77±8 %) compared to the conservation land and bushland (ca. 72±1 %) and lowest in grazing land (64.3 ±0.4 %) (See Table 2). Grazing land had the highest clay content (31.7±0.5 %) while cropland (19±2 %) had the lowest. Soil pH ranged between slightly acidic
370    in the grazing land (6.3±0.3), neutral in the bushland (7.2±0.4), and slightly alkaline in the conservation land and cropland (7.5±0.1 and 7.9±0.2 respectively). Carbon content ranged from 0.93 % in the conservation land to 0.60 % in the cropland. Nitrogen content did not vary significantly between sites (mean=0.08±0.01 %).

**Table 1:** Soil characteristics of the topsoil (a depth of 0–20 cm) from the four land-use types
375    investigated in this study. Values are given as mean ± SE.

| Land Use | % N | % C | Bulk Density (g cm$^{-3}$) | pH | Soil Texture | | |
| | | | | | % Clay | % Sand | % Silt |
|---|---|---|---|---|---|---|---|
| Bushland | 0.08 (0.03) | 0.77 (0.5) | 1.31 (0.2) | 7.2 (0.4) | 23.7 (0.7) | 71.6 (2.2) | 4.7 (2.3) |
| Conservation land | 0.09 (0.02) | 0.93 (0.7) | 1.27 (0.4) | 7.5 (0.1) | 26.4 (2.2) | 71.6 (0.5) | 2.0 (0.0) |
| Cropland | 0.07 (0.04) | 0.60 (0.2) | 1.26 (0.3) | 7.9 (0.2) | 19.1 (2.4) | 76.9 (8.1) | 4.0 (5.1) |
| Grazing land | 0.08 (0.02) | 0.83 (0.4) | 1.23 (0.5) | 6.3 (0.3) | 31.7 (0.5) | 64.3 (0.4) | 4.4 (0.4) |

**3.4. Soil greenhouse gas emissions**
**3.4.1.    Soil carbon dioxide ($CO_2$) emissions**

Mean annual soil $CO_2$ emissions were highest in the conservation land (75±6 mg $CO_2$-C m$^{-2}$ h$^{-1}$). Concurrently, no significant differences occurred between grazing land (50±5 mg $CO_2$-C m$^{-2}$ h$^{-1}$),
380    cropland (47±3 mg $CO_2$-C m$^{-2}$ h$^{-1}$), and bushland (45±4 mg $CO_2$-C m$^{-2}$ h$^{-1}$). We observed no significant difference in $CO_2$ emissions between the first three seasons, namely SD, onset-LW and mid-LW. However, towards the end of the wet season (end-LW) in May, $CO_2$ emissions in the conservation land and grazing land were significantly higher than cropland and bushland ($p < 0.05$). Through LD, onset-SW, and mid-SW, $CO_2$ emissions in the conservation land remained significantly
385    higher, while those from grazing land dropped during LD and were not different from bushland or cropland emissions thereafter.

Generally, $CO_2$ emissions were higher in the wet seasons than in the dry seasons at all sites (Fig. 4c). At the onset of the rainy season in early March, $CO_2$ emissions increased at all sites by over 200% from SD to LW and dropped during LD by approximately 70% in grazing land, bushland, and cropland. In the conservation land, the drop from LW to LD was about 20%. In the bushland, the highest seasonal mean fluxes were reached in mid-LW ($98\pm6$ mg $CO_2$-C m$^{-2}$ h$^{-1}$) while in the conservation land ($239\pm11$ mg $CO_2$-C m$^{-2}$ h$^{-1}$), grazing land ($160\pm16$ mg $CO_2$-C m$^{-2}$ h$^{-1}$), and cropland ($84\pm12$ mg $CO_2$-C m$^{-2}$ h$^{-1}$), the highest were observed during end-LW towards the end May. The lowest seasonal mean $CO_2$ emissions at all sites were observed during the SD campaign (below 20 mg $CO_2$-C m$^{-2}$ h$^{-1}$, Fig 4).

When comparing the two wet seasons (LW and SW), $CO_2$ emissions were 45 % (bushland), 55 % (conservation land), 56 % (cropland), and 57 % (grazing land) higher in LW than SW (Fig. 5a). For the two dry seasons, $CO_2$ emissions were significantly higher in LD than SD across all the sites (in SD all sites recorded emission below 20 mg $CO_2$-C m$^{-2}$ h$^{-1}$). During the LD, $CO_2$ emissions were 29 % (bushland), 38 % (cropland), 40 % (grazing land), and 77 % (conservation) higher than during SD. As much as $CO_2$ emissions in cropland, bushland, and grazing land dropped to less than 30 mg $CO_2$-C m$^{-2}$ h$^{-1}$ during LD, in the conservation land ($118\pm6$ mg $CO_2$-C m$^{-2}$ h$^{-1}$) the emissions remained high (Fig 4c).

**3.4.2. Soil nitrous oxide ($N_2O$) emissions**

Mean annual $N_2O$ emissions were very low (< 5 μg $N_2O$-N m$^{-2}$ h$^{-1}$) at all four sites (Fig. 4d). Cropland ($2.7\pm0.6$ μg $N_2O$-N m$^{-2}$ h$^{-1}$) recorded the highest mean $N_2O$ emissions than conservation land ($1.6\pm0.4$ μg $N_2O$-N m$^{-2}$ h$^{-1}$), grazing land ($1.5\pm0.4$ μg $N_2O$-N m$^{-2}$ h$^{-1}$), and bushland ($1.2\pm0.4$ μg $N_2O$-N m$^{-2}$ h$^{-1}$). $N_2O$ emissions did not show a clear temporal pattern as observed for $CO_2$ emissions. Within each season, no significant differences in $N_2O$ emissions were observed among the sites. However, at the onset of the rainy season (onset-LW), there were observable increases in $N_2O$ emissions from all the sites. During this period, mean $N_2O$ emissions at all the sites were ca. $2.6\pm0.4$ μg $N_2O$-N m$^{-2}$ h$^{-1}$. By mid-LW and end-LW periods, $N_2O$ emissions had dropped (<1 μg $N_2O$-N m$^{-2}$ h$^{-1}$) at all sites. In June during LD, $N_2O$ emissions from the cropland were significantly higher than at the other three sites ($2.35\pm0.03$ μg $N_2O$-N m$^{-2}$ h$^{-1}$, $p < 0.05$). During this period, the farmer had just harvested his crops.

When comparing the two wet seasons, $N_2O$ emissions did not differ between LW and SW at all sites (Fig. 5b). However, short $N_2O$ emission pulses were observed during both seasons. A notable peak

of about 70 μg $N_2O$-N $m^{-2} h^{-1}$ was observed in the cropland on 7 April 2018, a week after livestock manure application. It had also rained the night before the sampling day. At the same site, we also recorded a peak of 55.2 μg $N_2O$-N $m^{-2} h^{-1}$ on 30 September 2018, likely also due to manure application (personal communication from the farmer Mwadime Mjomba). Other notable peaks were 29.9 μg $N_2O$-N $m^{-2} h^{-1}$ (in the bushland on 3 September 2018) and 26.6 μg $N_2O$-N $m^{-2} h^{-1}$ (in grazing land on 4 September 2018). These were observed during the SW from chambers with animal droppings. For the dry seasons, $N_2O$ emissions did not differ between SD and LD in the bushland, conservation land, and grazing land, while $N_2O$ emissions in the cropland were significantly higher during LD than SD (Fig. 5b).

**3.4.3. Soil methane ($CH_4$) emissions**

Throughout the study period, $CH_4$ emissions did not vary significantly among sites and seasons (Fig. 4e and Fig. 5c). The studied sites were mostly $CH_4$ sinks rather than sources, and $CH_4$ fluxes were very low, ranging from -0.03 to 0.9 mg $CH_4$-C $m^{-2}$ $h^{-1}$ (Fig. 4e), often below the limit of detection (0.03 mg $CH_4$-C $m^{-2} h^{-1}$).

[Figure]

440 **Figure 4:** *Box plots showing differences in seasonal means for (a) soil moisture, (b) soil temperature,*
*and soil emissions of (c) $CO_2$, (d) $N_2O$, and (e) $CH_4$ for each site from November 2017 to October*
*2018. Season abbreviations on the x-axis denote SW for the short wet season and LW for the long wet*
*season with corresponding onset, mid and end of the wet season, along with SD for the short dry*
*season and LD for the long dry season.*

[Figure]

445

***Figure 5:*** *Seasonal differences in mean (a) $CO_2$, (b) $N_2O$, and (c) $CH_4$ emissions between the long and short wet seasons and the long and short dry season for the four land-use types.*

**3.5. Effects of soil temperature, soil water content, and vegetation indices on soil GHG emissions**

Soil water content (WC) was highest during the wet season (mean $0.19\pm0.06$ m$^3$ m$^{-3}$) and lowest
450   during the dry season (mean. $0.07\pm0.02$ m$^3$ m$^{-3}$) at all sites (see Fig. 4a). Soil temperature (T), were highest during the SD ($36.7\pm2.1$ $^o$C) and lowest ($24.5\pm0.6$ $^o$C) in LD (Fig.4b). Throughout all the campaigns, mean WC and mean T were highest in the conservation land, followed by grazing land, bushland, and lowest in the cropland. Regression results on soil $CO_2$ and soil $N_2O$ emissions against T and WC are shown in Table 2. The results showed positive correlations between soil $CO_2$ emissions
455   and WC ($p <0.05$). However, the $R^2$ was very weak at all sites. Conversely, $CO_2$ emissions showed no correlation with T ($P < 0.05$). We observed no correlation between $N_2O$ and $CH_4$ emissions with either WC or T ($p <0.05$). Separating data into the wet and dry season did not improve the correlations.

**Table 2:** Soil water content (WC) and soil temperature (T) control on carbon dioxide ($CO_2$) and nitrous oxide ($N_2O$) denoted by *Rs*, while *a, b,* and *c* denote the model coefficient

| Predictors | Land Use | $CO_2$-C mg m$^{-2}$ h$^{-1}$ | | $N_2O$-N ug m$^{-2}$ h$^{-1}$ | |
|---|---|---|---|---|---|
| Soil water content (WC) | | $Rs = a + bWC + cWC^2$ | | | |
| | Bushland | $6.12WC + 0.92WC^2$ | $R^2= 0.26$*** | $19.02WC - 64.11WC^2$ | $R^2= 0.008$ |
| | Conservation land | $135.27WC - 0.57WC^2$ | $R^2= 0.07$** | $11.63WC - 7.736WC^2$ | $R^2= 0.009$ |
| | Cropland | $17.83WC + 0.67WC^2$ | $R^2= 0.04$*** | $28.48WC - 66.63WC^2$ | $R^2= 0.005$ |
| | Grazing land | $15.03WC + 0.79WC^2$ | $R^2= 0.11$*** | $19.81WC - 53.56WC^2$ | $R^2= 0.002$ |
| Soil Temperature (T) | | $R = ae^{(bT+cT^2)}$ | | | |
| | Bushland | $1.078e^{0.26T-0.004T^2}$ | $R^2= 0.008$ | $360.25e^{-0.29T-0.004T^2}$ | $R^2= 0.008$ |
| | Conservation land | $0.001e^{0.81T-0.014T^2}$ | $R^2= 0.015$** | $0.007e^{0.45T-0.008T^2}$ | $R^2= 0.015$ |
| | Cropland | $4.568e^{-0.13T+0.002T^2}$ | $R^2= 0.008$* | $0.007e^{-0.05T+2.42T^2}$ | $R^2= 0.008$* |
| | Grazing land | $4.136e^{0.18T-0.003T^2}$ | $R^2= 0.015$ | $2.366e^{0.05T-0.001T^2}$ | $R^2= 0.015$ |

***: p<0.0001, **: p<0.001, *: p<0.05

Results from combined WC and T on soil $CO_2$ and $N_2O$ emissions did not improve the correlation, as shown in Table 3. Thus, we included vegetation indices in our model.

**Table 3:** Combined effects of soil water content (WC) and soil temperature (T) control on soil $CO_2$ and $N_2O$ emissions. Soil $CO_2$ and $N_2O$ emissions denoted by *Rs*, while *a, b, d,* and *e* signify the model coefficient, ($R^2$) the coefficient of determination, and AIC the Akaike's information criterion

| Functions | Land use | a | b | d | e | $R^2$ | AIC |
|---|---|---|---|---|---|---|---|
| **$CO_2$-C mg m$^{-2}$ h$^{-1}$** | | | | | | | |
| $Rs = e^{(aT+bT^2)} \times (dWC + eWC^2)$ | Bushland | -0.12 | 0.001 | 52.774 | -0.527 | 0.31*** | 1888 |
| | Conservation land | 0.90 | -0.016 | 0.0001 | 0.000 | 0.10*** | 2156 |
| | Cropland | -0.39 | 0.006 | 3701.901 | -84.001 | 0.08** | 1886 |
| | Grazing land | 0.14 | -0.003 | 0.842 | -0.008 | 0.12*** | 2024 |
| **$N_2O$-N ug m$^{-2}$ h$^{-1}$** | Bushland | -0.50 | 0.007 | 2008.345 | -58.559 | 0.009 | 785 |
| $Rs = e^{(aT+bT^2)} \times (dWC + eWC^2)$ | Conservation land | 0.56 | -0.010 | 0.000 | 0.000 | 0.003 | 811 |
| | Cropland | 0.67 | -0.017 | 0.003 | -0.0001 | 0.089 | 911 |
| | Grazing land | 0.11 | -0.003 | 0.187 | -0.005 | 0.003 | 770 |

***: p<0.0001, **: p<0.001, *: p<0.05

The annual change in vegetation cover at each site are shown in Fig. (6). The highest NDVI values were observed during the LW in April (ranging from 0.58 to 0.76) and the lowest during the SD (< 0.26). Vegetation greenness increased rapidly from mid-March at all sites coinciding with the onset of the rainy season and remained high (Fig.6). At end of the rainy season, NDVI gradually dropped. Highest NDVI values occurred in the conservation land (0.51±0.05); followed by bushland (0.44±0.05), cropland (0.41±0.05), and the lowest were recorded in the grazing land (0.33±0.05).

[Figure]

475 *Figure 6: Monthly NDVI time series showing the annual trend in vegetation cover from November 2017 to November 2018 for the four land-use types.*

Regression analysis results shows a positive correlation between NDVI and seasonal $CO_2$ emissions at all the sites (see Fig. 7). Combined WC and NDVI improved the correlation even further as shown in Table 4. No significant correlation was observed between $N_2O$ emissions and NDVI (Fig. 7).

[Figure]

480

ns= not significant

*Figure 7: Linear regression analyses of the measured seasonal means for soil $CO_2$ and soil $N_2O$ emissions during the campaign from November 2017 to November 2018 plotted against NDVI data acquired during each campaign.*

**Table 4:** Combined effects of soil water content (WC) and NDVI on soil $CO_2$ emissions. Soil $CO_2$ emissions denoted by *Rs*, while *a, b, c,* and *d* are the model coefficient and ($R^2$) the coefficient of determination.

| | Land use | a | b | c | d | $R^2$ |
|---|---|---|---|---|---|---|
| **$CO_2$-C mg m$^{-2}$ h$^{-1}$** | | | | | | |
| $Rs = a + bNDVI + (cWC + dWC^2)$ | Bushland | -29.69 | 196.47 | -83.74 | 781.29 | 0.86*** |
| | Conservation land | 6.48 | 382.96 | -861.98 | -256.66 | 0.82*** |
| | Cropland | 26.94 | 244.54 | -1250.22 | 3269.46 | 0.79*** |
| | Grazing land | -97.19 | 396.41 | 575.60 | -3440.80 | 0.96*** |

***: p<0.0001, **: p<0.001, *: p<0.05

**4. Discussion**

**4.1. Soil $CO_2$ emissions**

Soil $CO_2$ emissions differed significantly between the four LUTs. The highest mean $CO_2$ emissions were observed in the conservation land followed by grazing land and bushland, and the lowest from cropland. Soil C content, which is the primary source of energy for soil microorganisms that contribute to soil $CO_2$ emissions (Lal, 2009) also showed the same trend (conservation land > grazing land > bushland > cropland). Therefore, the difference in land use and land-use management activities between our sites played a vital role in modifying both biotic and abiotic factors that drive both soil C content and soil $CO_2$ emissions (Pinto et al., 2002).

Due to the difference in land use and management, vegetation type and cover differed between our sites. The dense grass network in the conservation land formed an almost closed ground cover, especially in the wet seasons (further confirmed by NDVI values). Being a private sanctuary, only wild mammals (no livestock allowed) grazed and browsed there, and thus we observed less damage on the grass cover throughout all the campaigns as compared to the grazing land (which had large patches of bare soil due to overgrazing) and bushland. This provides a good explanation for the difference in mean $CO_2$ emissions between these three LUTs, as vegetation is known to affect soil C concentration and root and microbial respiration that directly contribute to soil $CO_2$ emissions (Fanin et al., 2011; Rey et al., 2011).

With the lowest $CO_2$ emissions being measured in the cropland, we attribute this observation to the continued tillage and removal of crops and crop residues during land preparation, weeding and harvesting, which affects both root respiration and soil C content (Raich et al., 2000; Nandwa, 2001). In East Africa and especially in smallholder farming systems, most of the crop residues are used as livestock feed and fuel. In addition, manure inputs in cropland are very low (about 20 kg per month

on a 1.5 ha farm) and thus no measurable difference in $CO_2$ emissions was detected before and after manure input, and with the other LUTs. Several other studies observed the same scenario from low manure input in maize and sorghum plots (Rosenstock et al., 2016; Mapanda et al., 2011, and Pelster et al., 2017).

On average, $CO_2$ emissions were higher during the wet season than during the dry season. At the start of both rainy seasons (SW, LW), $CO_2$ emissions increased significantly in all LUTs. Emissions from the conservation land and grazing land are comparable to those in Brümmer et al. (2008), who observed $CO_2$ emissions ranging between 100 and 250 mg $CO_2$-C $m^{-2}$ $h^{-1}$ in a natural savanna in Burkina Faso. Several other studies from similar ecosystem have also documented comparable changes in $CO_2$ emissions with the onset of the rainy seasons (Castaldi et al., 2006; Livesley et al., 2011; Pinto et al., 2002). In the cropland, results in the wet season are similar to those measured by Rosenstock et al. (2016), ranging from 50 to > 200 mg $m^{-2}\,h^{-1}$. We attributed the increase in $CO_2$ emissions in the wet season to the response of soil microbes and vegetation to soil moisture (Livesley et al., 2011; Otieno et al., 2010). Soil moisture connects microorganisms with soluble substrates (Moyano et al., 2013) and increases microbial activity (Davidson et al., 2006; 2009; Grover et al., 2012) and thereby soil $CO_2$ emissions.

Furthermore, an increase in soil $CO_2$ emissions during the wet season can also be a result of increased root respiration due to more active plant and root growth (Macdonald et al., 2006). Grasses sprout more rapidly than trees and shrubs with the first rains (Merbold et al., 2009). This provides a possible explanation for the higher $CO_2$ emissions in the grassy conservation land, grazing land, and bushland compared to cropland during the rainy season. However, grazing land recorded higher $CO_2$ emissions than bushland (only the farmer's livestock grazed here). The main difference between these two sites – apart from grazing intensity – was that bushland had more trees (*Acacia spp.*) and shrubs (*Commiphora spp.*) and less herbaceous undergrowth than the grazing land, thus providing shade that might have interfered with growth and regrowth of plants below the canopy. Therefore, grass root production in the open conservation land and grazing land was likely higher than in the bushland (Janssens et al., 2001), although we cannot confirm this because root biomass was not determined in this study. In cropland, all grasses and weeds were cleared during regular weeding and therefore did not play a role in root respiration.

To our surprise, the highest mean seasonal $CO_2$ emissions in conservation land, grazing land, and cropland were observed at the end rather than at the peak of the wet season. During this time, both soil moisture and soil temperature had dropped in all LUTs. However, our data was only recorded up

to a depth of 5 cm, but roots of perennial grasses, shrubs and trees can tap moisture from greater soil
depths (Carbone et al., 2011). According to Carbone et al. (2011), while microbial activity is highest
and most variable in the upper soil layers, which are first to wet-up and dry-down, roots can access
water reserves in deeper soil layers that take longer to be exhausted, and therefore remain active at
the end of the wet and into the dry season.

**4.2. Soil $N_2O$ emissions**

Our results showed very low $N_2O$ emissions from all LUTs, which we attributed to low soil N content
observed in all the sites (see Table 1). Savanna ecosystems are characterized by very tight N cycling,
which transcends to low N availability (Pinto et al., 2002 and Grover et al., 2012), and most of this N
is rapidly taken up by vegetation, leaving very little for denitrification (Castaldi et al., 2006; Mapanda
et al., 2011). The $N_2O$ flux results observed from conservation land, grazing land and bushland are
consistent with those observed in a Brazilian savanna by Wilcke et al. (2005), and other studies from
similar ecosystems reported comparable $N_2O$ flux magnitudes (Scholes et al., 1997; Castaldi et al.,
2016; Mapanda et al., 2010). The higher $N_2O$ emissions observed in June and July from our cropland
site after the maize and bean harvests likely occurred due to the disturbance and following absence
of live plants, which led to higher soil N availability because of less N uptake by plants and increased
root decomposition.

In contrast to the patterns observed for $CO_2$ emissions, we did not detect any seasonal variations in
$N_2O$ emissions. The only exception to the otherwise very low $N_2O$ emissions was after the onset of
the rainy season, when $N_2O$ emissions slightly increased at all sites. Such patterns have previously
been shown by several similar studies (Scholes et al., 1997; Pinto et al., 2002; Castaldi et al., 2006;
Livesley et al., 2011). The increase in $N_2O$ flux at the onset of the rains has been attributed to an
increase in microbial activity and therefore faster decomposition of litter and plant residue facilitated
by an increase in soil moisture, thus increasing N availability (Rees et al., 2006). Furthermore,
according to Davidson et al., (2000) and Butterbach-Bahl et al., (2013), soil moisture affects soil gas
diffusion, oxygen ($O_2$) availability, and the movement of substrate necessary for microbial growth
and metabolism.

Negative $N_2O$ emissions were detected during the dry season. Such observations could result from
the low N contents observed at all sites coupled with low soil moisture in the dry season, which
facilitates diffusion of atmospheric $N_2O$ into the soil. Soil denitrifiers may, therefore, use $N_2O$ as an
N substrate in the absence of $NO_2^-$ and $NO_3^-$ (Rosenkranz et al., 2006). Negative $N_2O$ emissions have

575  also been reported in other tropical savanna soils under similarly dry conditions (Castaldi et al., 2006; Livesley et al., 2011).

Manure application in the cropland was very low (< 12 kg of N in 1.5ha for the crop-growing season), and thus $N_2O$ emissions from cropland were low and not different from the other LUTs, which was in contrast to what we had hypothesized. Due to low soil N levels in the cropland, the low amount of
580  manure added was not sufficient to stimulate $N_2O$ emissions, likely because soil N availability was still limiting for plant and microbial growth (Castaldi et al., 2006). Traditional farming systems in smallholder farms in Africa involve repeated cropping with no or very low N inputs that leads to soil N mining over time (Chianu et al., 2012). In line with this, in our cropland site maize and beans are grown during every wet season with no fallow in between years. In addition, the farmer did not use
585  any chemical fertilizer to increase soil N, and the N input from biological N fixation into the soil was likely small because beans were harvested for consumption and bean plant residues were used as livestock feed and not incorporated into the soil. Therefore, the small quantities of manure applied and legume N fixation may have likely been insufficient to compensate for N loss through leaching and crop harvests. According to the Taita Development plan, this is a common scenario in the county,
590  which translates to very low crop yields in this region (CIDP, 2014). Another possible explanation for not detecting the influence of manure on $N_2O$ emissions could be the fact that we did not manage to sample immediately after manure application and therefore might have missed the instant impact of manure application on $N_2O$ emissions. However, similar studies by Pelster et al. (2017) and Rosenstock et al. (2016) also did not see any influence of manure application on soil $N_2O$ emissions
595  and reported $N_2O$ emission values that were generally $< 10\ \mu g\ N_2O\text{-N}\ m^{-2}\ h^{-1}$). Equally, the deposition of dung and urine by animals in the grazing land and bushland did not have any measurable influence on soil $N_2O$ emissions.

**4.3. Soil $CH_4$ emissions**

Methane emissions did not vary between the land-use types or with seasons. Most values were below
600  the LOD at all the sites. Soil water content in our study is clearly the limiting factor for methanogenesis, which needs anoxic conditions for a certain period until methanogenic archaea are established (Serrano-silva et al., 2014). Furthermore, soil compaction by animal trampling may have limited $CH_4$ diffusion into the soil thus limiting $CH_4$ consumption by oxidation (Ball et al., 1997). In cropland, continuous tillage interferes with soil structure thus affecting the microenvironment that
605  favours methanotrophic (Jacinthe et al., 2014). Additionally, low soil C as observed in all the sites generally leads to low abundance of soil microorganisms and consequently also methane oxidisers

(Serrano-silva et al., 2014). Nevertheless, soils around lakes, waterholes and rivers can be $CH_4$ sources in semi-arid savanna ecosystems, but those were not investigated during this study.

**4.4. Effects of soil moisture, soil temperature, and vegetation indices on GHG emissions**

610    As is common for sub-tropical regions, seasonal variation in soil temperature was small in the study region and therefore soil temperature did not play a big role in modifying soil GHG emissions. Instead, changes in soil moisture were considered to be the main driver of $CO_2$ emissions in our study, as has previously been highlighted also by other studies (Grover et al. (2012), Brümmer et al. (2009) and Livesley et al. (2011)). However, we did not observe any significant relationship between

615    $N_2O$ emissions with either soil moisture or temperature apart from in the cropland, where we found a positive correlation between $N_2O$ and soil temperature ($p < 0.05$). As much as previous results have sometimes shown a positive relationship between temperature and $N_2O$ emissions (Castaldi et al., 2010), our results are in line with others (Scholes et al., (1997), Brümmer et al. (2008) who were also unable to link soil $N_2O$ emissions to variations in soil temperature. In fact, $N_2O$ emissions were very

620    low during both the wet and dry seasons, which is similar to the findings of Castaldi et al., (2004). The most likely explanation for the lack of seasonality effects on $N_2O$ emissions would be the low soil N levels observed at all the sites, which was probably the most limiting factor for $N_2O$ emissions and thus overruled all other potential controlling factors (Grover et al., 2012).

625    The vegetation cover as depicted by NDVI shows the status of the vegetation (NDVI values range from +1.0 to -.0). High NDVI values correspond to high vegetation cover, while low NDVI correspond to little or no vegetation (Gamon et al., 1995; Butt et al., 2011). Therefore, the increase in NDVI that we observed at the onset of the rainy season indicates sprouting and regrowth of vegetation at that time, while the drop in NDVI values at the end of the rainy season indicates reduction in vegetation cover due to plant senescence and grazing. In the cropland area, low NDVI

630    coincided with the harvesting of beans and the drying of the maize plants in June and July. Highest mean NDVI values were observed in the conservation land, mainly due to the dense grassy vegetation, while the lowest NDVI values were found in the grazing land, which we had expected because this area has large spots without vegetation due to overgrazing. Results from linear regression analysis showed a strong positive correlation of soil $CO_2$ emissions with NDVI ($p < 0.05$), explaining between

635    35 % and 82 % of the variation in soil $CO_2$ emissions at the four sites. This means that $CO_2$ emissions were highest when NDVI (i.e. vegetation cover) was high. Thus, the inclusion of both NDVI and soil moisture measurements is essential for reliably predicting soil $CO_2$ emission from savanna soils, which is consistent with other studies (Reichstein et al., 2003; Anderson et al., 2008; Lees et al.,

**Commented [WS15]:** RC: 264: the SI unit for time is seconds, but I am ok with presenting values per hour. (and 'represents'; minor usage errors should be reviewed throughout the manuscript, e.g. a missing period on p. 607 and 'value' on 595).

**Commented [WS16R15]:** We have corrected throughout the manuscript

2018). Concurrently, the same relationship between NDVI and $N_2O$ emissions could not be proven in our study.

**5. Conclusion**

The magnitude and temporal and spatial variability of soil GHG emissions in most developing countries have large uncertainties due to a lack of data, especially in dry areas and ecosystems facing land-use change. In our study, we quantified soil GHG emissions from four dominant LUTs in the dry lowlands of southern Kenya, namely bushland, conservation land, cropland, and grazing land. Our results showed significant variation between seasons and the respective LUTs. $CO_2$ emissions, in particular, were higher during the wet season, when soil moisture was high, compared to the dry season. Most of the variation in $CO_2$ emissions could be explained by soil moisture and NDVI, highlighting the importance of including proxies for vegetation cover in soil GHG emissions studies in savannas. $N_2O$ emissions and $CH_4$ emissions were of minor importance at all sites. However, we acknowledge that we might have missed some episodes of elevated soil $N_2O$ emissions, as these are often episodic and of short duration, for examples after fertilization or precipitation events. Following these results, there is still need for more continuous studies to cover spatial and temporal variation in soil GHG emissions, as well as the inclusion of other LUTs than the ones examined in this study (e.g. wetlands). Nevertheless, we believe that our results are important to reduce uncertainties in GHG emission baselines and to identify reliable and meaningful climate change mitigation interventions by informing the relevant policies.

**6. Data availability**

The data associated with the manuscript can be obtained from
https://figshare.com/articles/Final_data_for_Soil_Greenhouse_Gas_Emissions_under_Different_La nd-Use_Types_in_Savanna_Ecosystems_of_Kenya_/11673579

**Acknowledgments:** We acknowledge the Schlumberger Foundation under the Faculty for the Future programme for funding. The work was conducted under the Environmental sensing of ecosystem services for developing a climate-smart landscape framework to improve food security in East Africa, funded by the Academy of Finland (318645). A research permit from NACOSTI (P/18/97336/26355) is acknowledged. Taita Research Station of the University of Helsinki is acknowledged for technical and fieldwork support and Mazingira Centre of the International Livestock Research Institute for technical support in the laboratory work. Specifically, we would like to thank Mwadime Mjomba for

[revised manuscript text omitted]

Butterbach-Bahl, K, L Breuer, R Gasche, G Willibald, and H Papen.: "Exchange of Trace Gases between Soils and the Atmosphere in Scots Pine Forest Ecosystems of the Northeastern German

Lowlands: 1. Fluxes of N2O, NO/NO2 and CH4 at Forest Sites with Different N-Deposition." *Forest Ecology and Management* 167 (1): 123–34. https://doi.org/10.1016/S0378-1127(01)00725-3, 2002.

730   Carbone, Mariah S., Christopher J. Still, Anthony R. Ambrose, Todd E. Dawson, A. Park Williams, Claudia M. Boot, Sean M. Schaeffer, and Joshua P. Schimel.: "Seasonal and Episodic Moisture Controls on Plant and Microbial Contributions to Soil Respiration." Oecologia 167 (1): 265–78, 2011.

Carbone, Mariah S., Gregory C. Winston, and Susan E. Trumbore.: "Soil Respiration in Perennial Grass and Shrub Ecosystems: Linking Environmental Controls with Plant and Microbial Sources on
735   Seasonal and Diel Timescales." Journal of Geophysical Research: Biogeosciences 113 (G2). https://doi.org/10.1029/2007JG000611, 2008.

Castaldi, S., A. de Grandcourt, A. Rasile, U. Skiba, and R. Valentini.: "$CO_2$, $CH_4$ and $N_2O$ Fluxes from Soil of a Burned Grassland in Central Africa." Biogeosciences 7 (11): 3459–71. https://doi.org/10.5194/bg-7-3459-2010, 2010.

740   Castaldi, Simona, Antonella Ermice, and Sandro Strumia.: "Fluxes of $N_2O$ and $CH_4$ from Soils of Savannas and Seasonally-Dry Ecosystems." Journal of Biogeography 33 (3): 401–15. https://doi.org/10.1111/j.1365-2699.2005.01447.x, 2006.

Castaldi Simona, De Pascale Raffaele Ariangelo, Grace John, Nikonova Nina, Montes Ruben, and San José José.: "Nitrous Oxide and Methane Fluxes from Soils of the Orinoco Savanna under
745   Different Land Uses." Global Change Biology 10 (11): 1947–60. https://doi.org/10.1111/j.1365-2486.2004.00871.x, 2004.

Chapuis-Lardy, Lydie, Nicole Wrage, Aurélie Metay, Jean-Luc Chotte, and Martial Bernoux.: "Soils, a Sink for $N_2O$? A Review." *Global Change Biology* 13 (1): 1–17. https://doi.org/10.1111/j.1365-2486.2006.01280.x, 2007.

750   Chianu, Jonas N., Justina N. Chianu, and Franklin Mairura.: "Mineral Fertilizers in the Farming Systems of Sub-Saharan Africa. A Review." *Agronomy for Sustainable Development* 32 (2): 545–66. https://doi.org/10.1007/s13593-011-0050-0, 2012.

CIDP 2014: "Supporting Quality Life for the People of Taita Taveta" The First Taita Taveta County Integrated Development Plan 2013-2017. Accessed November 29, 2018.
755   https://ke.boell.org/sites/default/files/uploads/2014/05/revised_draft_cidp_30_april_2014_2.pdf, 2014.

Collier, Sarah M., Matthew D. Ruark, Lawrence G. Oates, William E. Jokela, and Curtis J. Dell.: "Measurement of Greenhouse Gas Flux from Agricultural Soils Using Static Chambers." JoVE (Journal of Visualized Experiments), no. 90 (August): e52110. https://doi.org/10.3791/52110, 2014.

760 Croghan, W., and Peter P. Egeghy.: "Methods of Dealing with Values Below the Limit of Detection using SAS". Southern SAS User Group, St. Petersburg, FL, September 22-24, Accessed at http://analytics.ncsu.edu/sesug/2003/SD08-Croghan.pdf, 2003.

Davidson, Eric A.: "The Contribution of Manure and Fertilizer Nitrogen to Atmospheric Nitrous Oxide since 1860." Nature Geoscience 2 (9): 659–62. https://doi.org/10.1038/ngeo608, 2009.

765 Davidson, Eric A., Elizabeth Belk, and Richard D. Boone.: "Soil Water Content and Temperature as Independent or Confounded Factors Controlling Soil Respiration in a Temperate Mixed Hardwood Forest." Global Change Biology 4 (2): 217–27. https://doi.org/10.1046/j.1365-2486.1998.00128.x, 1998.

Davidson, Eric A., and Ivan A. Janssens.: "Temperature Sensitivity of Soil Carbon Decomposition
770 and Feedbacks to Climate Change." Nature 440 (7081): 165–73. https://doi.org/10.1038/nature04514, 2006.

Didan, K.: "MOD13Q1 MODIS/Terra Vegetation Indices 16-Day L3 Global 250m SIN Grid V006." distributed by NASA EOSDIS Land Processes DAAC, https://doi.org/10.5067/MODIS/MOD13Q1.006, 2015. Accessed 2020-01-21.

775 Fanin, Nicolas, Stephan Hättenschwiler, Sandra Barantal, Heidy Schimann, and Nathalie Fromin.: "Does Variability in Litter Quality Determine Soil Microbial Respiration in an Amazonian Rainforest?" Soil Biology and Biochemistry 43 (5): 1014–22. https://doi.org/10.1016/j.soilbio.2011.01.018, 2011.

FAO 1996.: "Agro-Ecological Zoning Guidelines." FAO Soils Bulletin 76.
780 https://www.mpl.ird.fr/crea/taller-colombia/FAO/AGLL/pdfdocs/aeze.pdf, 1996.

Flechard, Christophe R., Albrecht Neftel, Markus Jocher, Christof Ammann, and Jürg Fuhrer.: "Bi-Directional Soil/Atmosphere N2O Exchange over Two Mown Grassland Systems with Contrasting Management Practices." *Global Change Biology* 11 (12): 2114–27. https://doi.org/10.1111/j.1365-2486.2005.01056.x, 2005.

785

Gamon, John A., Christopher B. Field, Michael L. Goulden, Kevin L. Griffin, Anne E. Hartley, Geeske Joel, Josep Peñuelas, and Riccardo Valentini.: "Relationships Between NDVI, Canopy Structure, and Photosynthesis in Three Californian Vegetation Types." Ecological Applications 5 (1): 28–41. https://doi.org/10.2307/1942049, 1995.

790   GoK. 2013: "Kenya Agricultural and Livestock Research Act, 2013 (No. 17 of 2013)." 2013. https://www.ecolex.org/details/legislation/kenya-agricultural-and-livestock-research-act-2013-no-17-of-2013-lex-faoc122139/, 2013.

Grace, John, Jose San Jose, Patrick Meir, Heloisa S. Miranda, and Ruben A. Montes.: "Productivity and Carbon Fluxes of Tropical Savannas." Journal of Biogeography 33 (3): 387–400.
795   https://doi.org/10.1111/j.1365-2699.2005.01448.x, 2006.

Grover, S. P. P., S. J. Livesley, L. B. Hutley, H. Jamali, B. Fest, J. Beringer, K. Butterbach-Bahl, and S. K. Arndt.: "Land Use Change and the Impact on Greenhouse Gas Exchange in North Australian Savanna Soils." Biogeosciences 9 (January): 423–37. https://doi.org/10.5194/bg-9-423-2012, 2012.

Hanson, R. S., and T. E. Hanson.: "Methanotrophic Bacteria." *Microbiological Reviews* 60 (2): 439–
800   71, 1996.

Hickman, Jonathan E., Cheryl A. Palm, Patrick Kiiti Mutuo, J. M. Melillo, and Jianwu Tang.: "Nitrous Oxide ($N_2O$) Emissions in Response to Increasing Fertilizer Addition in Maize (Zea Mays L.) Agriculture in Western Kenya." Nutrient Cycling in Agroecosystems 100: 177–87. https://doi.org/10.1007/s10705-014-9636-7, 2014.

805   Hickman, Jonathan E., Katherine L. Tully, Peter M. Groffman, Willy Diru, and Cheryl A. Palm.: "A Potential Tipping Point in Tropical Agriculture: Avoiding Rapid Increases in Nitrous Oxide Fluxes from Agricultural Intensification in Kenya." Journal of Geophysical Research: Biogeosciences 120 (5): 938–51. https://doi.org/10.1002/2015JG002913, 2015.

Hutchinson, G. L., and A. R. Mosier.: "Improved Soil Cover Method for Field Measurement of
810   Nitrous Oxide Fluxes 1." Soil Science Society of America Journal 45 (2): 311–16. https://doi.org/10.2136/sssaj1981.03615995004500020017x, 1981.

IPCC: Climate Change 2013: The Physical Science Basis." Rationale Reference. European Environment Agency. Accessed November 29, 2018. https://www.eea.europa.eu/data-and-maps/indicators/glaciers-2/ipcc-2013-climate-change-2013, 2013.

815   IPCC: 2019 Refinement to the 2006 IPCC Guidelines for National Greenhouse Gas Inventories https://www.ipcc.ch/site/assets/uploads/2019/06/SB-50_TFI-side-event_2019-Refinement.pdf, 2019.

Jacinthe, Pierre-André, Warren A. Dick, Rattan Lal, Raj K. Shrestha, and Serdar Bilen.: "Effects of No-till Duration on the Methane Oxidation Capacity of Alfisols." *Biology and Fertility of Soils* 50
820   (3): 477–86. https://doi.org/10.1007/s00374-013-0866-7, 2014.

K'Otuto, G. O., D. O. Otieno, B. Seo, H. O. Ogindo, and J. C. Onyango.: "Carbon Dioxide Exchange and Biomass Productivity of the Herbaceous Layer of a Managed Tropical Humid Savanna Ecosystem in Western Kenya." Journal of Plant Ecology 6 (4): 286–97. https://doi.org/10.1093/jpe/rts038, 2013.

825   Janssens, I. A., H. Lankreijer, G. Matteucci, A. S. Kowalski, N. Buchmann, D. Epron, K. Pilegaard.: "Productivity Overshadows Temperature in Determining Soil and Ecosystem Respiration across European Forests." *Global Change Biology* 7 (3): 269–78. https://doi.org/10.1046/j.1365-2486.2001.00412.x, 2001.

La Scala, N., J. Marques, G.T. Pereira, and J.E. Corá.: "Carbon Dioxide Emission Related to
830   Chemical Properties of a Tropical Bare Soil." *Soil Biology and Biochemistry* 32 (10): 1469–73. https://doi.org/10.1016/S0038-0717(00)00053-5, 2000.

Lal, R.: "Challenges and Opportunities in Soil Organic Matter Research." *European Journal of Soil Science* 60 (2): 158–69. https://doi.org/10.1111/j.1365-2389.2008.01114.x, 2009.

Lal, R.: "Soil Carbon Sequestration Impacts on Global Climate Change and Food Security." Science
835   304 (5677): 1623–27. https://doi.org/10.1126/science.1097396, 2004.

Lees, K. J., T. Quaife, R. R. E. Artz, M. Khomik, and J. M. Clark.: "Potential for Using Remote Sensing to Estimate Carbon Fluxes across Northern Peatlands – A Review." Science of the Total Environment 615 (February): 857–74. https://doi.org/10.1016/j.scitotenv.2017.09.103, 2018.

Livesley, Stephen J., Samantha Grover, Lindsay B. Hutley, Hizbullah Jamali, Klaus Butterbach-Bahl,
840   Benedikt Fest, Jason Beringer, and Stefan K. Arndt.: "Seasonal Variation and Fire Effects on $CH_4$, $N_2O$ and $CO_2$ Exchange in Savanna Soils of Northern Australia." Agricultural and Forest Meteorology 151 (11): 1440–52. https://doi.org/10.1016/j.agrformet.2011.02.001, 2011.

Macdonald, Lynne M., Eric Paterson, Lorna A. Dawson, and A James S. McDonald.: "Defoliation and Fertiliser Influences on the Soil Microbial Community Associated with Two Contrasting Lolium

845 Perenne Cultivars." Soil Biology and Biochemistry 38 (4): 674–82. https://doi.org/10.1016/j.soilbio.2005.06.017, 2006.

Mapanda, F., J. Mupini, M. Wuta, J. Nyamangara, and R. M. Rees.: "A Cross-Ecosystem Assessment of the Effects of Land Cover and Land Use on Soil Emission of Selected Greenhouse Gases and Related Soil Properties in Zimbabwe." European Journal of Soil Science 61 (5): 721–33.

850 https://doi.org/10.1111/j.1365-2389.2010.01266.x, 2010.

Mapanda, Farai, Menas Wuta, Justice Nyamangara, and Robert M. Rees.: "Effects of Organic and Mineral Fertilizer Nitrogen on Greenhouse Gas Emissions and Plant-Captured Carbon under Maize Cropping in Zimbabwe." Plant and Soil 343 (1): 67–81. https://doi.org/10.1007/s11104-011-0753-7, 2011.

855 Marteau, Romain, Benjamin Sultan, Vincent Moron, Agali Alhassane, Christian Baron, and Seydou B. Traoré.: "The Onset of the Rainy Season and Farmers' Sowing Strategy for Pearl Millet Cultivation in Southwest Niger." Agricultural and Forest Meteorology 151 (10): 1356–69. https://doi.org/10.1016/j.agrformet.2011.05.018, 2011.

Moyano, F.E., Manzoni, S., Chenu, C.: "Responses of soil heterotrophic respiration to moisture

860 availability: An exploration of processes and models." Soil Biology and Biochemistry 59, 72–85. doi:10.1016/j.soilbio.2013.01.002, 2013.

[revised manuscript text omitted]